# 1 Chironomid- and pollen-based quantitative climate reconstructions

# **for the post-Holsteinian (MIS 11b) in Central Europe**

- 3 Tomasz Polkowski<sup>1</sup>, Agnieszka Gruszczyńska<sup>1,7</sup>, Bartosz Kotrys<sup>2</sup>, Artur Górecki<sup>3</sup>, Anna Hrynowiecka<sup>4</sup>,
- 4 Marcin Żarski<sup>5</sup>, Mirosław Błaszkiewicz<sup>1</sup>, Jerzy Nitychoruk<sup>6</sup>, Monika Czajkowska<sup>1</sup>, Stefan Lauterbach<sup>1,8</sup>,
- 5 Michał Słowiński¹
- <sup>1</sup>Institute of Geography and Spatial Organization Polish Academy of Sciences, Warsaw, 00-818, Poland
- <sup>2</sup>Polish Geological Institute National Research Institute, Szczecin, 71-130, Poland
- 8 <sup>3</sup>Institute of Botany, Jagiellonian University, Cracow, 30-387, Poland
- <sup>4</sup>Polish Geological Institute National Research Institute, Gdańsk, 80-328, Poland
- <sup>5</sup>Polish Geological Institute National Research Institute, Warsaw, 00-975, Poland
- <sup>6</sup>Pope John Paul II State School of Higher Education, Biała Podlaska, 21-500, Poland
- <sup>7</sup>Faculty of Physics and Earth System Sciences, Leipzig University, Linnéstraße 5, 04103 Leipzig,
- 13 Germany
- <sup>8</sup>GFZ Helmholtz Centre for Geosciences, Section 4.6 Geomorphology, Working Group Terrestrial
- 15 Climate Archives, 14473 Potsdam, Germany

- 17 Correspondence to: Tomasz Polkowski (tomasz.polkowski@twarda.pan.pl)
- 18 **Abstract.** Investigating climatic and environmental changes during past interglacials is crucial to improve our understanding
- of the mechanisms that govern changes related to current global warming. Among the numerous proxies that can be used to
- 20 reconstruct past environmental and climatic conditions, pollen allows quantitative reconstructions of annual, warmest month
- 21 and coldest month air temperatures as well as precipitation sums, and Chironomidae larvae are widely used to infer past
- summer air temperature. Chironomidae have mostly been used for reconstructing Holocene and Late Weichselian summer
- 23 temperatures whilst there are only four sites in Europe with chironomid-based summer air temperature reconstructions for the
- Late Pleistocene and no such records for any Middle Pleistocene warm period as of the writing of this paper. In this study we
- present the first quantitative palaeoclimate reconstruction for the post-Holsteinian (Marine Isotope Stage (MIS) 11b) in
- Central Europe based on both pollen and fossil chironomid remains preserved in palaeolake sediments recovered from Krepa,
- southeastern Poland. Besides being used for the palaeoclimatic reconstruction, pollen analysis provides the biostratigraphic
- framework and a broader perspective of climate development at the end of Holsteinian Interglacial. Fossil Chironomidae
- assemblages at Krępa consist mainly of oligotrophic and mesotrophic taxa (e.g. Corynocera ambigua, Chironomus
- anthracinus-type) while eutrophic taxa (e.g. Chironomus plumosus-type) are less abundant. The chironomid-based summer
- temperature reconstruction indicates July air temperatures between 15.3 and 20.1°C during the early post-Holsteinian. In any

- case, results from Krepa prove that conducting Chironomidae analysis is feasible for periods as early as the Middle Pleistocene,
- improving our understanding of the mechanisms that control present-day climatic and environmental changes.

# 1 Introduction

Earth's history is characterised by repeated climate fluctuations, which had not been influenced by humans until the Holocene (marine isotope stage (MIS) 1), the most recent interglacial period. This offers the opportunity to compare natural climatic changes in the past with current ones in order to assess anthropogenic impact on the present climate. With respect to human impact during the Holocene, the so-called "Anthropocene" is widely debated across various scientific disciplines though its exact timing, and the actual dimension of human influence on the environment are still debated (Brondizio et al., 2016). Holocene environmental archives, such as lake, palaeolake and ocean sediments provide material for comprehensive palaeoecological analyses. The sensitivity of some groups of organisms in these archives to changing hydrological or climatic conditions allows reconstruction of past events that directly affected the abundance or structure of the communities (Battarbee, 2000). Species, which are characterised by narrow ecological preferences, such as air temperature, water chemistry or water depth, are used for certain palaeoenvironmental reconstructions (Juggins and Birks, 2012). Many ecological parameters can be reconstructed using different proxies. For example, foraminiferas can be used to reconstruct ocean pH (Foster and Rae, 2016; Roberts et al., 2018), pollen can provide information about vegetation changes (Ralska-Jasiewiczowa et al., 2004; Kupryjanowicz et al., 2018) and can be used to reconstruct past human activity (Chevalier et al., 2020) or past climate conditions (e.g. Rylova and Savachenko, 2005; Hrynowiecka and Winter, 2016). Head capsules of chironomids can serve as the basis for summer air temperature reconstructions (Eggermont and Heiri, 2012) and for assessing the trophic state or pH of freshwater ecosystems (Płóciennik, 2005). In general, palaeoecological and palaeoclimatological reconstructions record human impact on the environment from the Iron Age (Dumayne-Peaty, 1998; Szal et al., 2014). However, these reconstructions neither provide unequivocal information about air temperature changes nor allow the relative contribution of natural and human drivers to be distinguished. To gain a deeper understanding of the present human impact on climate and environment, it is therefore essential to investigate natural climate variability and environmental changes during past warm periods prior to any anthropogenic impact. In this regard, a particularly suitable targets are interglacial periods, e.g. Holsteinian Interglacial (or Mazovian Interglacial in Poland), which is commonly estimated to have lasted from 423 to 395 ka BP, thus corresponding to MIS 11c (Lauer and Weiss, 2018; Lauer et al., 2020; Fernández Arias et al., 2023). Holsteinian Interglacial is considered the analogue of the Holocene in terms of astronomical parameters (eccentricity, precession, insolation), climatic conditions and greenhouse gases levels (Koutsodendris et al., 2010; Yin and Berger, 2012; Kleinen et al., 2016). To date, there are only a few chironomid-based reconstructions of climatic and ecological conditions for the Middle and Late Pleistocene in Europe available (Engels et al., 2008; Bolland et al., 2021; Ilyashuk et al., 2022; Lapellegerie et al., 2024; Rigterink et al., 2024), but none for the Holsteinian Interglacial and the time thereafter. Hence, knowledge about climatic conditions at this time is mainly derived from pollen data, e.g. from the Praclaux maar in southern France (Reille and de Beaulieu, 1995), Tenaghi Philippon in north-eastern Greece (Tzedakis et al., 2006; Ardenghi et al., 2019). Lake Ohrid on the North Macedonian-Albanian border (Kousis et al., 2018). Lake Fucino in central Italy (Vera-Polo et al., 2024) and marine core from ODP site 976 in the Alboran Sea (Sassoon et al., 2023, 2025). In Central Europe, high-resolution MIS 11 pollen records are available from the Ossówka palaeolake in eastern Poland (Nitychoruk et al., 2005, 2018; Bińka et al., 2023) as well as from Nowiny Żukowskie in eastern Poland (Hrynowiecka and Winter, 2016) and Dethlingen in northern Germany (Koutsodendris et al., 2010). Also notable is Bilhausen in central Germany, which provided a pollen record for the so-called Bilshausen Interglacial, which might correspond to MIS 11 or MIS 13 (Kühl and Gobet, 2010). In Northern Europe, there are even fewer records covering MIS 11 e.g. the record from Hoxne in eastern England (Horne et al., 2023) where temperature reconstructions were performed using chironomids (e.g., (Brooks, 2006), ostracods (Horne, 2007) and beetle remains (Atkinson et al., 1987). The contemporary state of knowledge on MIS 11 has been reviewed by Candy et al. (2014). Climate conditions in Central Europe were in general temperate at that time (Nitychoruk et al., 2018), but vegetation reconstructions suggest warmer and more humid conditions compared to the Holocene climatic optimum (Hrynowiecka and Winter, 2016). Two major climatic

Europe were in general temperate at that time (Nitychoruk et al., 2018), but vegetation reconstructions suggest warmer and more humid conditions compared to the Holocene climatic optimum (Hrynowiecka and Winter, 2016). Two major climatic oscillations have so far been documented during the Holsteinian Interglacial, the Older Holsteinian Oscillation (OHO) and the Younger Holsteinian Oscillation (YHO). The OHO occurred around 418 ka BP (Koutsodendris et al., 2010, 2012; Górecki, 2023) and is clearly connected to a rapid cooling as indicated by the disappearance of temperate vegetation (mostly *Picea-Alnus* forests) and the spread of pioneer tree taxa including *Betula*, *Pinus* and *Larix* (Koutsodendris et al., 2010, 2012; Candy et al., 2014; Hrynowiecka and Pidek, 2017; Górecki et al., 2022). Although the OHO has been described at multiple sites across northern Europe (Koutsodendris et al., 2012), it has so far been identified in few southern European sites (Kousis et al., 2018; Sassoon et al., 2023, 2025). In contrast to the OHO, the YHO occurred around 400 ka BP within the climatic optimum of the Holsteinian Interglacial (*Carpinus-Abies* phase) and was apparently not connected to a significant cooling (Górecki et al., 2022). Records from Germany and eastern Poland suggest a sudden regression of *Carpinus* from forest communities (Koutsodendris et al., 2010; Hrynowiecka et al., 2019; Górecki et al., 2022). Particularly in Poland a rapid spread of *Abies* with an admixture of *Corylus* is observed, with *Taxus* also found in southern sites (Górecki et al., 2022), suggesting that temperature was not limiting the growth of *Carpinus*.

The climate during Holsteinian Interglacial (MIS 11c) was characterised by relatively stable warm and moist conditions with global temperatures approximately 1.5–2 °C above the pre-industrial level (Masson-Delmotte et al., 2010). Raymo and Mitrovica (2012) and Muhs et al. (2012)suggest sea level was possibly 6–13 m higher than present in this period. This can be partially attributed to the melting of the Greenland Ice Sheet (Robinson et al., 2017), as pollen and palaeoDNA data suggest the existence of spruce forests in Greenland at this time (Willerslev et al., 2007; de Vernal and Hillaire-Marcel, 2008).

In Europe, warm and wet oceanic climate conditions extended far to the east as evidenced by the presence of *Taxus* and *Abies* pollen at sites in Lithuania (Kondratiene and Gudelis, 1983), Belarus (Mamakowa and Rylova, 2007), and western Ukraine (Łanczont et al., 2003; Benham et al., 2016), whilst modern distribution limits of these taxa are estimated further to the west (Benham et al., 2016). Evidence from several terrestrial records from Eurasia suggests that the MIS 11c climate was highly

2010; Prokopenko et al., 2010; Tzedakis, 2010; Oliveira et al., 2016; Tye et al., 2016; Górecki et al., 2022). Holsteinian 100 Interglacial was followed by gradual cooling period (MIS 11b) which resulted in annual temperature decline and forest 101 contractions (Tzedakis et al., 2006; Kousis et al., 2018; Hrynowiecka et al., 2019; Sassoon et al., 2025). 102 The pollen succession of the Holsteinian Interglacial in Poland is characterised by a dominance of *Picea-Alnus* at first, then 103 Carpinus and Abies, as well as a significant proportion of Taxus. Thermophilic taxa also occur frequently, examples including: 104 Pterocarya, Celtis, Juglans, Ilex, Carya, Parrotia, Buxus, Vitis, Brasenia, Trapa, and Azolla (Janczyk-Kopikowa, 1991). 105 Temperature reconstructions based on the indicator species method suggest the warmest period was the Carpinus-Abies phase, 106 with estimated temperatures of 0-3 °C in January and 21-26 °C in July. This, along with high precipitation created a suitable 107 environment for the spread of rare warmth-adapted taxa (Krupiński, 1995; Hrynowiecka and Winter, 2016). However, 108 palaeotemperature reconstructions from Dethlingen (Koutsodendris et al., 2012) suggest slightly lower temperatures in 109 Western Europe for both January (-2.2  $\pm$  3.1 °C) and July (17.8  $\pm$  2.1 °C). Pollen-based temperature reconstruction from Lake 110 Ohrid (SE Europe) indicates higher January (MTCO) maximum (4.4 °C) (Kousis et al., 2018). 111 MIS 11b brought the AP percentages decrease in Central Europe (Hrynowiecka et al., 2019). Lake Ohrid pollen record reveals 112 the domination of *Pinus* and plant open communities at the time, with Poaceae and *Artemisia* species included (Kousis et al., 113 2018). ODP Site 976 pollen-based climate reconstructions shows annual temperature drop to around 10 °C and summer 114 temperature to 20 °C (Sassoon et al., 2025). 115 The warm phase of the Holsteinian Interglacial was also confirmed by oxygen isotope analyses on endogenic lake carbonates 116 (Nitychoruk et al., 2005) and snail shells (Szymanek, 2018). These showed significant changes in climatic conditions 117 throughout the Holsteinian Interglacial, during which, continental and maritime influences intertwined in Central Europe. 118 Continental influences resulted in a shortened vegetation period with long winters, whilst the opposite occurred under maritime 119 influence, i.e. the vegetation period was significantly longer, temperatures were milder and precipitation rates were higher. 120 also reflected by the appearance of stenothermal plant species (Nitychoruk et al., 2005). Aiming at improving the knowledge 121 about climate variability at the demise of the Holsteinian Interglacial, we present the first quantitative climate reconstructions 122 for the post-Holsteinian in Central Europe, based on chironomid and pollen analyses. The aim of analysing this post-interglacial 123 period is to investigate temperature and vegetation changes and to determine if climate at the time was considerably cooler 124 than today. This choice was also dictated by Chironomidae head capsules' presence in post-Holsteinian section of the core 125 (unlike the Holsteinian part). In addition, we discuss the potential of chironomid analysis for palaeoecological study of 126 Quaternary sediments as well as the challenges for chironomid analysis arising from both the evolution and interchanging 127 adaptations to species ecological preferences and the preservation of fossil remains.

complex, with pronounced climate variability occurring on both centennial and millennial timescales (Koutsodendris et al.,

# 2 Study site and methods

#### 2.1 Study area

133134

glaciation.

The Krepa palaeolake sediment succession (51°37'53.2"N, 22°18'38.1"E, 146 m asl.) is located in SE Poland, near the city of Kock, approximately 120 km southeast of Warsaw (Fig. 1). It is under influence of humid continental climate (Dfb) in terms of the Köppen-Geiger climate classification (Peel et al., 2007). Average annual temperature for this region is ~ 8.6 °C, with July mean temperature of ~ 19 °C and January mean temperature of ~ -1 °C, while average annual precipitation is ca. 600 mm (Ustrnul et al., 2021). Geomorphologically, it is situated in the central-eastern part of the North European Plain behind the maximum extent of the Saalian glaciation (Marks et al., 2018) and the sediment core analysed in this paper was obtained on a moraine plateau related to this ice sheet. Holsteinian Interglacial deposits in the area were first identified by Jesionkiewicz (1982) during cartographic work for the 1:50 000 Detailed Geological Map of Poland (DGMP; Sheet 676 - Kock) (Drozd and Trzepla, 2007). On the moraine plateau, the interglacial deposits are found under a thin cover of moraine deposits, whereas at the slopes of the nearby Wieprz River valley, they are exposed directly on the surface. This study's material was obtained from a sediment core that was drilled at Krepa in 2015, using a Geoprobe drilling device (Górecki, 2023). The basal part of the 23.8-m-long sediment core that was recovered from the Krepa sediment succession in 2015 (Fig. 1) consisted of a 2-m-thick layer of light grevish brown sandy clays with a large number of rock fragments (unit 1), which is interpreted as till. As indicated by its stratigraphic position and its petrographic characteristics (Drozd and Trzepla, 2007), this till was likely accumulated during the Elsterian glaciation (Sanian 2 glaciation in Poland), which is considered to correspond to MIS 12. Directly above the till, a 0.6-m-thick layer of laminated sandy silts and sandy-clayey silts is present (unit 2). These sediments are interpreted as the result of glaciolimnic sedimentation in a relatively shallow water body between blocks of dead ice during the recession of the Elsterian ice-sheet. The glaciolimnic sediments of unit 2 gradually turn into a carbonate gyttja with small interlayers of carbonatic-minerogenic gyttja (unit 3), which was most likely deposited in the profundal zone of an already relatively deep lake. Between 1187 and 760 cm core depth, non-carbonatic organic-minerogenic gyttjas are found with mineral content generally increasing towards the top of the core (unit 4). The limnic sediments of unit 4 are interpreted to reflect the gradual shallowing of the lake due to continuous sediment infilling. At the same time, the systematic increase in mineral components in the sediments most likely reflects increased denudation and erosion in the catchment, possibly favoured by reduced vegetation cover in response to a climatic shift towards colder conditions. The gyttja sequence of unit 4 is overlain by a 1.9-m-thick layer of clays (unit 5), which probably represent accumulation in a periglacial lake. The following 1.1-mthick layer of fine- to medium-grained sands (unit 6) as well as the overlying 3.1-m-thick layer of rhythmically laminated sandy silts (unit 7) are interpreted as proglacial sediments (units 6 and 7) of the transgressing Early Saalian (MIS 6) ice sheet. Above this, the profile is capped by a 1.5-m-thick layer of sandy morainic till with rock fragments (unit 8) related to Saalian

Figure 1: (a) Location of selected sites with deposits from the Holsteinian Interglacial in Poland with the Krępa site indicated by the big yellow dot. Glaciation ranges are based on Żarski et al. (2024), Pochocka-Szwarc et al. (2024) and Marks (2023). (b) Lithological profile of the Krępa sediment succession and (c) location of the drilling site (picture M. Żarski).

#### 2.2 Chironomidae analysis

Initially, 79 sediment samples of 1 cm³, taken between 800 and 2160 cm depth at 5-40 cm intervals, were investigated for the presence of Chironomidae head capsules. However, only 30 of them (965-1155 cm depth) simultaneously contained more than 0-2 individuals. Chemical preparation followed Brooks et al. (2007). The precipitate was initially heated with KOH. The wet sediment was then passed through 212 μm (to remove larger sediment particles) and 100 μm mesh sieves and subsequent residues were treated in an ultrasonic bath for 3 sec. The processed sediment was subsequently examined under a stereomicroscope (Zeiss Axio Lab A1) at 25× magnification. Chironomid head capsules from each sample were picked and mounted in Euparal. In case of damaged head capsules, individuals were counted as one if more than half of a body was preserved. Identification of chironomid head capsules followed Wiederholm (1983), Schmid (1993), Klink and Moller Pillot (2003), Brooks et al. (2007) and Andersen et al. (2013). Ecological preferences of identified taxa are based mainly on Brooks et al. (2007), Brundin (1949), Brodersen and Lindegaard (1999a) and Saether (1979).

Preliminary tests of sample preparation avoided the use of chemicals and included soaking the samples in water for a long time instead to reduce mechanical stress exerted to the head capsules during sample sieving as much as possible. Nevertheless, intact head capsules could not be extracted from some sediment samples even when using this gentle way of sample preparation, likely because of the already highly compacted sediment. As small numbers of head capsules may hinder palaeoecological and palaeoclimatic reconstructions, it was therefore partly necessary to combine samples (see below) or to increase the volume of the analysed sediment material (some samples were even as large as 20 cm3).

Chironomidae subfossil larvae were obtained from a total of 30 samples from the gyttja sediments (unit 4 on Fig. 1). Samples that contained fewer than 50 head capsules were merged except for a solitary sample at 1000 cm core depth. For 5 samples the required number of 50 head capsules was obtained and the remaining 24 samples were merged into seven clusters. After merging, sample clusters at 975 cm, 1080 cm, 1120 cm and 1125 cm core depth still did not reach 50 head capsules, but nonetheless, these samples and the one from 1000 cm core depth were included in the reconstruction as preliminary results seemed credible in terms of obtained temperature values.

#### 2.3 Chironomid-based mean July air temperature reconstruction

In order to reconstruct mean July air temperatures ( $T_{jul-Ch}$ ) from the Krępa chironomid assemblage, the Swiss-Norwegian-Polish (SNP) training set (Kotrys et al., 2020) was used as this covers a higher temperature span than other available European training sets (e.g. the Finnish, Russian, Swiss-Norwegian training sets) (Kotrys et al., 2020). The SNP training set includes 357 lakes, 134 taxa, covers a temperature range between 3.5 and 20.1 °C. Weighted averaging-partial least squares transfer function (WA-PLS) was used for performing the reconstruction. The RMSEP for this combined training set is 1.39 °C, and the  $R^2$  is 0.91 (Kotrys et al., 2020). Detrended Correspondence (MinDC) was also calculated. The temperature reconstruction was carried out using the C2 (v. 1.6) software (Juggins, 2007).

The lowest number of head capsules used for the T<sub>jul-Ch</sub> reconstruction was 5 individuals at 1070 cm core depth whereas the highest number was 78 at 985 cm core depth. After merging, the total number of samples used for the T<sub>jul-Ch</sub> reconstruction was 13.

# 2.4 Pollen analysis

The Krępa sediment core obtained in 2015 was sampled for palynological analyses at 5-cm intervals between 770 and 2180 cm depth, totaling 281 samples. A volume of 1 cm³ was collected from organic sediments (peat, gyttja), while minerogenic sediments (clays, silts, sands) were sampled with a volume of 3 cm³ due to the anticipated low pollen grain concentration. Samples were further processed following the standard methodology outlined by Erdtman (1960) with modifications such as the use of HF (Berglund and Ralska-Jasiewiczowa, 1986). Prior to laboratory processing, a *Lycopodium* tablet (Lund University, batch number 100320201, 20,408±543 spores per tablet) was added to each sample to determine the absolute sporomorph concentration (Stockmarr, 1971). Pollen grains were counted using a ZEISS Axio Imager A2 light microscope. Palynomorphs were identified using pollen keys and atlases (Beug, 1961; Stuchlik, 2001, 2002, 2009; Lenarczyk, 2014), as well as online resources (PalDat, 2000; NPP Database, Shumilovskikh et al., 2022). For most samples, counts were conducted up to a sum of 500 pollen grains from arboreal (AP) and non-arboreal (NAP) plants. However, samples from glacial sediments with low palynomorph concentration were counted up to a sum of 300 pollen grains only. Percentages were calculated based on the sum of pollen grains from trees and shrubs (AP), as well as herbaceous plants, and dwarf shrubs (NAP). The results of the palynological analysis are depicted in a simplified pollen diagram (Fig. 3) that was plotted using R Studio with the package riojaPlot (Juggins, 2022). Local Pollen Assemblage Zones (LPAZ) were established using the CONISS cluster analysis function within riojaPlot and were visually adjusted if necessary.

#### 2.5 Pollen-based climate reconstructions

Climate variables reconstructed using pollen data include mean annual air temperature (TANN), mean annual precipitation (PANN), mean temperature of the warmest month (MTWA), and mean temperature of the coldest month (MTCO). All reconstructed climatic factors were based on modern data sourced from the Northern Hemisphere database compiled by Herzschuh et al. (2023a, b). Two reconstruction approaches were applied: the Modern Analog Technique (MAT; (Overpeck et al., 1985; Guiot, 1990) and Weighted Averaging Partial Least Squares regression (WA-PLS; (ter Braak et al., 1993; ter Braak and Juggins, 1993). In the MAT approach, the best number of analogues (k) was chosen by comparing model performance (RMSE and R²) across k values from 1 to 10. This analysis indicated that using k = 7 nearest analogues minimised prediction error, and thus 7 analogues were used in the final MAT reconstructions. For WA-PLS model selection, including the determination of the optimal number of components, was based on predictive accuracy assessed through leave-one-out (LOO) cross-validation and supported by randomization tests, following the methodology outlined by Chevalier et al. (2020). Based on these criteria, a four component WA-PLS was adopted. For each reconstruction model, the coefficient of determination (R²) and root mean square error (RMSE) were calculated to evaluate model performance. To express the

uncertainty in the fossil climate reconstructions, we calculated standard errors of prediction (SEP) and depicted them as error bars in the figures. In the WA-PLS approach, sample-specific SEP were obtained via a bootstrapping implemented in the rioja package (Juggins, 2022). For the MAT model we used the cross-validated RMSE as a uniform error estimate for the fossil MAT reconstructions. Modern pollen data used in the reconstructions were sourced from the Northern Hemisphere database compiled by Herzschuh et al. (2023a, b). To enhance spatial relevance, the modern dataset was geographically filtered to include only samples within a 3000 km radius of the fossil site. This geographic filtering yielded a regional calibration set of 4955 modern pollen samples, out of the original global dataset. From the fossil pollen dataset, only taxa present in at least 50% of the samples and reaching at least 1% pollen value at least once were included. Additionally we ensured taxonomic consistency between the modern and fossil pollen data by harmonizing taxa names and then removing taxa with zero abundance in the filtered modern set. After this filtering, 10 pollen taxa remained in common between the modern calibration set and the fossil record (primarily major arboreal and herb taxa such as *Larix*, *Betula*, *Pinus*, *Salix*, *Picea*, *Juniperus*, *Artemisia*, Asteraceae, Poaceae, and Amaranthaceae). Using only these common taxa helps avoid noise from spurious taxa and improves model robustness. All data processing and modeling were carried out in R (RStudio), making use of the analogue and rioja packages for calibration and reconstruction. The pollen-based reconstructions were restricted to the interval of the succession where chironomid remains were also present and were performed on 44 samples.

# 3. Results and interpretation

#### 3.1 Ecological reconstruction based on Chironomidae assemblages from the Krepa site

In general, the chironomid assemblages preserved in the Krępa sediments are dominated by two species *Corynocera ambigua* and *Chironomus anthracinus*-type. *Corynocera ambigua* is a species often described as cold-adapted oligotrophic (Fjellberg, 1972; Pinder and Reiss, 1983; Walker and Mathewes, 1988; Brooks et al., 2007; Luoto et al., 2008; van Asch et al., 2012), inhabiting shallow lakes in arctic and subarctic regions, though it is also found in eutrophic lakes (Halkiewicz, 2008; Kotrys et al., 2020) Adults of this species are not able to fly, and breed on the water surface when the temperature reaches approximately 7-8 °C (optimum 13.7 °C). Mothes (1968) concluded that *Corynocera ambigua* larvae develop in autumn and winter, but not during summer. The decline in their numbers may be due to growth of filamentous algae in summer. Larvae of *Corynocera ambigua* are eurythermic, while the pupae are cold-stenothermic (Brundin, 1949). They only reproduce at low temperatures and inhabit water bodies with a maximum depth of approximately 25 m. The abundance of *Corynocera ambigua* has been shown to be correlated with charophyte contents (Brodersen and Lindegaard, 1999b). Although this species does not feed on charophytes, their presence may increase the number of diatoms and stabilise the trophic status and water clarity (Forsberg, 1965; Blindow, 1992). *Corynocera ambigua* live in dendritic tubes, its main food source being diatoms/algal detritus. (Fjellberg, 1972; Boubee, 1983).

This species has been recorded during cold episodes or glacial periods, at sites in England (Bedford et al., 2004), Norway

(Velle et al., 2005), Poland (Płóciennik et al., 2015), and the Baltic region (Hofmann and Winn, 2000). However, Corynocera

ambigua, cannot be considered merely a cold species. Some authors believe that its occurrence depends on high oxygen content in the water (Brodersen and Lindegaard, 1999a) and for other authors, it is a pioneer species that appears first after glacial retreat, similarly to *Chironomus anthracinus*-type (e.g. Heiri and Millet, 2005; Ilyashuk et al., 2005, 2013, 2022; Gandouin et al., 2016). Luoto and Sarmaja-Korjonen (2011) suggest this is how the species adapts to existing climatic conditions. The locally observed decline in *Corynocera ambigua* numbers in the Krępa sediments could also be attributed to changes in lake productivity related to changes in the environment. For example, when the production of soil and trees increased, the number of this species has been found to decrease (Magny et al., 2006; Larocque-Tobler et al., 2009).

Chironomus anthracinus-type occurs in various lake zones and is capable of surviving approximately 2–4 months of oxygen

deficiency in water (Hamburger et al., 1994). It is a species which easily occupies niches that are inaccessible to others. According to some authors, it is a eutrophic (Kansanen, 1985; Brodersen and Lindegaard, 1999b) or cold-adapted species (Rohrig et al., 2004; Brooks et al., 2007; Płóciennik et al., 2011) and prefers soft, more organic sediments (McGarrigle, 1980). Therefore, the appearance of *Chironomus anthracinus*-type and *Glyptotendipes pallens*-type in the Krępa sediment may indicate the onset of eutrophication. Both *Chironomus anthracinus*-type and *Corynocera ambigua* are found in stratified lakes (e.g., Saether, 1979; Heiri, 2004). As we observe in our record, both species are relatively resistant to unfavourable environmental conditions, so possess a wide range of conditions in which they can occur.

Lower part of the sediment sequence (2180 cm-1160 cm) is almost completely devoid of Chironomid remains, except few badly preserved Chironomus anthracinus. Chironomus plumosus and Glyptotendipes pallens head capsules at 2000 cm. 1680 cm, and 1205-1190 cm depths. Head capsules are recorded again at 1155 cm-1122.5 cm depths – mostly Corynocera ambigua, Chironomus anthracinus, Chironomus plumosus and Glyptotendipes pallens (Fig. 2). 1072.5-1122.5 cm part predominantly contains cold-adapted species such as Corynocera ambigua and freeze-resistant species such as Glyptotendipes pallens-type and Glyptotendipes severini-type, which are often associated with algae and diatoms or mine leaves (Tarkowska-Kukuryk, 2014). 1022.5-1072.5 cm depth range is characterised by species highly resistant to difficult environmental conditions, such as Chironomus anthracinus-type, Corynocera ambigua and Glyptotendipes pallens-type From 1022.5 cm to 967.5 cm depth there are several species observed. This is the part most abundant in Chironomidae head capsules, with over 40 individuals per sample on average and maximum 78 head capsules at 985 cm sample. Species composition during this part is dominated by Corynocera ambigua, Chironomus anthracinus-type, Chironomus-plumosus-type and Propsilocerus lacustris-type. Additionally, some *Tanytarsus glabrecens*-type head capsules appear – this species was almost unseen in remaining sections. Between 967.5 cm and 877.5 cm depth, the number of chironomid head capsules started to decline above 965 cm depth, with only single unidentified Chironomidae head capsules at 955 cm and 950 cm. . . In subsequent section (877.5-765 cm) the number of Chironomidae is very low - only 2 Chironomus plumosus-type individuals were identified. Even Corynocera ambigua, abundant in previous sections, disappears.

Figure 2: Stratigraphic diagram of the Chironomidae assemblages. Caption: Chironomidae species are presented as counted numbers of specimens.

#### 3.2 July air temperature reconstruction based on Chironomidae assemblages from the Krepa site

Due to the low number of chironomid head capsules preserved in the Krępa sediments, a chironomid-based July temperature reconstruction was only possible for the uppermost part of the sediment core, encompassing the post-Holsteinian stadial that is most likely equivalent to MIS 11b. LPAZ KR-12a marks the onset of MIS 11b that directly follows the Holsteinian Interglacial. In this period, average July temperatures still ranged between 17 and 19 °C before rapidly dropping to about 16 °C and increasing again to 18-20 °C in LPAZ KR-12b. July temperatures remained at this level throughout LPAZ KR-12c, before significantly dropping to 15-17 °C in the middle of LPAZ KR-13a. Only at the end of LPAZ KR-13a, which is equivalent to the transition to the following interstadial that most likely corresponds to MIS 11a, July temperatures markedly increased again to about 20 °C.

Table 1: Temperature reconstruction from Chironomidae preserved in the Krępa sediments with reconstructed mean July air temperature ( $T_{jul-Ch}$ ), error of the estimated  $T_{jul-Ch}$ , minimum dissimilarity between the chironomid assemblage in the Krępa sediments training set samples (MinDC), principal component analysis values (PCA) and corresponding LPAZ

| Core depth (cm) | Tjul-Ch | error of est. (T <sub>jul-Ch</sub> ) | MinDC   | PCA        | Number of<br>Chironomidae<br>head capsules | LPAZ   |
|-----------------|---------|--------------------------------------|---------|------------|--------------------------------------------|--------|
| 969             | 20.10   | 1.60                                 | 9.82830 | -1.8144135 | 51                                         | KR-13a |
| 975             | 15.26   | 1.64                                 | 6.08105 | -1.2383560 | 48                                         |        |
| 980             | 16.82   | 1.57                                 | 7.89471 | -1.7518844 | 67                                         | KR-12c |
| 985             | 17.23   | 1.59                                 | 8.37351 | -1.4636110 | 78                                         | KR-12b |
| 990             | 15.93   | 1.70                                 | 7.35685 | -1.9244674 | 52                                         |        |
| 995             | 15.84   | 1.72                                 | 6.77137 | -0.6709448 | 53                                         |        |
| 1000            | 18.77   | 1.52                                 | 8.27430 | 6.5934818  | 42                                         |        |
| 1011            | 18.09   | 1.63                                 | 7.90763 | 0.4039345  | 51                                         | KR-12a |
| 1022            | 19.25   | 1.50                                 | 7.06444 | 0.4114688  | 53                                         |        |
| 1080            | 20.20   | 1.53                                 | 8.02666 | 0.5281182  | 52                                         |        |
| 1102            | 18.55   | 1.52                                 | 8.95789 | 1.3132870  | 48                                         |        |
| 1125            | 17.69   | 1.52                                 | 8.63666 | -0.2629876 | 64                                         |        |
| 1148            | 18.97   | 1.56                                 | 6.86405 | -0.1236256 | 57                                         |        |

According to the SNP training set-based reconstruction, 10 samples with good modern analogues remain below the 5 % percentile threshold (minDC), while 3 samples with average modern analogues have values above the 5 % percentile threshold (6.08105 < minDC > 9.82830). PCA values range between ~ -1.92 and 6.59 (Tab. 1).

# 3.3 Vegetation changes during the Early Saalian Glaciation at Krepa site

Dominance of *Chironomus anthracinus*-type (25 %) and Corynocera ambigua (16 %).

- Initially, 14 Local Pollen Assemblages Zones (LPAZ) covering the end of MIS 12 and MIS 11 period were extracted. Post-holsteinian (MIS 11b) covers LPAZ from 12a to 13a.
- LPAZ KR-12a (1122.5-1187.5 cm) At the beginning of the zone, the development of *Pinus* forests with an admixture of *Picea* (up to 6 %) is observed. Low NAP percentages suggest a very dense vegetation. However, percentages of *Pinus* and other tree species gradually decrease, and open herbaceous communities appear. The end of the zone is associated with a decrease in the percentage of *Pinus* pollen. Low number of Chironomidae head capsules (approximately 15 per sample).
- LPAZ KR-12b (1072.5-1122.5 cm) A further decrease in *Pinus* pollen is observed. At the end of the zone, the landscape was likely already dominated by open communities (NAP up to 40 %) and sparse *Pinus* forests. Dominance of *Corynocera ambigua* (24 %) and high contents of *Chironomus anthracinus*-type. **Disappearance of** *Glyptotendipes pallens*-type and appearance of *Glyptotendipes severini*-type.
- LPAZ KR-12c (1022.5-1072.5 cm) Initially, dense *Betula* forests with *Larix* as an admixture dominated the landscape.

  Subsequently, a rapid development of *Pinus* forests is observed. The end of the zone is associated with a sudden drop in the

percentage of *Pinus* pollen. The number of Chironomidae declines. Dominant species are *Chironomus anthracinus*-type (17 %), Corynocera ambigua and *Glyptotendipes pallens*-type (13 %).

**LPAZ KR-13a** (967.5-1022.5 cm) - Initially, there was a significant opening in the vegetation, and herbaceous plants and shrubs dominated the landscape. In the middle of this zone, there was a temporary return of very sparse *Pinus* and *Betula* forests, followed by another expansion of herbaceous vegetation. The end of the zone is associated with an increase in *Betula* pollen. Significant increase in the number of Chironomidae (on average 45 individuals per sample). Dominant species are Corynocera ambigua (approx. 29 %) and *Chironomus anthracinus*-type (18 %).

Figure 3: Simplified percentage pollen diagram from the Krepa 2015 sediment core on depth scale (cm) with zonation of the diagram.

## 3.4 Pollen-based climate reconstructions from the Krepa site

Pollen-based climate reconstructions from the Krępa sediment core reveal distinct climate variability throughout MIS 11b, reflecting stadial—interstadial transitions (Fig. 4). The two pollen-based methods show broadly similar trends across all zones, with MAT generally producing higher summer temperature values than WA-PLS except in KR-12c. Where chironomid data are available, pollen-based MTWA reconstructions reproduce similar patterns, with differences falling within their respective uncertainty ranges. Among the two pollen-based models, MAT generally corresponds better to the chironomid WA-PLS reconstructions, showing overall closer alignment in reconstructed summer temperatures.

WA-PLS reconstructions were somewhat less robust, especially for precipitation, while the TANN and MTWA estimates still showed moderate predictive ability (Tab. 2). Reconstructed MTWA from both pollen-based methods generally ranged between approximately 15°C and 19°C. The two pollen-based methods show similar trends across all zones, with MAT often producing slightly higher summer temperature values than WA-PLS.

- During LPAZ KR-12a, MAT- and WA-PLS-derived MTWA averaged approximately 16.8°C, close to the chironomid-inferred
- mean of 18.3°C. In LPAZ KR-12b, both pollen methods indicate further warming (~18.7°C MAT, ~16.4°C WA-PLS),
- consistent with the chironomid estimate of 19.4°C, reflecting peak interstadial conditions. In LPAZ KR-12c, MTWA values
- dropped to ~15.1°C (MAT) and ~15.7°C (WA-PLS), indicating cooling during this interval. A moderate rebound is evident in
- LPAZ KR-13a, with MTWA increasing again to ~17.3°C (MAT) and ~15.3°C (WA-PLS), while the mean chironomid MTWA
- is 17.5°C.

- TANN values generally followed the summer temperature trends, beginning with relatively warm conditions in LPAZ KR-
- 12a (~3°C). A slight increase was observed in LPAZ KR-12b (~3.1°C), followed by cooling in LPAZ KR-12c (~1.2°C). In
- LPAZ KR-13a, a modest recovery occurred with TANN rising to around 1.57°C.
- MTCO showed greater variability. Winters in LPAZ KR-12a and KR-12b were comparably cold, with MTCO values around
- 369 −9.6°C and −11.7°C, respectively. LPAZ KR-12c showed slightly less severe winters (~-10.72°C). A more pronounced
- cooling occurred in LPAZ KR-13a, where MTCO reached around -13.2°C.
- PANN reconstructions showed some uncertainty but generally ranged between 500 and 900 mm. LPAZ KR-12a was
- characterized by relatively high precipitation (~640 mm), followed by moderately high values in LPAZ KR-12b (~510 mm).
- A moderate increase occurred in LPAZ KR-12c (~580 mm). In LPAZ KR-13a, PANN remained lower, typically around 520
- mm, suggesting continued reduction in annual precipitation.

Figure 4: Pollen-basedreconstructions of mean temperature of the warmest month (MTWA), mean annual temperature (TANN), mean temperature of the coldest month (MTCO), and annual precipitation sum (PANN) for the Krępa site using MAT and WAPLS. Error bars indicate the standard error of prediction (SEP). The chironomid-based mean July air temperature reconstruction is given for comparison.

# 4. Discussion

# 4.1 Chironomidae analysis as a method of palaeoclimate reconstruction

The analysis of subfossil Chironomidae is part of palaeoecological analysis conducted in geological, geomorphological, and archaeological research. Chironomidae, which are insects belonging to the suborder of Nematocera, are common, and inhabit various types of aquatic environments, from moist soil to lakes. Their development cycle can last from 20 days to several years as they can extend the duration of the larval stage depending on environmental conditions (Butler, 1982). Because of the excellent preservation of their larvae's head capsules in lake and peat bog sediments, the analysis of their subfossil remains offers the possibility to reconstruct environmental and climatic changes in the past. This includes quantitative reconstructions of the average July air temperature and the trophic state of the inhabited water body as well as the type and dynamics of the

lake, the water pH, and microhabitats. Furthermore, training sets are also available to reconstruct the historic water level, salinity or oxygen content of the studied water body (Lotter et al., 1997).

# 4.1.1 Possible difficulties in temperature reconstruction based on Chironomidae analysis during past interglacials

The basic principle of palaeoecological reconstructions is uniformitarianism, implying that processes taking place on Earth in the past were the same as today (Krzeminski and Jarzembowski, 1999). This, for example, allows temperature to be reconstructed based on fossil Chironomidae assemblages by assuming that a given species has the same habitat requirements as thousands or hundreds of thousands years ago. The oldest recorded chironomid remains date back to the Late Triassic, i.e. ~200 1 Ma BP (Krzeminski and Jarzembowski, 1999). Data from the MIS 11 Krepa sediments indicate a large difference in the number and state of preservation of chironomid remains compared to Holocene sites. Usually, at least 50 individuals per sample are required for robust reconstructions of the average July temperature, as smaller numbers of identified head capsules considerably increase the error range of the air temperature reconstruction. It is therefore commonly recommended to combine adjacent samples in case of low head capsule amounts (Heiri and Lotter, 2001). To enable selection of sites that could potentially yield chironomid-based palaeoenvironmental reconstructions, it is critical to analyse the factors that could limit the degree of preservation in chironomid remains, or cause a marked decrease/complete disappearance in the number of individuals. Chironomidae inhabit all moist or aquatic habitats from moist wood to the ocean between the tropics and the Arctic. The high specialisation of individual species is thereby decisive for their common occurrence and their ability to survive under difficult environmental conditions. Among the features that allow specimens to succeed are: a short life cycle (in some cases only 8 days) (Reyes-Maldonado et al., 2021), osmoregulation, which enables survival in high-salinity waters (Kokkinn, 1986), or parthenogenesis, which implies a high efficiency of population reproduction, faster colonisation rate and high fertility (Lencioni, 2004; Nondula et al., 2004; Donato and Paggi, 2008; Orel and Semenchenko, 2019; Lackmann et al., 2020), as well as a short DNA chain (Gusev et al., 2010; Cornette et al., 2015). Some species are able to change food resources depending on the availability in their habitat (Tokeshi, 1995; Davis et al., 2003). Large lakes, such as the one that probably existed at Krepa (1) have a greater variety of habitats, thus being characterised by a larger biodiversity of Chironomidae (Allen et al., 1999; Heino, 2000; Tarr et al., 2005), and (2) are more resilient to extreme droughts and other extreme events. In contrast, small lakes with less diverse, isolated habitats exhibit reduced species diversity and dispersal (Roberts, 2003). Despite the specialisation of chironomids, there are many conditions that limit the number of communities. One of the main factors limiting and determining the life processes of Chironomidae is temperature as each life stage is highly dependent on this factor. The development of eggs, larvae and pupae, nutrition and growth, the emergence of individuals and the ability to fly are all constrained by temperature maxima and minima, beyond which the given processes can no longer take place. Most groups can tolerate low sub-zero temperatures; the temperature below which the development of most species does not occur is -15°C (Walker and Mathewes, 1989; Płóciennik, 2005). At Krepa, however, our July temperature reconstruction indicates temperatures well above that threshold, so even in case of severe winters, Chironomids should have been able to develop family (Danks, 1971). In the case of the Krepa sediments, species of both families were found (e.g. *Propsilocerus lacustris*type and *Procladius* respectively) with Orthocladinae being more abundant than Tanypodinae (57 vs. 5 head capsules) with the highest number of head capsules being preserved during a period with relatively cool summers (15-17°C). Another important factor causing the decline of Chironomidae populations is the lack of oxygen in the water, although this cannot be directly captured by palaeoreconstructions. Instead, low-oxygen conditions are generally only indicated by an abundant occurrence of organic matter in the sediment. Such increases in organic matter commonly increase bacterial respiration and result in oxygen deficiency in the profundal of water bodies (Charlton, 1980; Matzinger et al., 2010; Müller et al., 2012). Another factor limiting the preservation of chironomid head capsules in sediments are mechanical factors that cause damage to the head capsules. For example, Tanypodinae remains are, due to their large size, not very resistant to disintegration and the number of preserved capsules may therefore be smaller (Walker et al., 1984). This would be consistent with our finding of only 5 Tanypodinae individuals in the Krepa sediments across four different depths. The preservation of remains only from the 3rd and 4th larval stages is most likely related to the increased amount of chitin in these developmental stages, making remains of these stages more resistant to disintegration. The remains of Chironomidae may also not be preserved if

during the warmer periods of the year. Frost tolerance is highest in the Orthocladinae family and lowest in the Tanypodinae

accumulation rate is low and remains of species from shore habitats could be poorly preserved. However, studies confirm a 440 positive relationship between biocenosis and thanatocoenosis (Iovino, 1975; Walker et al., 1984). The number of generations

per year may also affect the abundance of Chironomidae, i.e. subfossils of multivoltine species can be overrepresented

compared to bivoltine species, however, it is difficult to determine whether changes in species composition correspond with

voltinism (Tokeshi, 1995).

The main factor influencing the preservation of Chironomidae remains is the content of CaCO3, especially in moderately and strongly acidified lakes. This factor is often more important than pH, depth or time since the deposition of remains (Bailey et al., 2005). The microenvironment and the presence of organic matter are of great importance for the preservation of remains (Briggs and Kear, 1993; Sageman and Hollander, 1999). The faster mineralisation occurs, the better the preservation of any remains (Briggs and Kear, 1993; Park, 1995). Further factors reducing the abundance of chironomids are extreme temperatures, low nutrient levels, acidic waters, high Se concentrations (Del Wayne et al., 2018; Mousavi, 2002), the content of hydrogen

sulphide during holomixis, as well as paludification of the lake (Takagi et al., 2005; Płóciennik et al., 2020).

The lack of oxygen in the sediment could have limited not only the number of Chironomidae but also the number of preserved head capsules in the sediment. In particular, chitin does not usually accumulate in anaerobic sediment, because it is more easily broken down by bacteria, effectively mineralising it into CH4 and CO2 (Wörner and Pester, 2019).

Chironomid species found in the Krepa sediments have a wide range of environmental conditions in which they occur. In particular, we observe dominance of species resilient to harsh conditions, such as the oxygen-deficiency-resistant *Chironomus* anthracinus-type, the eutrophic Chironomus plumosus-type (18.7 and 22.2 % of the total number of head capsules, respectively), as well as the cold-adapted Corynocera ambigua (25.7%) and the freeze-resistant Propsilocerus lacustris-type (7.5%).

Corynocera ambigua is a species often described as cold-adapted oligotrophic (Fjellberg, 1972; Pinder and Reiss, 1983; 460 Walker and Mathewes, 1988; Brooks et al., 2007; Luoto et al., 2008; van Asch et al., 2012), inhabiting shallow lakes in arctic 461 and subarctic regions, though it is also found in eutrophic lakes (Halkiewicz, 2008; Kotrys et al., 2020) Adults of this species 462 are not able to fly, and breed on the water surface when the temperature reaches approximately 7-8 °C (optimum 13.7 °C). 463 Mothes (1968) concluded that Corynocera ambigua larvae develop in autumn and winter, but not during summer. The decline 464 in their numbers may be due to growth of filamentous algae in summer. Larvae of Corynocera ambigua are eurythermic, while 465 the pupae are cold-stenothermic (Brundin, 1949). They only reproduce at low temperatures and inhabit water bodies with a 466 maximum depth of approximately 25 m. The abundance of Corynocera ambigua has been shown to be correlated with 467 charophyte contents (Brodersen and Lindegaard, 1999b). Although this species does not feed on charophytes, their presence 468 may increase the number of diatoms and stabilise the trophic status and water clarity (Forsberg, 1965; Blindow, 1992). 469 Corynocera ambigua live in dendritic tubes, its main food source being diatoms/algal detritus. (Fjellberg, 1972; Boubee, 1983). 470 This species has been recorded during cold episodes or glacial periods, at sites in England (Bedford et al., 2004), Norway 471 (Velle et al., 2005), Poland (Płóciennik et al., 2015), and the Baltic region (Hofmann and Winn, 2000). However, Corynocera 472 ambigua, cannot be considered merely a cold species. Some authors believe that its occurrence depends on high oxygen content 473 in the water (Brodersen and Lindegaard, 1999a) and for other authors, it is a pioneer species that appears first after glacial 474 retreat, similarly to Chironomus anthracinus-type (e.g. Heiri and Millet, 2005; Ilyashuk et al., 2005, 2013, 2022; Gandouin et 475 al., 2016). Luoto and Sarmaja-Korjonen (2011) suggest this is how the species adapts to existing climatic conditions. The 476 locally observed decline in Corynocera ambigua numbers in the Krepa sediments could also be attributed to changes in lake 477 productivity related to changes in the environment. For example, when the production of soil and trees increased, the number 478 of this species has been found to decrease (Magny et al., 2006; Larocque-Tobler et al., 2009).

deficiency in water (Hamburger et al., 1994). It is a species which easily occupies niches that are inaccessible to others.

According to some authors, it is a eutrophic (Kansanen, 1985; Brodersen and Lindegaard, 1999b) or cold-adapted species

(Rohrig et al., 2004; Brooks et al., 2007; Płóciennik et al., 2011) and prefers soft, more organic sediments (McGarrigle, 1980).

Therefore, the appearance of *Chironomus anthracinus*-type and *Glyptotendipes pallens*-type in the Krępa sediment may indicate the onset of eutrophication. Both *Chironomus anthracinus*-type and *Corynocera ambigua* are found in stratified lakes (e.g., Saether, 1979; Heiri, 2004). As we observe in our record, both species are relatively resistant to unfavourable environmental conditions, so possess a wide range of conditions in which they can occur.

Chironomus anthracinus-type occurs in various lake zones and is capable of surviving approximately 2–4 months of oxygen

*Chironomus plumosus*-type, also quite abundant in the sediment sequence, occurs in a wide range of habitats and is particularly resistant to anoxia (Saether, 1979; Brooks et al., 2007). Moreover, along with *Dicrotendipes nervosus*-type, this species is an

indicator of progressive eutrophication (Brodersen and Lindegaard, 1999b)

Both eutrophic and oligotrophic species, as well as warm- and cold-adapted species, occur in the Krepa sediments.

The origin of the sedimentary basin at Krępa is difficult to interpret. Most sites with deposits from the Holsteinian Interglacial in this region of Poland are associated with tunnel valleys that formed during the Elsterian glaciation (Żarski et al., 2005;

Nitychoruk et al., 2006). However, these sites are usually located beyond the maximum extent of the Older Saalian glaciation 494 (Drenthe Stage in Germany: Odra glaciation in Poland: MIS 6) and thus, are subtly visible in the present surface morphology. 495 In the case of Krepa, these deposits have been covered by the Older Saalian glacial advance, resulting in the complete 496 transformation of the post-Elsterian landscape. Based on the geological cross section presented in the DGMP sheet 676 - Kock 497 (Drozd and Trzepla, 2007) and the distribution of interglacial deposits in the study area (Jesionkiewicz, 1982), it can be inferred 498 that the depression hosting the Krepa palaeolake was a relatively extensive kettle hole, formed during the recession of the 499 Elsterian ice sheet. 500 As there are obviously only very few habitats where no invertebrates occur, the absence of chironomid remains during most 501 of the Holsteinian Interglacial could be best explained by sediment-related disintegration and/or anoxic conditions at the 502 bottom of a relatively deep lake. Another reason for the lack of remains could be the mineralisation of chitin. This would be 503 in agreement with the parallel observed lack of cellulose remains from plants as well as with the very low number of 504 Tanypodinae head capsules. However, satisfyingly explaining the lack of chironomid remains in most of the interglacial lake

remains.

505

# 4.1.2 Chironomid-inferred temperature reconstruction from the Krępa site in relation to pollen-based climate reconstructions

deposits requires further research and an in-depth comparison of our results with other lake sediments that lack chitinous

A chironomid-based July temperature reconstruction was only possible for the part of the Krępa sediment core that corresponds to LPAZ KR-12 and early LPAZ KR-13, which most likely corresponds to MIS 11b. Chironomid-based July temperatures during the early part of this interval (LPAZ KR-12a and LPAZ KR-12b) were probably still relatively high and stable, ranging from 19 to 21 °C, but dropping rapidly in LPAZ KR-12c and LPAZ KR-13a to 15-17 °C. The following increase to ~20 °C at the top of LPAZ KR-13a possibly reflects the transition into the post-Holsteinian interstadial that corresponds to MIS 11a. This data indicates the July temperature maximum during the post-Holsteinian is consistent with the temperature range of the SNP training set (3.5-20.0 °C) (Kotrys et al., 2020). Comparing MIS 11 to the Holocene, it is crucial to mention that insolation patterns for both periods differ - MIS 11 was characterised by two insolation maxima, whilst there was only one (though more distinct) during the Holocene (Rohling et al., 2010). In fact, summer temperature increase during MIS 11b might be explained by increasing insolation.

by increasing insolation.

In general, most Chironomidae remains in the Krępa sediments are found during cool periods, but are absent during warm periods. In contrast, Chironomidae were most abundant in LPAZ KR-12, which roughly corresponds to MIS 11b, the first cold phase after the Holsteinian Interglacial (Imbrie et al., 1984; Fawcett et al., 2011). To date, studies using subfossil Chironomidae to reconstruct past climate conditions mainly focused on the Weichselian Late Glacial and the Holocene (Gandouin et al., 2016; Nazarova et al., 2018; Druzhinina et al., 2020). As a result, there are very few chironomid-based July temperature reconstructions for the Late and Middle Pleistocene older than 20 ka BP available (Gandouin et al., 2007; Samartin et al., 2016; Plikk et al., 2019; Ilyashuk et al., 2020; Bolland et al., 2021; Lapellegerie et al., 2024; Rigterink et al., 2024), and no studies

for the MIS 11 complex. In general, chironomid records from other sites and time intervals are characterised by a higher abundance and species diversity of Chironomidae, whilst at Krepa, Chironomidae occur only during the early glacial period following the Holsteinian Interglacial. A similar phenomenon has so far only been observed in the Laptev Sea region (Arctic Siberia), where Chironomidae also appear only in the cold period after the Eemian Interglacial, when the site was surrounded by wet grass-sedge shrub tundra period (Andreev et al., 2004). Assemblages from this site consist mostly of unidentified Tanytarsini individuals, eutrophic Chironomus plumosus and semi-aquatic taxa such as Limnophyes/Paralimnophyes, Smittia and Paraphaenocladius. The three species from the latter group were not identified at Krepa as opposed to Chironomus plumosus and Tanytarsini. Contrary to the patchy occurrence of chironomids, pollen-based climate reconstructions using MAT and WA-PLS provide continuous and robust records that have been successfully applied across various European regions and time periods (Mauri et al., 2015; Chevalier et al., 2020). During LPAZ KR-12a, pollen reconstructions indicate relatively stable and moderate summer temperatures. Additionally, PANN remains relatively high during this phase, suggesting consistently moist conditions supporting dense forest coverage. This is in agreement with the observed dominance of *Pinus* forests with some admixed *Picea* during this phase, reflecting more humid but not necessarily warmer conditions (Caudullo et al., 2016). The significant NAP increase LPAZ KR-12b suggests substantial forest decline, although the pollen-based MTWA reconstructions indicate relatively warm summers. This combination of ecological and climatic signals strongly suggests that the decline in forest cover was primarily driven by colder winter temperatures rather than summer thermal conditions. The pollen-based MTWA reconstruction confirms peak interstadial warmth in terms of summer temperatures are comparable to current mean July temperatures in Eastern Poland (Mauri et al., 2015; Kotrys et al., 2020; Gedminiene et al., 2025). Furthermore, the pollen-based TANN reconstruction also highlights peak interstadial warmth during LPAZ KR-12b, indicating overall favourable climatic conditions during the growing season. The pronounced increase in open-ground vegetation (NAP dominance) and herbaceous taxa thus likely reflects an ecological response to severe winter conditions, that restricted the establishment and survival of forest taxa, particularly those sensitive to extreme winter frosts (Körner and Paulsen, 2004; Harrington and Gould, 2015). LPAZ KR-12c begins with pioneer Betula-Larix forests, reflecting a significant climatic shift towards colder and possibly drier conditions. The appearance of Larix, a cold-tolerant, light-demanding taxon adapted to short growing seasons and low

LPAZ KR-12c begins with pioneer *Betula-Larix* forests, reflecting a significant climatic shift towards colder and possibly drier conditions. The appearance of *Larix*, a cold-tolerant, light-demanding taxon adapted to short growing seasons and low temperatures, reinforces the interpretation of subarctic or boreal-like climate conditions. *Larix* is typically associated with northern coniferous forests, and reaches its distributional limits in areas with low winter temperatures and moderate precipitation (San-Miguel-Ayanz et al., 2022). Gradually increasing pollen signals from *Pinus* indicate a modest rise in thermal conditions later within LPAZ KR-12c, but within a generally cool and moisture-limited climatic regime. The absence of chironomids during this interval corroborates the interpretation of sustained cooler and drier conditions. Chironomid assemblages are sensitive to environmental harshness, and under extremely cold or oligotrophic conditions, their production may be so low that remains are not preserved in sediment records (Eggermont and Heiri, 2012).

The gradual cooling indicated by our chironomid-based reconstruction during LPAZ KR-13a is consistent with the presence

of sparse Betula forests at the onset of this zone. Pollen-based reconstructions suggest that MTWA remained relatively mild

(~17.3 °C MAT, ~15.3 °C WA-PLS), closely aligning with the chironomid-inferred mean T<sub>iul-Ch</sub> value of ~17.5 °C for this interval. Although the chironomid data exhibits a broader range (15–20°C), this variability falls within typical reconstruction uncertainties and does not suggest a fundamentally different climatic signal. Meanwhile, declines in T<sub>ann</sub> and especially MTCO indicate cold-season severity remained the primary constraint on forest development (Nienstaedt, 1967; Körner and Paulsen, 2004; Harrington and Gould, 2015). In parallel, a reduction in P<sub>ann</sub> further supports increasing climatic stress, potentially limiting moisture availability and forest resilience during this transitional phase (Körner and Paulsen, 2004). The broader relevance of the climatic conditions reconstructed from Krepa pollen data is given by the comparison with other MIS 11 palaeotemperature reconstructions from Southern Europe (Fig. 5) (Rodrigues et al., 2011; Oliveira et al., 2016; Kousis et al., 2018; Ardenghi et al., 2019; Sassoon et al., 2023, 2025). These Mediterranean records indicate generally warm conditions during MIS 11b, punctuated by recurrent cooling and drying events. For instance, the Lake Ohrid from SE Europe record shows a transition from temperate deciduous to cold mixed forests, with T<sub>ann</sub> dropping to ~2 °C and mean temperature of the coldest month below -8 °C during the coldest events, despite precipitation often remaining near or above 800-900 mm (Kousis et al., 2018). Meanwhile, records from the SW Mediterranean reflect similar climate oscillations, with Sassoon et al. (2025) documenting synchronous declines in T<sub>ann</sub> and P<sub>ann</sub> centered ~398 ka BP. Krepa record reflects relatively steady summer cooling alongside more marked declines in winter temperatures and moderately decreasing precipitation. Mediterranean vegetation is primarily water-limited, making it especially vulnerable to fluctuations in atmospheric moisture and reductions in winter rainfall (Giorgi, 2006; Lionello et al., 2006). Vegetation in Eastern Europe, however, is highly responsive to winter climate extremes. In particular, cold-season frost events, snow cover variability, and late-winter cold snaps affect plant performance, especially in temperate and continental zones (Kreyling, 2010; Camarero et al., 2022). The lack of accurate absolute dating for terrestrial sediment sequences from the Holsteinian Interglacial makes it difficult to directly compare the results from Krepa to other MIS 11 sites. However, as there are a few quantitative temperature reconstructions based on pollen and biomarkers from other sites in Europe for the post-Holsteinian, a general comparison of temperature levels during this interval is feasible. For example, Tenaghi Philippon record indicates mild summer temperature drop to ~16 °C at the coolest period of MIS 11b (Ardenghi et al., 2019). Climatic fluctuations at another Mediterranean region site – ODP 986 at Alboran Sea – were not abrupt, especially during first half of MIS 11b. Initially summer temperature stayed above 20 °C, only at further stage decreasing to ~17 °C (Fig. 5) (Sassoon et al., 2025). Pollen analyses on marine sediments from the Iberian margin show similar climatic and ecological patterns for MIS 11b as observed at Krepa, namely repeated forest decline events. These were paralleled by reductions in sea surface temperature, although temperatures were still relatively high during most of MIS 11b – only about 1 °C below MIS 11c levels (Rodrigues et al., 2011; Oliveira et al., 2016).

21

A similar pattern between still relatively high air temperature during early MIS 11b, and a temperature drop only during late

MIS 11b is also seen in palynological data from Lake Ohrid in SE Europe (Kousis et al., 2018).

Figure 5: Comparison of (top to bottom) the Marine Isotope Stage (MIS) 11b pollen- and chironomid-based summer temperature reconstructions from Krępa, a summer temperature reconstruction based on branched glycerol dialkyl glycerol tetraethers (brGDGTs) from Tenaghi Philippon, Greece (Ardenghi et al., 2019), a pollen-based summer temperature reconstruction from Lake Ohrid, Balkan Peninsula (Kousis et al., 2018; Kountsodendris et al., 2020), a pollen-based summer temperature reconstruction from ODP Site 976, Alboran Sea (Sassoon et al., 2025), and a biomarker-based (Uk'37) sea surface temperature (SST) reconstruction from marine core MD03-2699, Iberian margin (Rodrigues et al., 2011). The LR04 d<sup>18</sup>O stack (solid black line; Lisiecki and Raymo, 2005) and the 21 June insolation at 50° N (approximate latitude of Krępa; dashed grey line; Laskar et al., 2004) are provided as a palaeoclimatic context. The timing of the MIS boundaries 12/11c and 11a/10 is given according to Lisiecki and Raymo (2005); the timing of the MIS boundaries 11c/11b and 11b/11a is tentative. The insert map shows the locations of the individual proxy records.

In line with our chironomid-based July temperature reconstruction from Krępa, these results show that the temperature decline at the demise of the Holsteinian Interglacial was not abrupt, and at least summer temperatures likely remained at a relatively high level for several thousand years. The general summer temperature variability that is seen in the Krępa record throughout the post-Holsteinian, i.e. the initial drop during the early MIS 11b, the following increase and the more pronounced decrease

during late MIS 11b, as well as the marked increase at transition into MIS 11a, closely resembles vegetation and sea surface temperature variability at the Iberian margin, and may indicate a substantial impact of insolation variability (Rodrigues et al., 2011; Oliveira et al., 2016).

#### Conclusion

- This study presents the first combined chironomid- and pollen-based palaeoclimatic reconstruction for the post-Holsteinian i.e. MIS 11b, offering a new perspective on climate variability in Eastern Europe during this period. The results highlight the complementarity and reliability of both proxies, as pollen-based MAT and WA-PLS reconstructions show strong internal consistency and correspond well with chironomid-inferred summer temperatures where data is available. The summer temperatures range from 15 to 19 °C and between 15 and 20 °C for the pollen- and chironomid-based reconstruction respectively. This indicates colder summers compared to present times for most of the post-Holsteinian period. The pollen-based MAT reconstructions exhibit particularly high predictive skill, especially for temperature variables. The analysed part of the Krępa sediment record reveals a progressive shift towards a more continental climate throughout MIS 11b. This is reflected by gradually cooling summers, increasingly severe winters, and a decline in annual precipitation. These climatic trends coincide with marked vegetation changes, including forest retreat and a rise in herbaceous taxa during colder phases.
- To date, the vast majority of studies addressing terrestrial palaeoclimate variability during the Middle Pleistocene relies on pollen analysis. However, this does not imply a complete lack or low abundance of Chironomid-inferred reconstruction in sites other than Holocene. Moreover, they may prove to be a priceless source of knowledge on temperature, considering potential differences between pollen and Chironomid-inferred records. By comparing the results from different sites, it will be possible to identify the factor(s) that influenced the preservation of Chironomidae subfossil remains.
- Ultimately, this study underscores the value of multi-proxy approaches in palaeoclimate reconstruction, particularly for pre-Holocene periods. Chironomids show significant potential as a summer temperature proxy in older sediments, as long as preservation conditions are favourable.

#### Financial support

The research was funded by the National Science Centre project, "Novel multi-proxy approaches for synchronisation of European palaeoclimate records from the Holstein interglacial". Project no. 2019/34/E/ST10/00275.

#### **Authors contribution**

TP proposed the idea of the main text, and contributed to the figures. TP,AGr and AG wrote the original draft version of the manuscript. BK performed the chironomid-inferred summer temperature reconstruction. AG performed the pollen-inferred climate reconstructions. MŻ collected and described the core in the field. AG, AH, and MS analysed the pollen data. TP, AGr and MS analysed the chironomid data. TP, AGr, AG,, SL, MB and MS did the visualisations (graphs and maps). AG, AH, MŻ,

- MB, JN, MC, SL and MS reviewed the paper. All authors have made substantial contributions to the submission of this
- manuscript.

# 638 Competing interests

The authors declare that they have no conflict of interest.

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

- Supplementary Tools for the Quantitative Analysis of Palaeoecological Data, Geochronometria, 42,
- https://doi.org/10.1515/geochr-2015-0021, 2015.
- Płóciennik, M., Pawłowski, D., Vilizzi, L., and Antczak-Orlewska, O.: From oxbow to mire: Chironomidae and Cladocera as
- habitat palaeoindicators, Hydrobiologia, 847, 3257–3275, https://doi.org/10.1007/s10750-020-04327-6, 2020.
- Pochocka-Szwarc, K., Żarski, M., Hrynowiecka, A., Górecki, A., Pidek, I. A., Szymanek, M., Stachowicz-Rybka, R.,
- Stachowicz, K., and Skoczylas-Śniaz, S.: Mazovian Interglacial sites in the Sosnowica Depression and the Parczew-Kodeń
- Heights (Western Polesie, SE Poland), and their stratigraphic, palaeogeographic and palaeoenvironmental significance, Geol.
- Q., 68, 68: 18-doi: 10.7306/gq.1746, https://doi.org/10.7306/gq.1746, 2024.
- Prokopenko, A. A., Bezrukova, E. V., Khursevich, G. K., Solotchina, E. P., Kuzmin, M. I., and Tarasov, P. E.: Climate in
- continental interior Asia during the longest interglacial of the past 500 000 years: the new MIS 11 records from Lake Baikal,
- SE Siberia, Clim. Past, 6, 31–48, https://doi.org/10.5194/cp-6-31-2010, 2010.
- Ralska-Jasiewiczowa, M., Nalepka, D., and Goslar, T.: Some problems of forest transformation at the transition to the
- oligocratic/ Homo sapiens phase of the Holocene interglacial in northern lowlands of central Europe, Veg. Hist.
- Archaeobotany, 13, 71–71, https://doi.org/10.1007/s00334-003-0027-2, 2004.
- Raymo, M. E. and Mitrovica, J. X.: Collapse of polar ice sheets during the stage 11 interglacial, Nature, 483, 453–456,
- https://doi.org/10.1038/nature10891, 2012.
- Reille, M. and de Beaulieu, J.-L.: Long Pleistocene Pollen Records from the Praclaux Crater, South-Central France, Quat.
- Res., 44, 205–215, https://doi.org/10.1006/gres.1995.1065, 1995.
- Reyes-Maldonado, R., Marie, B., and Ramírez, A.: Rearing methods and life cycle characteristics of Chironomus sp. Florida
- (Chironomidae: Diptera): A rapid-developing species for laboratory studies, PLOS ONE, 16, e0247382,
- https://doi.org/10.1371/journal.pone.0247382, 2021.
- Rigterink, S., Krahn, K. J., Kotrys, B., Urban, B., Heiri, O., Turner, F., Pannes, A., and Schwalb, A.: Summer temperatures
- from the Middle Pleistocene site Schöningen 13 II, northern Germany, determined from subfossil chironomid assemblages,
- Boreas, bor.12658, https://doi.org/10.1111/bor.12658, 2024.
- Roberts, J., Kaczmarek, K., Langer, G., Skinner, L. C., Bijma, J., Bradbury, H., Turchyn, A. V., Lamy, F., and Misra, S.:
- Lithium isotopic composition of benthic foraminifera: A new proxy for paleo-pH reconstruction, Geochim, Cosmochim, Acta,
- 236, 336–350, https://doi.org/10.1016/j.gca.2018.02.038, 2018.

- Roberts, M.: Investigation into the population dynamics and key life history characteristics of non-biting midge (Diptera:
- Chironomidae) which can be utilised to improve nuisance control at Lake Joondalup, Western Australia, Theses Honours,
- 2003.
- Robinson, A., Alvarez-Solas, J., Calov, R., Ganopolski, A., and Montoya, M.: MIS-11 duration key to disappearance of the
- Greenland ice sheet, Nat. Commun., 8, 16008, https://doi.org/10.1038/ncomms16008, 2017.
- Rodrigues, T., Voelker, A. H. L., Grimalt, J. O., Abrantes, F., and Naughton, F.: Iberian Margin sea surface temperature during
- MIS 15 to 9 (580-300 ka): Glacial suborbital variability versus interglacial stability, Paleoceanography, 26,
- https://doi.org/10.1029/2010PA001927, 2011.
- Rohling, E. J., Braun, K., Grant, K., Kucera, M., Roberts, A. P., Siddall, M., and Trommer, G.: Comparison between Holocene
- and Marine Isotope Stage-11 sea-level histories, Earth Planet, Sci. Lett., 291, 97-105,
- https://doi.org/10.1016/j.epsl.2009.12.054, 2010.
- Rohrig, R., Beug, H., Trettin, R., and Morgenstern, P.: Subfossil chironomid assemblages as paleoenvironmental indicators in
- lake faulersee (Germany), Stud. Quat., 117–127, 2004.
- Rylova, T. and Savachenko, I.: Reconstruction of palaeotemperatures of pleistocene interglacial intervals of Belarus from
- palynological evidences, Pol. Geol. Inst. Spec. Pap., Vol. 16, 2005.
- Saether, O. A.: Chironomid communities as water quality indicators, Ecography, 2, 65–74, https://doi.org/10.1111/j.1600-
- 0587.1979.tb00683.x, 1979.
- Sageman, B. B. and Hollander, D. J.: Cross correlation of paleoecological and geochemical proxies: A holistic approach to the
- study of past global change, Spec. Pap. Geol. Soc. Am., 332, 365–384, https://doi.org/10.1130/0-8137-2332-9.365, 1999.
- Samartin, S., Heiri, O., Kaltenrieder, P., Kühl, N., and Tinner, W.: Reconstruction of full glacial environments and summer
- temperatures from Lago della Costa, a refugial site in Northern Italy, Quat. Sci. Rev., 143, 107-119,
- https://doi.org/10.1016/i.quascirev.2016.04.005, 2016.
- San-Miguel-Ayanz, J., de Rigo, D., Caudullo, G., Houston Durrant, T., and Mauri, A.: European atlas of forest tree species.
- Publication Office of the EU, Luxembourg, 2022.
- Sassoon, D., Lebreton, V., Combourieu-Nebout, N., Peyron, O., and Moncel, M.-H.: Palaeoenvironmental changes in the
- southwestern Mediterranean (ODP site 976, Alboran sea) during the MIS 12/11 transition and the MIS 11 interglacial and
- implications for hominin populations, Quat. Sci. Rev., 304, 108010, https://doi.org/10.1016/j.quascirev.2023.108010, 2023.
- Sassoon, D., Combourieu-Nebout, N., Peyron, O., Bertini, A., Toti, F., Lebreton, V., and Moncel, M.-H.: Pollen-based climatic
- reconstructions for the interglacial analogues of MIS 1 (MIS 19, 11, and 5) in the southwestern Mediterranean: insights from
- ODP Site 976, Clim. Past, 21, 489–515, https://doi.org/10.5194/cp-21-489-2025, 2025.
- Schmid, P. e.: Random patch dynamics of larval Chironomidae (Diptera) in the bed sediments of a gravel stream, Freshw.
- Biol., 30, 239–255, https://doi.org/10.1111/j.1365-2427.1993.tb00806.x, 1993.

- Shumilovskikh, L. S., Shumilovskikh, E. S., Schlütz, F., and van Geel, B.: NPP-ID: Non-Pollen Palynomorph Image Database
- as a research and educational platform, Veg. Hist. Archaeobotany, 31, 323–328, https://doi.org/10.1007/s00334-021-00849-8,
- 2022.
- Stockmarr, J.: 1971: Tablets with spores used in absolute pollen analysis, Pollen et Spores 13, 615-621, 1971.
- Stuchlik, L.: Atlas of pollen and spores of the Polish Neogene, W. Szafer Institute of Botany, Polish Acadamy of Sciences,
- 2001.
- Stuchlik, L.: Atlas of Pollen and Spores of the Polish Neogene: Gymnosperms, W. Szafer Institute of Botany, Polish Acadamy
- of Sciences, 2002.
- Stuchlik, L.: Atlas of pollen and spores of the Polish Neogene. 3. Angiosperms (1), W. Szafer Inst. of Botany, Polish Academy
- of Sciences, 2009.
- Szal, M., Kupryjanowicz, M., Wyczółkowski, M., and Tylmann, W.: The Iron Age in the Mragowo Lake District, Masuria,
- NE Poland: the Salet settlement microregion as an example of long-lasting human impact on vegetation, Veg. Hist.
- Archaeobotany, 23, 419–437, https://doi.org/10.1007/s00334-014-0465-z, 2014.
- Szymanek, M.: Elemental geochemistry of freshwater snail shells: palaeolimnology of a Holsteinian (MIS 11) deposit from
- eastern Poland, Boreas, 47, 643–655, https://doi.org/10.1111/bor.12283, 2018.
- Takagi, S., Kikuchi, E., Doi, H., and Shikano, S.: Swimming Behaviour of Chironomus acerbiphilus Larvae in Lake Katanuma,
- Hydrobiologia, 548, 153–165, https://doi.org/10.1007/s10750-005-5196-9, 2005.
- Tarkowska-Kukuryk, M.: Spatial distribution of epiphytic chironomid larvae in a shallow macrophyte-dominated lake: effect
- of macrophyte species and food resources, Limnology, 15, 141–153, https://doi.org/10.1007/s10201-014-0425-4, 2014.
- Tarr, T. L., Baber, M. J., and Babbitt, K. J.: Macroinvertebrate community structure across a wetland hydroperiod gradient in
- southern New Hampshire, USA, Wetl. Ecol. Manag., 13, 321–334, https://doi.org/10.1007/s11273-004-7525-6, 2005.
- Tokeshi, M.: Life cycles and population dynamics, in: The Chironomidae: Biology and ecology of non-biting midges, edited
- by: Armitage, P. D., Cranston, P. S., and Pinder, L. C. V., Springer Netherlands, Dordrecht, 225-268,
- https://doi.org/10.1007/978-94-011-0715-0 10, 1995.
- Tye, G. J., Sherriff, J., Candy, I., Coxon, P., Palmer, A., McClymont, E. L., and Schreve, D. C.: The δ18O stratigraphy of the
- Hoxnian lacustrine sequence at Marks Tey, Essex, UK: implications for the climatic structure of MIS 11 in Britain, J. Quat.
- Sci., 31, 75–92, https://doi.org/10.1002/jqs.2840, 2016.
- Tzedakis, P. C.: The MIS 11 MIS 1 analogy, southern European vegetation, atmospheric methane and the "early
- anthropogenic hypothesis," Clim. Past, 6, 131–144, https://doi.org/10.5194/cp-6-131-2010, 2010.
- Tzedakis, P. C., Hooghiemstra, H., and Pälike, H.: The last 1.35 million years at Tenaghi Philippon: revised chronostratigraphy
- and long-term vegetation trends, Quat. Sci. Rev., 25, 3416–3430, https://doi.org/10.1016/j.quascirev.2006.09.002, 2006.
- Ustrnul, Z., Wypych, A., Marosz, M., Biernacik, D., Czekierda, D., Chodubska, A., Wasielewska, K., Kusek, K., and
- Kopaczka, D.: Climate of Poland 2021, IMGW-PIB, n.d.

- Velle, G., Brooks, S. J., Birks, H. J. B., and Willassen, E.: Chironomids as a tool for inferring Holocene climate: an assessment
- based on six sites in southern Scandinavia, Ouat. Sci. Rev., 24, 1429–1462, https://doi.org/10.1016/j.guascirev.2004.10.010.
- 2005.

- Vera-Polo, P., Sadori, L., Jiménez-Moreno, G., Masi, A., Giaccio, B., Zanchetta, G., Tzedakis, P. C., and Wagner, B.: Climate,
- vegetation, and environmental change during the MIS 12-MIS 11 glacial-interglacial transition inferred from a high-resolution
- pollen record from the Fucino Basin of central Italy, Palaeogeogr. Palaeoclimatol. Palaeoecol., 655, 112486,
- https://doi.org/10.1016/j.palaeo.2024.112486, 2024.
- de Vernal, A. and Hillaire-Marcel, C.: Natural Variability of Greenland Climate, Vegetation, and Ice Volume During the Past
- Million Years, Science, 320, 1622–1625, https://doi.org/10.1126/science.1153929, 2008.
- Walker, I. R. and Mathewes, R. W.: LATE-QUATERNARY FOSSIL CHIRONOMIDAE (DIPTERA) FROM HIPPA LAKE,
- QUEEN CHARLOTTE ISLANDS, BRITISH COLUMBIA, WITH SPECIAL REFERENCE TO CORYNOCERA ZETT.,
- Can. Entomol., 120, 739–751, https://doi.org/10.4039/Ent120739-8, 1988.
- Walker, I. R. and Mathewes, R. W.: Chironomidae (Diptera) remains in surficial lake sediments from the Canadian Cordillera:
- analysis of the fauna across an altitudinal gradient, J. Paleolimnol., 2, 61–80, https://doi.org/10.1007/BF00156985, 1989.
- Walker, I. R., Fernando, C. H., and Paterson, C. G.: The chironomid fauna of four shallow, humic lakes and their representation
- by subfossil assemblages in the surficial sediments, Hydrobiologia, 112, 61–67, https://doi.org/10.1007/BF00007667, 1984.
- Wiederholm, T.: Chironomidae of the holarctic region. Keys and diagnoses. Part 1: larva, Ent Scand Suppl, 19, 1–457, 1983.
- Willerslev, E., Cappellini, E., Boomsma, W., Nielsen, R., Hebsgaard, M. B., Brand, T. B., Hofreiter, M., Bunce, M., Poinar,
- H. N., Dahl-Jensen, D., Johnsen, S., Steffensen, J. P., Bennike, O., Schwenninger, J.-L., Nathan, R., Armitage, S., de Hoog,
- C.-J., Alfimov, V., Christl, M., Beer, J., Muscheler, R., Barker, J., Sharp, M., Penkman, K. E. H., Haile, J., Taberlet, P., Gilbert,
- 1088 M. T. P., Casoli, A., Campani, E., and Collins, M. J.: Ancient Biomolecules from Deep Ice Cores Reveal a Forested Southern
- Greenland, Science, 317, 111–114, https://doi.org/10.1126/science.1141758, 2007.
- Wörner, S. and Pester, M.: The Active Sulfate-Reducing Microbial Community in Littoral Sediment of Oligotrophic Lake
- Constance, Front. Microbiol., 10, https://doi.org/10.3389/fmicb.2019.00247, 2019.
- Yin, Q. Z. and Berger, A.: Individual contribution of insolation and CO2 to the interglacial climates of the past 800,000 years,
- Clim. Dyn., 38, 709–724, https://doi.org/10.1007/s00382-011-1013-5, 2012.
- Żarski, M., Nita, M., and Winter, H.: Nowe stanowiska interglacjalne w rejonie dolin Wilgi i Okrzejki na Wysoczyźnie
- Želechowskiej (Polska południowo-wschodnia), Przeglad Geol., 53, 137–144, 2005.
- Żarski, M., Hrynowiecka, A., Górecki, A., Winter, H., and Pochocka-Szwarc, K.: The maximum extent of the Odranian
- Glaciation (Saalian, MIS 6) in the South Podlasie Lowland (SE Poland) in the light of sites with lacustrine deposits of the
- Mazovian Interglacial, Acta Geol. Pol., e14-e14, https://doi.org/10.24425/agp.2024.150006, 2024.