# Peer review of "Chironomid- and pollen-based quantitative climate reconstructions"

_EGUsphere, 2024_

## Author Response (AR1)

Hereby we thank the Editor and Reviewers for valuable and constructive comments.

Substantial changes were made in the revised manuscript in order to improve our research quality. Below we provide the list of major changes made in the revised manuscript and a point-by-point response to the reviewers' comments.

**Major changes made in the revised version of the manuscript:**

- section "2.1 Data compilation" was deleted due to lack of age-depth models for mapped Polish sites and different time intervals investigated we decided to delete the whole section, integrating former Fig. 1 with the lithological diagram of the Krępa site (formerly Fig. 2) (see our response to Reviewer #1);
- In line with deleting section 2.1, Supplement materials are not attached to the revised version of the manuscript anymore;
- pollen-based temperature reconstruction for the post-Holsteinian (including annual, January and July mean air temperature) is now included in the manuscript (see section 3.5) to improve our research and is compared with chironomid-based temperature reconstruction in section 4.1.2 Chironomid-inferred reconstructions from the Krępa site in relation to pollen-based reconstructions; additional (including suggested) references was added to the discussion;
- section 4.1.1. was substantially restructured we shortened the part about possible causes of Chironomidae head capsules' absence in the sediments and linked the remaining part with our actual observations from Krepa;
- Table 1 was transformed into text (see section 3.2) to improve content layout, presentation and to decrease the volume of the manuscript;
- Former Table 2 (now Table 1) was edited: two columns was added with PCA values and number of Chironomidae head capsules in accordance with Revierwers' suggestions;
- Table 2 (new) was added. It includes cross-validation results for pollen-based MAT and WA-PLS reconstructions (see section 3.5);

- Figure 4 (new) was added. It includes pollen-based reconstructions of mean July air temperature (Tjul), mean annual temperature (Tann), mean January air temperature (Tjan), and annual precipitation sum (Pann) for the Krępa site using MAT and WA-PLS;
- Abstract and conclusion were completely rewritten.

**Response to Reviewer #1 comments:**

The manuscript by Polkowski et al. presents results of chironomid-inferred temperature reconstruction and vegetation changes during the Holsteinian interglacial from a site located in Poland (Krepa). The authors also present a literature review of sites covering the Holsteinian in Poland. The authors discuss in details the possible reasons of poor preservation or absence of chironomid remains in different parts of their record. Because chironomid-inferred temperature reconstructions are rare during this period, the results of the manuscript are valuable for the community. However, I have comments that should be discussed before acceptance of the manuscript for publication in Climate of the Past.

We thank the reviewer for the detailed and constructive comments on our manuscript and provide point-by-point answers to the issues raised.

**Main comments:**

1. Right now, I am a bit sceptical about the chironomid inferred temperature reconstruction. First, the low concentration of chironomid remains makes it hard to reach 50 chironomid per sample. I understand that you merged some adjacent samples to reach higher numbers of chironomids but I think it would be very useful to indicate in Table 2 the number of chironomids in all the samples used for the temperature reconstruction since even after merging some samples still didn't reach 50 head capsules. Also, I don't understand why you decided to keep the sample at 1000 cm (see line 291) since this sample is surrounded by other samples and therefore could be merged. Finally, I don't understand how you get 15 samples (see line 297) after merging since you write that 5 samples had at least 50

chironomids, 7 samples were merged and you kept the sample at 1000 cm alone: this is 13 samples and not 15. And in Figure 4, I only see 14 dots (which I assume are the samples) on the temperature reconstruction curve. This issue should be solved because at the it is confusing for the reader.

In the revised version, we provided a modified Table 2 (now changed to Table 1) with a column containing the number of Chironomid head capsules used for the temperature reconstruction. As far as solitary sample at 1000 cm is concerned - its taxa composition substantially differs from surrounding samples - for example warm-related *Chironomus plumosus* dominates at 1000 cm, whereas at 995 cm cold-related *Corynocera ambigua* dominates, together with mesotrophic *Chironomus anthracinus* at 1011 cm. Therefore, we decided to leave the 1000 cm as a separate sample. The number of samples was given mistakenly as 15 - there were 13 samples - it was corrected in the revised version

2. I find it difficult, at the moment, to understand the relevance of the literature review of polish sites covering the Holsteinian interglacial since you do not use these sites in the discussion of your results. I understand that you made a literature review to show the readers that these sites could also be used in the future in the context of chironomid studies but it would be interesting to compare the results of some of these sites that looked at pollen or diatoms, molluscs or other aquatic indicators with the results (chironomid, pollen) you present for your study site. You could also add, in Supplement Table 2, the proxies analysed for each of the sites.

We admit that the literature review of the Polish sites is not really used in the discussion of our results and therefore doesn't really contribute to the story of our manuscript as mentioned by the reviewer. Therefore, we decided to delete the entire paragraph on the Polish sites (former section 2.1 "Data compilation") in the revised version of our manuscript. However, the related overview map (former Fig. 1) was modified and remained in the revised manuscript. It is now an integral part of a new Fig. 1 and includes also the lithological profile presented previously in Fig. 2 that is part of a revised section 2.1 "Study area" (formerly section 2.3). Comparing these sites might be difficult due to lack of the depth-age models of these sites and their different investigated time intervals. Therefore, we decided to delete the entire section.

3. I would suggest to work on the discussion because, at the moment, most of your interpretations are often not supported by any other publications. I would suggest trying to find publications supporting your interpretations especially in the section "1.2 Summer temperature and ecological reconstructions based on Chironomids from the Krępa site in relation to environmental change" which is very interesting. I would also suggest using more the pollen in the discussion as, if I understood correctly, these results are not published yet. It would be interesting to compare the vegetation changes at Krepa with other know records.

We addressed these issues in the revised manuscript by adding references supporting our interpretations. Comparing vegetation changes with other Central European pollen records is a great idea. However, a separate publication including such comparison is planned. Therefore, we would prefer not to extensively develop this section as the temperature reconstruction remains the main scope of this paper.

4. I don't see the relevance of the section "1.1 Possible difficulties in climate reconstruction based on Chironomidae analysis during past interglacials" in the discussion. I understand that you want to show possible explanations for the low concentrations, or absence chironomids in some parts of your record. However, you don't really make the link with your chironomid assemblages. In this section you mention species/morphotypes that are not present in your chironomid record, so they should probably not be mentioned there. Also, most of the studies you cite in this section worked with specific species so you should not write "-type" after the species names. My suggestion would be to restructure this section to discuss the possible causes of absence of chironomids in some parts of your record, which is very interesting, by linking them with your actual results.

We acknowledge the justified criticism raised by the reviewer regarding our admittedly rather weak attempt to explain the low concentrations / absence of chironomids in our record and regret that we did not provide a proper connection between our actual observations and the fairly theoretical description of ecological preferences of individual taxa provided in former section 4.1.1. Accordingly, we restructured this section, now paying attention to a more proper connection of individual species preferences and our findings.

5. I would suggest to work on the writing as it is sometimes difficult to understand what you want to say. You also sometimes use the wrong words such as "recreate" instead of "reconstruct". Also please pay attention on the writing of Chironomidae, which should not be written in italic, and the morphotype/species spelling. The morphotypes should always be written with "-type", which should not be in italic, and when you are referring to individual species don't add "-type" after the species name.

We addressed all these issues in the revised version of the manuscript.

**Other specific comments**

We appreciate the reviewer's detailed suggestion for improvements listed below and modified the revised manuscript accordingly where appropriate. If changes were not justified in our opinion, we provided an explanation.

Line 1: don't write Chironomidae in italic but in regular font as it is a family name and family names are written in regular font.

This was corrected accordingly.

Line 19: "utilised" → used

This was corrected accordingly.

Line 21: "Chironomidae" should not be written in italic as it is a family name but in the regular form "Chironomidae". Please change it throughout the manuscript.

This was corrected accordingly.

Line 21: "recreate summer thermal conditions"  $\rightarrow$  reconstruct past summer air temperatures or infer past summer air temperatures.

This was corrected accordingly.

Line 22: "Non-biting midges remains indicate trophy and pH of water bodies as well."  $\rightarrow$  Chironomid remains can also indicate changes in the trophic state or pH of water bodies.

Entire abstract was rewritten.

**Line 23: "MIS 11 period" → the MIS 11 period**

Entire abstract was rewritten.

Lines 26-28: "The stratigraphic context for the chironomid-based summer temperature reconstruction is provided by pollen data, together allowing to compare our results in the context of climate development at the end of the Holsteinian Interglacial."  $\rightarrow$  Please reformulate this sentence to make it easier to understand.

Entire abstract was rewritten.

Line 28: "species"  $\rightarrow$  taxa. If you are talking about morphotypes you can not write species as several species can represent each morphotype. See also line 29.

Entire abstract was rewritten.

Line 29: "e.g"  $\rightarrow$  e.g.

Entire abstract was rewritten.

Line 29: "Corynocera ambigua-type"  $\rightarrow$  Corynocera ambigua. This one is not a morphotype but rather a species as indicated in the different training set available.

This was corrected accordingly throughout the manuscript.

Line 29: "Chironomus anthracinus-type"  $\rightarrow$  Chironomus anthracinus-type. The "-type" should not be formatted in italic but in regular font. This is the case for all the morphotypes. Please change this issue throughout the text.

This was corrected accordingly throughout the manuscript.

Line 30: "July temperature" → July air temperature.

This was changed to: "summer air temperature".

Line 30: "15,3 °C"  $\rightarrow$  15.3 °C. In English the decimals should be indicated with a dot and not a comma. Please change this throughout the manuscript. See also line 31.

The manuscript has been inspected again and mistakes of this sort were corrected accordingly.

Line 33: remove "even".

Entire abstract was completely restructured.

Lines 34-37: "The additional element of this research is indicating sites within the Polish borders that were investigated so far - mostly on the basis of pollen analysis, occasionally Cladocera, isotopes, etc. - and might be new objects of studies based on Chironomid-inferred temperature reconstructions."  $\rightarrow$  Please reformulate this sentence to make it easier to understand and precise which time interval was investigated in these sites.

This sentence was deleted in line with the handling of main comment #2 (see above).

Line 37: "Chironomid" → chironomid. Please write chironomid without capital letter and check throughout the manuscript.

This was corrected accordingly.

Line 37: "of challenges of"  $\rightarrow$  on

This sentence was deleted as the entire abstract was completely restructured.

Line 42: "participation" → influence

This was corrected accordingly.

Lines 44-46: "various scientific disciplines from the establishment of the boundary of the unit through the scale of human influence on the functioning of the natural environment in the Holocene throughout all scales starting from micro, through regional to global (Brondizio et al., 2016)."  $\rightarrow$  Please reformulate this sentence.

This sentence was reformulated as follows: "With respect to human impact during the Holocene, the so-called "Anthropocene" is presently widely debated across various scientific disciplines though its exact timing as well as the actual dimension of human influence on the environment are still debated (Brondizio et al., 2016)."

Line 47: "has"  $\rightarrow$  is

This sentence was completely rephrased.

**Line 48: "i.a." $\rightarrow$ i.e. Please check the spelling throughout the manuscript.**

The manuscript has been inspected again and mistakes of this sort was corrected accordingly

Line 49: "etc."  $\rightarrow$  remove

This was corrected accordingly.

Line 50: "climatic conditions change" → climatic condition changes

This sentence was rephrased.

Line 52: "water table depth" → water depth

This was corrected accordingly.

Line 54: "is a reconstruction tool for ocean pH" → can be used to reconstruct pH in the ocean

This was corrected accordingly.

Line 55: "vegetation migration" → remove migration

This was corrected accordingly.

Line 55: "can be used" → and can be used

This was corrected accordingly.

Line 56: "the activities of a human in the past"  $\rightarrow$  past human activities

This was corrected accordingly.

Line 58: "Chironomidae remnants analysis allows the assessment of the water reservoir trophy and pH as well."  $\rightarrow$  The analysis of chironomid remains also allows the assessment of the trophic state or pH of freshwater ecosystems.

The sentence was corrected as follows: "...and head capsules of chironomids can serve as the basis for summer air temperature reconstructions (Eggermont and Heiri, 2012) as well as for assessing the trophic state or pH of freshwater ecosystems (Płóciennik, 2005)."

Lines 61-63: "However, these reconstructions are not capable of giving unequivocal information about exact air temperature changes nor whether these changes and their pace are induced by natural causes or human activity"  $\rightarrow$  Please rephrase this sentence.

The sentence was rephrased as follows:: However, these reconstructions neither provide unequivocal information about air temperature changes nor allow to distinguish between the relative contribution of natural drivers and human impact to these changes.

Line 67: "Northern Europe"  $\rightarrow$  Please be consistent in the spelling of Northern Europe throughout the manuscript. See Line 75 "northern Europe".

This was corrected accordingly.

Line 75: "southern European" → "southern Europe"

This was corrected accordingly.

Line 82: "In this research"  $\rightarrow$  In the present study

This sentence was deleted.

Line 83: "(Eggermont and Heiri, 2012)" → Here I would cite other references as examples of temperature reconstructions based on chironomids. For example: Bolland et al., 2021; Engels et al., 2008; Ilyashuk et al., 2022; Rigterink et al., 2024...

This sentence was deleted.

Line 84: "recreate" → "reconstruct

This sentence was deleted.

Line 90: "Nowiny Żukowskie site"  $\rightarrow$  Here I would specify the location of the site by at least mentioning the country

This was corrected as follows:. "...Nowiny Żukowskie site in eastern Poland (Hrynowiecka and Winter, 2016)."

Lines 93-94: "One of the exceptions is Hoxne site in eastern England (Horne et al., 2023)."  $\rightarrow$  Here I would give more information about this study as it is covering the MIS 11 like your

site. You could, for example, specify that they also did a temperature reconstruction based on chironomid.

This was corrected accordingly. Our proposition: "In Northern Europe, there are even fewer records covering MIS 11 e.g. the record from Hoxne in eastern England (Horne et al., 2023) where temperature reconstructions were performed using chironomids (e.g., Brooks, 2006), ostracods (Horne, 2007) and beetle remains (Atkinson et al., 1986)."

Lines 97-98: "We tested temperature reconstruction using the Swiss-Norwegian-Polish Training Set and presented the first Chironomid-inferred temperature reconstruction from Poland before the Last Glacial Period and even for the post-Holsteinian." → Here we present the first chironomid-inferred July air temperature from Poland for the post-Holsteinian.

This was corrected as follows: "Aiming at improving the knowledge about climate variability at the demise of the Holsteinian Interglacial, we present in the following the first quantitative climate reconstructions for the post-Holsteinian in Central Europe, which are based on chironomid and pollen analyses."

**Line 104: "quaternary"** → **Quaternary**

This was corrected accordingly.

Line 112: "The research covered sites located in Poland. Holsteinian (Mazovian) Interglacial has been included."  $\rightarrow$  The research included sites located in Poland and covering the Holsteinian (Mazovian) interglacial

This sentence was deleted in line with the handling of main comment #2 (see above).

Line 114: "several sites located in western half of the country"  $\rightarrow$  several sites located in the western half of the country

This sentence was deleted in line with the handling of main comment #2 (see above).

**Line 114: "area contained between" → remove "contained"**

This sentence was deleted in line with the handling of main comment #2 (see above).

**Line 115: "The sites' locations were" → The sites' location are**

This sentence was deleted in line with the handling of main comment #2 (see above).

Line 117: "- it"  $\rightarrow$  and therefore

This sentence was deleted in line with the handling of main comment #2 (see above).

Lines 117-118: "location estimation tools"  $\rightarrow$  What are these tools?

This sentence was deleted in line with the handling of main comment #2 (see above).

Line 121: "Supplement Figure 2" → Supplement Figure 1"

Fig. 1 and its caption were modified (please see our response to main comment #2 above).

Line 122: "Glaciation ranges based" → Glaciation ranges are based

Fig. 1 and its caption were modified (please see our response to main comment #2 above).

Line 133: "while modern distribution limits of these taxa are located estimated further to the west" → remove "located"

This was corrected accordingly.

Line 152: "146 m amsl." → 146 m asl

This was corrected accordingly.

Line 160: In this section "2.4" you already interpret the sediment of your site which does not really fit in the section "2. Data and methods" section. You could maybe add a paragraph in section "3. Results and interpretation" for the interpretation of the sediment?

Please see our response to the following comment, which is directly related.

Lines 161-187: In this section it would help to better link the first and second paragraph to better understand your interpretations of the sediment. For example: Because of the presence of laminated sandy silts and sandy-clayey silts the unit 2 is interpreted as a result of glaciolimnic sedimentation in a relatively shallow water body...

As suggested in this and the previous comment, we combined the first and second paragraph for better comprehensibility and also move the combined paragraph from section 2 (previously "Data and methods", now "Study site and methods") to the very beginning of section 3 ("Results and interpretation") in the revised version of our manuscript, then appearing as the first part of the new section 3.1 "Lithological description of the Krępa sediment succession and palaeoenvironmental interpretation". After changes, the paragraph reads as follows:

"The basal part of the 23.8-m-long sediment core that was recovered from the Krepa sediment succession in 2015 (Fig. 1) consists of a 2-m-thick layer of massive, light greyish brown sandy clays with a large number of rock fragments (unit 1), which is interpreted as glacial till. As indicated by its stratigraphic position and its petrographic characteristics (Drozd and Trzepla, 2007), this till was accumulated during the Elsterian glaciation (Sanian 2 glaciation in Poland), which is considered to correspond to MIS 12. Directly above the till, a 0.6-m-thick layer of laminated sandy silts and sandy-clayey silts is found (unit 2). These sediments are interpreted as the result of glaciolimnic sedimentation in a relatively shallow water body between blocks of dead ice during the recession of the Elsterian ice- sheet. The glaciolimnic sediments of unit 2 gradually turn into a carbonate gyttja with small interlayers of carbonatic-minerogenic gyttja (unit 3), which was most likely deposited in the profundal of an already relatively deep lake. Between 1187 and 760 cm core depth, non-carbonatic organic-minerogenic gyttjas with a generally increasing mineral content towards the top are found (unit 4). The limnic sediments of unit 4 are interpreted to reflect the gradual shallowing of the lake due to continuing sediment infill. At the same time, the systematic increase in mineral components in the sediments most probably reflects increased denudation and erosion in the catchment, likely favoured by reduced vegetation cover in response to a change towards colder climate conditions. The gyttja sequence of unit 4 is overlain by a 1.9-m-thick layer of massive clays (unit 5), which probably represent accumulation in a periglacial lake. The following 1.1-m-thick layer of fine- to medium-grained sands (unit 6) as well as the overlying 3.1-m-thick layer of rhythmically laminated sandy silts (unit 7) are interpreted as proglacial sediments (units 6 and 7) of the transgressing Early Saalian (MIS 10) ice sheet. Above this succession, the profile is capped by a 1.5-m-thick layer of sandy morainic till with rock fragments (unit 8) related to the Early Saalian glaciation."

Lines 171-173: "The sediments of unit 2 are interpreted as the result of glaciolimnic sedimentation in a relatively shallow water body between blocks of dead ice during the recession of the Elsterian glacier. The glaciolimnic sediments gradually pass into limnic

sediments (unit 3), which are interpreted to be deposited in the profundal of an already relatively deep lake."  $\rightarrow$  Did you take in consideration in your interpretation of the chironomid results these possible changes in water depth? This could have a strong influence on the chironomid assemblages and could potentially explain why sometimes the concentration of chironomids is very low or even you don't find any chironomids in your samples.

We took the water level into account here. This is one of the hypotheses that we unfortunately can neither confirm nor deny. The problem is the lack of comparative data that we could refer to. Another difficulty is that with three analyses, such as: XRF, pollen and Chironomidae, we do not have certain information about the water level in the reservoir. It is also common in the literature not to write about the lack of individuals. Often in articles, fragments with low numbers or with the lack of remains of some proxy are simply not described. This makes it difficult to refer to the literature.

Line 190: In the section "2.5 Pollen analysis" please indicate the number of pollen samples analysed, the volume of sediment analysed and the batch number and number of Lycopodium spores per tablets that you used. It would also be good to indicate which identification keys/books were used if the pollen data are not already published which I assume is the case since you do not refer to any publications. It would also be good to indicated how the Local Pollen Assemblage Zones were determined. Did you use any statistics (optimal sum of squares partitioning, broken stick model) to divide the pollen record into zones? Also, if you did numerical analyses please indicate which software was used.

The requested information on the methodology (e.g. number of samples, information on Lycopodium spores, determination of local pollen assemblage zones) was added in the revised manuscript. The pollen data are so far not published in a peer-reviewed manuscript but only part of PhD thesis - we added the respective reference.

Line 198: "in a shortened pollen diagram" → in a simplified pollen diagram

This was corrected accordingly.

Line 201: "The Holsteinian (Mazovian) commences" → The Holsteinian (Mazovian) starts

This sentence was deleted in the revised version of the manuscript

Line 205: In the section "2.6 Chironomidae analysis" please indicate the number of samples analysed. How did you measure the volume of your samples? And why are you writing "approximately 1 cm3? As it seems that the chironomid remains in your samples were often damaged I think it would be good to specify how you counted them (halves, presence/absence of mandibles...). As you are dealing with very old chironomid remains, I think it would be valuable to add a plate with pictures of the main chironomid taxa present in your samples. Please indicate what was the KOH concentration used and how long did you leave your samples in heated KOH. Also indicate why you used a 212 µm and if at then end you combined the chironomid remains present in the 212 and 100 µm fractions. Please also indicate which microscope and which magnification was used for the identification of chironomid remains.

The information requested was added in the revised manuscript. As long as the plate with the pictures of chironomids is concerned - according to the Chironomid-inferred reconstruction author, the differences between individuals found at Krępa and those from younger periods aren't significant. The paragraph after proposed changes reads as follows:

"Initially, 79 sediment samples of 1 cm³, taken between 800 and 2160 cm depth at 5-40 cm intervals, were investigated for the presence of Chironomidae head capsules. However, only 30 of them (965-1155 cm depth) simultaneously contained more than 0-2 individuals, creating a sequence that enabled a summer temperature reconstruction. Chemical preparation followed Brooks et al. (2007). The precipitate was initially heated with KOH. The wet sediment was then passed through 212 µm (to remove larger sediment particles) and 100 µm mesh sieves and subsequent residues were treated in an ultrasonic bath for 3 sec. The processed sediment was subsequently examined under a stereomicroscope (Zeiss Axio Lab A1) at 25× magnification. Chironomid head capsules from each sample were picked and mounted in Euparal. In case of damaged head capsules, individuals were counted as one if more than half of a body was preserved. Identification of chironomid head capsules followed Wiederholm (1983), Schmid (1993), Klink and Moller Pillot (2003), Brooks et al. (2007) and Andersen et al. (2013). Ecological preferences of identified taxa are based mainly on Brooks et al. (2007), Brundin (1949), Brodersen and Lindegaard (1999b) and Saether (1979)."

I don't think Brooks et al. (2007) is the best reference to find the ecological preferences of chironomid taxa. I would probably also look in other references such as Saether (1979), Brundin (1949), Brodin (1986), Janececk et al. (2017)...

This was corrected accordingly.

**Line 209: "stereo binocular microscope" → stereomicroscope**

This was corrected accordingly.

**Line 210: "followed by" → followed**

This was corrected accordingly.

Line 214: In the section "2.7 Mean July air temperature reconstruction" please indicate why you chose the Swiss-Norwegian-Polish training set and not other available training sets (Finnish, Russian, Swiss-Norwegian)? I assume it is probably because it contains lakes from Poland but it I think it would be good to specify it. Also did you calculated the nearest modern analogues for each of your fossil samples? And the goodness of fit? If so it would be good to mention it here as well as the software used for that. If not, I would recommend to calculate these diagnostic statistics that you could show in the Supplementary material (see Bolland et al., 2021). In this section it would also probably be good to mention how many samples (after merging) were used for the temperature reconstruction, as well as how the samples were merged.

The requested information was included in the text. The modified wording of the paragraph is:

"In order to reconstruct mean July air temperatures (Tjul-Ch) from the Krępa chironomid assemblage, the Swiss-Norwegian-Polish (SNP) training set (Kotrys et al., 2020) was used as this covers a higher temperature span than other available European training sets (e.g. the Finnish, Russian, Swiss-Norwegian training sets) (Kotrys et al., 2020). The SNP training set includes 357 lakes, 134 taxa, covers a temperature range between 3.5 and 20.1 °C. and uses the weighted averaging-partial least squares transfer function (WA-PLS). The RMSEP for this combined training set is 1.39°C, and the R2 is 0.91 (Kotrys et al., 2020). Detrended Correspondence (MinDC) was also calculated. The temperature reconstruction was carried out using the C2 software (Juggins, 2007).

Chironomidae subfossil larvae were obtained from a total of 30 samples from the lacustrine sediments. Samples that contained fewer than 50 head capsules were merged except for a solitary sample at 1000 cm core depth. For 5 samples the required number of 50 head capsules was obtained and the remaining 24 samples were merged into seven clusters. After merging, sample clusters at 975 cm, 1080 cm, 1120 cm and 1125 cm core depth still did not reach 50 head capsules, but

nonetheless, these samples and the one from  $1000 \, \text{cm}$  core depth were included in the reconstruction because the test of the reconstruction showed acceptable results. The lowest number of head capsules used for the  $T_{jul\text{-Ch}}$  reconstruction was 5 individuals at  $1070 \, \text{cm}$  core depth whereas the highest number was 78 at 985 cm core depth. After merging, the total number of samples used for the  $T_{jul\text{-Ch}}$  reconstruction was 13."

**Line 219: Please indicate the version of the software**

Version of the C2 software used was 1.6.

Line 224: Please also mention the chironomids in the caption of the table. For the column dealing with the chironomids you could write: "Main features in the chironomid record" or replace "significant" with "significance". Please also indicate the unit of the depth column.

The information previously contained in the table was organised as plain text (now section 3.2) to improve presentation of our data and to lower the manuscript volume

Line 237-238: "Assemblages could indicate a deterioration of environmental conditions (*Chironomus anthracinus-type* and *Corynocera ambigua-type*)."  $\rightarrow$  Could you explain your interpretation in more details and link it to other publications?

Paragraph was modified. Proposed wording would be as follows:

"Assemblages could indicate a wide range of environmental conditions (e.g. Chironomus anthracinus-type is a profundal species that is tolerant to a wide thermal spectrum (Brooks et al. 2007; Luoto et al. 2019) and Corynocera ambigua is indicative for colder conditions (Brooks, 2006; Brooks et al., 2007)."

Lines 239-240: "contains mainly cold-adapted and freeze-resistant species like *Corynocera* ambigua-type, Glyptotendipes pallens-type and Glyptotendipes severini-type, which are often associated with algae and diatoms or mine leaves (Tarkowska-Kukuryk, 2014)." → Actually, Glyptotendipes pallens-type and Glyptotendipes severini-type are often associated with relatively warm conditions (Heiri et al., 2011; Nazarova et al., 2015; Luoto, 2009; Kotrys et al., 2020).

The paragraph was reformulated as follows:

"LPAZ KR-12b (1072.5-1122.5 cm) contains mainly cold-adapted species like Corynocera ambigua and freeze-resistant species like Glyptotendipes pallens-type and Glyptotendipes severinitype, which are often associated with algae and diatoms or mine leaves, (Tarkowska-Kukuryk, 2014). LPAZ KR-12c is characterised by species highly resistant to difficult environmental conditions, such as Chironomus anthracinus-type, which is typical for nutrient-rich conditions with wide environmental tolerances (Seather 1979, Self et al. 2011), Corynocera ambigua, which has a broad thermal tolerance (Brodersen & Lindegaard 1999) and Glyptotendipes pallens-type, which can better tolerate harsh winter conditions and lives in different types of substrates (Moller Pilot 2013, Cerba et al. 2022)."

Lines 241-242: "LPAZ KR-12c (1022.5-1072.5 cm) is characterized by species highly resistant to difficult environmental conditions, i.a. *Chironomus anthracinus-type*, *Corynocera ambigua-type* and *Glyptotendipes pallens-type*."  $\rightarrow$  Please provide references to other publications to support your interpretation.

We added the following references:

Brodersen, K. P. and Lindegaard, C.: Mass occurance and sporadic distribution of Corynocera ambigua Zetterstedt (Diptera, Chironomidae) in Danish lakes. Neo- and palaeolimnological records, J. Paleolimnol., 22, 41–52, https://doi.org/10.1023/A:1008032619776, 1999.

Čerba, D., Koh, M., Vlaičević, B., Turković Čakalić, I., Milošević, D., and Stojković Piperac, M.: Diversity of Periphytic Chironomidae on Different Substrate Types in a Floodplain Aquatic Ecosystem, Diversity, 14, 264, https://doi.org/10.3390/d14040264, 2022.

Moller Pillot, H. M.: 2 General Aspects of the Systematics, Biology and Ecology of the Chironomini, in: Chironomidae Larvae, Vol. 2: Chironomini, KNNV Publishing, 8–21, 2013.

Saether, O. A.: Chironomid communities as water quality indicators, Ecography, 2, 65–74, https://doi.org/10.1111/j.1600-0587.1979.tb00683.x, 1979.

Self, A. E., Brooks, S. J., Birks, H. J. B., Nazarova, L., Porinchu, D., Odland, A., Yang, H., and Jones, V. J.: The distribution and abundance of chironomids in high-latitude Eurasian lakes with respect to temperature and continentality: development and application of new chironomid-based climate-inference models in northern Russia, Quat. Sci. Rev., 30, 1122–1141, https://doi.org/10.1016/j.quascirev.2011.01.022, 2011.

Lines 243-246: "During LPAZ KR-13b (877.5-244 967.5 cm) the number of *Chironomidae* gradually increased with indicators of progressive eutrophication (e.g. *Chironomus plumosustype* and *Dicrotendipes nervosus-type* (Iwakuma and Yasuno, 1981)) and cold oligotrophic but post-eutrophic environments ( $Corynocera\ ambigua-type$ )(Brooks et al., 2007) occurring more frequently."  $\rightarrow$  I would suggest to reformulate this sentence as it is hard to understand what you want to say here. Is there an increase of taxa indicator of eutrophication and then, after, an increase of oligotrophic indicators? Or they both increase at the same time?

This fragment of the text was reformulated as follows:

"During LPAZ KR-13b the number of chironomid head capsules gradually increased with indicators of progressive eutrophication (e.g. Chironomus plumosus-type and Dicrotendipes nervosus-type (Iwakuma and Yasuno, 1981)) and cold oligotrophic species (such as Corynocera ambigua) (Brooks et al., 2007) still occurring frequently."

Line 254: "inhabiting shallow Arctic"  $\rightarrow$  inhabiting shallow arctic.

This was corrected accordingly.

Lines 279-282: "Both Chironomus anthracinus-type and Corynocera ambigua-type are species found in stratified lakes (e.g., Saether, 1979; Heiri, 2004). As we can see, both species can be called resistant to unfavorable environmental conditions. They have a fairly wide range of conditions in which they occur today and can even withstand long periods of anaerobic conditions in lake reservoirs."  $\rightarrow$  Please provide a reference to a publication explaining that Corynocera ambigua is tolerant to anaerobic conditions.

Sentences were rephrased as follows:

"The appearance of Chironomus anthracinus-type and Glyptotendipes pallens-type in the Krępa sediment may thus indicate the onset of eutrophication. Both Chironomus anthracinus-type and Corynocera ambigua are found in stratified lakes (e.g., Saether, 1979; Heiri, 2004). As we can see, both species are relatively resistant to unfavourable environmental conditions, thus having a fairly wide range of conditions in which they can occur."

Line 290: "Chironomidae subfossil larvae were obtained from a total of 30 samples from the lacustrine sediments."  $\rightarrow$  Please specify the sedimentary units of the samples.

This paragraph was moved to the "Study site and methods" chapter to the new section "2.6 Chironomid-based mean July air temperature reconstruction". New wording of this sentence is as follows:

"Chironomidae subfossil larvae were obtained from a total of 30 samples from the gyttja sediments (unit 4 on Fig. 1)"

Lines 290-291: "Samples that contained significantly fewer than 50 head capsules were merged except for a solitary sample at 1000 cm core depth."  $\rightarrow$  Please explain why you kept a solitary sample at 100 cm. Because to me it seems that this sample is surrounded by other samples on the diagram of Figure 4 and therefore could have merged with other samples.

Keeping a solitary sample at 1000 cm instead of merging it with the remaining clusters was dictated by the differences with the species composition between this particular solitary sample and samples below. Moreover, the number of head capsules was considered sufficient (even though slightly below 50) to avoid merging in this case.

Lines 294-295: "were included in the reconstruction because the test of the reconstruction showed acceptable results."  $\rightarrow$  Please which test did you perform.

This part was reformulated as it was unclear - no statistical test was performed. It was also moved to section 2.6. At the initial stage of Chironomidae analysis, the performance of the reconstruction was checked - including or excluding the solitary sample from 1000 cm depth. Including this sample seemed to give acceptable results (which was assessed based on the knowledge and experience of reconstruction's author).

This sentence after corrections reads as follows:

"After merging, sample clusters at 975 cm, 1080 cm, 1120 cm and 1125 cm core depth still did not reach 50 head capsules, but nonetheless, these samples and the one from 1000 cm core depth were included in the reconstruction as preliminary results seemed credible in terms of obtained temperature values."

Lines 296-297: "After merging, the total number of samples used for the Tjul reconstruction was 15."  $\rightarrow$  From your explanation just above, I understood that you used 5 samples with sufficient amount of chironomids, 7 merged samples and 1 solitary sample to calculate the

temperature reconstruction. And these are 13 samples, not 15. Please modify the text where it is necessary.

This was corrected accordingly (and moved to section 2.6) - total number of samples after merging is 13.

Line 310: "(MinDC") → How did you calculate the dissimilarity? Please indicate that in the section "2.7 Mean July air temperature reconstruction"

The Chironomidae temperature reconstruction was performed using the Modern Analogue Technique (MAT) (Guiot 1190).

Lines 325-326: "to reconstruct the average July palaeotemperature quantitatively"  $\rightarrow$  to quantitatively reconstruct July air temperature.

This fragment was rephrased as follows:

"Because of the excellent preservation of their larvae's head capsules in lake and peat bog sediments for several hundreds of thousands of years, the analysis of their subfossil remains offers the possibility to reconstruct environmental and climatic changes in the past, quantitative reconstructions of the average July air temperature and the trophic state of the inhabited water body as well as the type and dynamics of the lake, the water pH, and microhabitats. Furthermore, training sets are also available to reconstruct the water level, salinity or oxygen content (Lotter et al., 1997)."

Line 326: "the trophy of the reservoir"  $\rightarrow$  the trophic state of the reservoir.

**See rephrased fragment from the comment above.**

Line 327: "Training sets were also created" → Training sets are also available.

**See rephrased fragment from the comment above.**

Lines 453-545: "These data indicate that summer temperature maximum during the post-Holsteinian period was even slightly higher than indicated in the Polish training set (17-20°C)(Kotrys et al., 2020)."  $\rightarrow$  Please reformulate this sentence as it is unclear to me what you want to say.

This sentence was reformulated as follows: "These data indicate that summer temperature maximum during the post-Holsteinian period is consistent with the temperature range of the Polish training set (3.5-20.0°C)(Kotrys et al., 2020)

Lines 470-472: "Considering the dominance of herbs and dwarf shrubs in the pollen spectrum, the limiting factor for the development of forest communities was more likely connected to low winter temperatures as summer temperatures were still relatively high.  $\rightarrow$  Please develop your interpretation and support it with other publications.

This issue was addressed in the revised manuscript and this section was completely rewritten.

Line 474: "In the following" → Following zone?

This section (4.1.2) was completely rewritten.

Lines 480-481: "Summer temperatures during this period reached only  $15^{\circ}$ C, but the limiting factor for vegetation development still remained the winter temperatures."  $\rightarrow$  Here again I would suggest developing your interpretation and refer to other publications.

This section (4.1.2) was completely rewritten.

Line 482: "being equivalent" → corresponding

This was corrected accordingly.

Lines 483-484: "As the pollen record during stadials is mostly controlled by wind-pollinated overproducers such as Poaceae and the long-distance transport of tree pollen (mostly Pinus)" → Here you need a reference.

This section (4.1.2) was completely rewritten.

**Figures and Tables**

Figure 2: I don't understand what "Clay, Silt, Sand, Gravel" at the bottom of the figure represent. Also, there is no unit for the numbers between the units and the sediment profile. I assume the unit is meters but I think it should be indicated on the figure. "glaciolimnic sedimentation" at the top of the figure  $\rightarrow$  Glaciolimnic sedimentation with a capital "G" to be consistent with the other sediment types.

"Clay, Silt, Sand, Gravel" at the bottom of Fig. 2 refer to the predominant grain size of the individual units. The numbers along the sediment profile indeed refer to the profile depth in centimeters. The figure was corrected according to the suggestions.

Table 1: Depth of KR-4 is overlapping with depth of KR-3. I suggest to add a column specifying the Marine Isotope Stage of each Local Pollen Assemblage Zone. What is the difference between "No Chironomidae" and "No individuals of Chironomidae"? Please specify it, in the caption of the table, if there is a difference. Please check the writing of the depths (the decimals should be indicated with a dot and not a comma in English): see for example "KR-8 1497,5 – 1647.5". For LAPZ KR-12b, I would suggest to change "high contents of Chironomus anthracinus-type" to "relatively high abundances of Chironomus anthracinus-type". Also for the same LAPZ you probably forgot words in the second sentence describing the Chironomidae: "The number of Glyptotendipes pallens-type and Glyptotendipes severinitype." For LAPZ KR-13a, you write that "on average 450 individuals per sample" but in the Figure 4 the maximum sum of chironomid in samples is around 80. Why is that? I would also suggest condensing the table because it is on 9 pages now. You could, for example, reduce the space between each LAPZ and shorten the description of the pollen results.

This was corrected accordingly. Table 1 was transformed into text to improve the content layout and presentation.

Figure 3: I think it would be good to have a horizontal line (or dotted line) on the diagram for each zones so that it is easier for the reader to see the differences between the zones. For the lithology it would probably be good to followed the same code as in Figure 2. Please specify the type of spores shown in the diagram (Fern? Fungal?). Please also write the unit of the different pollen types, which I assume is percentage, and for Pediastrum (number of remains?). If possible, it would be good to specify what is included in "Other thermophilic", "Other AP", "Other NAP".

Horizontal dotted line for each zone was added as well as the unit for pollen types (percentages)(in the title of the Fig. 2). We decided to delete the lithological part of the diagram and leave it only in Fig. 1.

"Other NAP" category includes: Achillea t., Alchemilla, Alnus viridis, Amaranthaceae, Anemone, Anthemis t., Apiaceae undiff., Armeria maritima, Aster t. Asteraceae undiff., Brassicaceae, Bupleurum, Calluna vulgaris, Caltha t., Campanula, Cannabis sativa, Carduus t., Caryophyllaceae

undiff., Centaurea cyanus, C. jacea t., C. montana, Cerastium t., Cichorioideae, Cicuta virosa, Cirsium t., Elymus t., Ephedra, Ephedra distachya t., Ephedra fragilis t., Epilobium t. 38 Ericaceae undiff., Euphorbia, Fabaceae undiff., Filipendula, Galeopsis t., Geum, Helianthemum nummularium t., Heracleum, Lathyrus, Ledum palustre, Liliaceae undiff., Linum austriacum t., Lysimachia nummularia t., Lythrum salicaria t., Mentha t., Papaver rhoeas, Persicaria, Peucedanum palustre, Phyteuma t., Plantago lanceolata, P. major, P. media, Polygonum, Polygonum aviculare t., P. bistorta, P. persicaria, P. viviparum, Potentilla t., Ranunculus acris t., Ribes alpinum, R. spicatum, Rosaceae undiff., Rubiaceae, Rumex undif., R. acetosella, Salix herbacea t., Saxifraga oppositifolia t., Saxifragaceae, Stellaria holostea, S. nemorum, Succisa t., Thalictrum, Urtica, Vaccinium t., Valeriana undiff., V. officinalis t.

Table 2: I would suggest to add the number of chironomids per sample in the table so that the readers know which samples might be problematic because they have "too low" numbers of chironomids.

Table 2 was corrected accordingly.

Figure 4: Why the chironomids from LAPZ-14 are not shown on the diagram? Based on Table 1, the abundance of chironomids is very low in this zone but you still found two Chironomus plumosus-type so I think it would be good to also show them on the diagram. Also, in this figure the y axis unit is in meters whereas it is in centimeters in Figure 3 and in Table 2. Please be consistent in all figures and tables. Add "Chironomid diagram" in the caption of the figure as you also show the abundances of chironomid and not only the temperature reconstruction. Please indicate what the grey bars indicate on the temperature reconstruction curve (I assume they are the errors?). I would suggest, if possible, to have a better quality of the figure because when zooming on it the names become a bit fuzzy. Please specify the units of the x axes (percentages, counts,  $^{\circ}$ C). "sume"  $\rightarrow$  "Sum" or "Total chironomids". Please also mention and explain what "MJAT  $^{\circ}$ C" in the text or in the caption of the figure.

The LPAZ-14 zone was not included in the chironomid diagram due to very low abundance of chironomids and as this zone was not included in the temperature reconstruction. Units (cm) were added to y axis in all figures. Caption was modified according to the suggestions.

Supplement table 2: Why are some references in brackets? See for example "Barkowice Mokre".

Supplementary Table 2 was deleted in the revised manuscript.

**Response to Reviewer #2 comments:**

The manuscript egusphere-2024-3129 presents the first Chironomidae-inferred mean July air temperature reconstruction for the post-Holsteinian (MIS-11b) period. The reconstruction is unique, as few studies use Chironomidae as a palaeoclimatic proxy for periods older than the Eemian Interglacial. The research is valuable, and the authors provide interesting data interpretations, referencing a wide range of relevant literature.

The manuscript is well-structured, although the entire review in Section 4.1.1 partly repeats information provided earlier in the text. In this paper, it is unnecessary to extensively review the ecology of Chironomidae and their subfossil deposition in sediments. This content is out of context and should be removed from the manuscript. It could be published separately as a review paper in another journal, rather than in Climate of the Past. A significant challenge for climate reconstructions based on Chironomidae subfossils from such ancient sediments is the speciation rate and potential changes in the species' environmental preferences represented by morphotypes over such a long timescale. Another issue is the zoogeographical context that influences assemblage composition. Due to successive glacialinterglacial cycles up to the present day, when the SNP TS was conducted, species ranges have changed multiple times, affecting regional faunal composition. Can we assume that the morphotypes of subfossils collected from the Krepa post-Holsteinian sediments represent the same species as in the SNP TS? This issue should be briefly discussed in the context of Section 4.1.1. The authors refer to the "actuality of geological processes," but can we make a similar assumption for the climatic preferences of species after such a long time? This is a central issue for climatic reconstructions from such ancient subfossils. As the authors note, Chironomidae have short life cycles, which suggests a fast rate of phylogenetic processes. I agree that temperature reconstructions from such old sediments are possible and reliable, but they should be treated with caution when compared to those from the Holocene, Weichselian, or Eemian periods. Chironomidae are present in only a short section of the Krepa sediments, whereas pollen is ample throughout the entire core. I wonder how a pollen-inferred temperature reconstruction would compare in this case. Would it confirm the chironomid-inferred July air temperatures or not? I leave this question to the authors for consideration. It could be an interesting addition, though the midge-based reconstruction from the post-Holsteinian is already highly unique and sufficient for a strong paper in Climate of the Past.

We thank the reviewer for the detailed and constructive comments on our manuscript and provide point-by-point answers to the issues raised. With respect to the issues raised here briefly, we will address them as follows:

We restructured section 4.1.1, now paying attention to a more proper connection of individual species preferences and our findings. However, we would rather try to better integrate this part with the remaining part of the discussion than entirely delete it from the revised manuscript.

Responding to the issue considering the influence of speciation rate / changing environmental preferences through time - We believe it can be applied to any multi-proxy analysis in post-Holsteinian sediments. The applicability of temperature reconstruction is determined using various statistical methods, including canonical correspondence analysis (CCA). Relationships between chironomid communities and summer temperature using the cross-correlation coefficient differentiation (DCCA) on square-root transformed data in CANOCO v. 4.5 (ter Braak and S\*milauer 2002).

Also the range of analogy of fossil communities to contemporary communities using Modern Analog Technique (MAT) (Birks et al. 1990). By marking individuals, we can determine that the same morphologically similar individuals still occur in the same area as in the Holocene (Płóciennik et al. 2023).

For example, we can give the rate of speciation of the tribe Tanytarsini. Subfossil individuals are from Palaeogene (Eocene/Oligocene ~ 40–45 Ma), Fenno-Sarmatia: 4 species, 3 genera). Currently there are 187 species of 16 genera recorded in Europe. Speciation within the Tanytarsini is mainly ecological and geographic isolation (Giłka 2011). The second factor confirming the applicability of the analysis is its correlation with the results from other multi-proxy analyses. If the results create a coherent whole of factors and their responses with the results from the analyses of pollen, Cladocera, diatoms, Ostracoda, or macroremains, then they confirm the results and applicability of the method. Of course, we can assume that each organism undergoes speciation, but each of these organisms has a different life cycle length. Therefore, here we can assume that this speciation would occur at a different rate. This is a very interesting aspect. That is why it is so important to use other analyses, especially for such old sediments, but with just a few sites investigated we can see that this can be a big challenge, because the sediment is periodically poor in any remains.

As far as pollen-inferred temperature reconstruction is concerned: we decided to add pollen-based temperature reconstruction for the MIS 11b period to be consistent with chironomid-based temperature reconstruction (see new section 3.5 and substantially modified section 4.1.2).

The authors indicate in Section 2.1 that they reviewed 80 sites with Holsteinian sequences in Poland in Table 1, but the table contains different content. I could not locate this data compilation in the manuscript. The text is already lengthy, so I suggest saving this subject for another paper. Additionally, I recommend reducing the manuscript's length by 20% by deleting Section 4.1.1 and moving Table 1 to the supplementary materials.

We completely removed the entire section 2.1 as the presentation of the other Polish sites is not a part of the revised manuscript anymore. As stated in the general comment before, section 4.1.1 was reduced and integrated with the remaining part of the discussion but not entirely deleted.

The manuscript's English requires substantial revision by a native speaker fluent in ecology. I also suggest the following minor comments:

Lines 1-3: The title should be rephrased to be more compact and maybe focused on temperature more than Chironomidae. The paper does not refer to the "central European perspective".

The title was rephrased as follows:

"Chironomid- and pollen-based quantitative climate reconstructions for the post-Holsteinian (MIS 11b) in Central Europe".

In the whole text, lines 1, 21, 23, 28, 33, 54-58, 58-59, 82, 95, 97, 103, Table 1, 227, 228, 230, 231, 233, 237, 244, 247, 289, 290, 319, 321, 325, 331, 334, 351, 357, 359, 362, 364, 373, 388, 389, 390, 392, 395, 398, 401, 430, 435, 441, 495, 461, 475, 491, 493, 494, 507, 509, 514, 515, 520, 521 and elsewhere – Chironomidae is a Family name and according to taxonomic nomenclature it should be written in the regular font, not italics. The chironomid/chironomids is an informal name like a dog, a fox, or a cat, and should be written starting from the small, not a capital letter.

This was corrected accordingly.

Lines 18, 34, 101, 103, 104, 106, 107, 344, 345, 348, 515, 516, and in many other places please change the personal mood of the sentence to impersonal.

This was corrected accordingly.

Lines 18-19: This sentence is complex and hard to read. Please use shorter sentences all over the text.

We checked the entire text with respect to comprehensiveness and shortened some sentences where possible and necessary.

Line 20: Pollen can be used for quantitative reconstructing of the annual temperature, the temperature of the warmest month, the temperature of the coldest month, as well as precipitation and vegetational season duration.

Entire abstract was rewritten.

Line 21: Please change "recreate" to 'reconstruct'.

This was corrected accordingly.

Lines 22, 326, and elsewhere: Please change "trophy" to "trophic state" in the text.

This was corrected accordingly.

Line 23: ".. of the Holocene" - please add 'and Late Weichselian'.

This was corrected accordingly.

Lines 30, and 31: In English decimals should be written with '.' not ','. Please change here and elsewhere in the text.

This was corrected accordingly.

Line 32: How do you know, if there is no quantitative temperature reconstruction inferred from the pollen?

The pollen data from Krępa site are (at the moment) published only as a part of Artur Górecki's (co-author of this manuscript). However, as stated above, we added pollen-based temperature

reconstruction which is the first quantitative palaeoclimate reconstruction for the post-Holsteinian in Central Europe.

Line 34: I'm not sure the word 'enhancing' well fits in this context, could you use some other?

The word was changed to "improving".

Lines 34-37: This sentence is hard to read. Please rephrase. Also according to the taxonomic codex, Cladocera should be written here and elsewhere with the regular font.

We completely removed this sentence as the presentation of the other Polish sites is not a part of the revised manuscript anymore.

**Lines 37-38: Please rephrase the sentence**

We completely removed this sentence as the presentation of the other Polish sites is not a part of the revised manuscript anymore.

Line 38: Maybe it is better to refer to research than data.

Entire abstract was rewritten.

Line 39: I suggest to discuss them wider in the 'Discussion' chapter.

Entire abstract was rewritten.

Line 41: Please change the words 'numerously' and 'triggered' to some others. They don't fit the context.

This was corrected as follows:

"Earth's history is characterised by repeated climate fluctuations, which had until the present interglacial, the Holocene (marine isotope stage (MIS) 1), only natural causes and were not influenced by humans."

Line 42: Maybe 'gives' would be better than "creates"?

This was corrected as follows:

"This offers the opportunity to compare natural climatic changes in the past with the current ones in order to better assess the anthropogenic impact on the present climate."

**Lines 43-46: This sentence is long and hard to read, please rephrase.**

This sentence was rephrased as follows:

"With respect to human impact during the Holocene, the so-called "Anthropocene" is presently widely debated across various scientific disciplines though its exact timing as well as the actual dimension of human influence on the environment are still debated (Brondizio et al., 2016)."

**Lines 47-49: From 11,500 years (cal?) BP there is only one Marine Isotope Stage - MIS 1.**

The complete sentence was rewritten as follows:

"Holocene environmental archives, such as lake, palaeolake and ocean sediments provide material for comprehensive palaeoecological analyses."

Lines 49-51. Please try to write this sentence in more simple words. Please change "species structure" to 'structure of the communities.

This sentence was corrected as follows:

"The sensitivity of some groups of organisms in these archives to changing hydrological or climatic conditions allows to reconstruct past events that directly affected the abundance or structure of the communities (Battarbee, 2000)."

Line 52: Please change "requirements" to 'preferences' and delete "table", just stay with 'water depth'.

This was corrected accordingly.

Line 54: Foraminifera should start with a capital letter as this is the higher taxa name

This was corrected accordingly.

Lines 54-58: This sentence is too long and hard to read. Please cut it to a few shorter sentences.

This was corrected as follows:

"For example, Foraminifera can be used to reconstruct ocean pH (Foster and Rae, 2016; Roberts et al., 2018), pollen provide information about changes in vegetation (Ralska-Jasiewiczowa et al., 2004; Kupryjanowicz et al., 2018) and can be used to reconstruct past human activity (Chevalier et al., 2020) or past climate conditions (e.g. Rylova and Savachenko, 2005; Hrynowiecka and Winter, 2016). Head capsules of chironomids can serve as the basis for summer air temperature reconstructions (Eggermont and Heiri, 2012) as well as for assessing the trophic state or pH of freshwater ecosystems (Płóciennik, 2005)."

Lines 58-59: The word "remnants" does not fit the context, better use 'subfossils' or 'head capsules' instead.

This was corrected accordingly.

Lines 61-63: This sentence is hard to read, please rephrase. The word "pace" does not fit to the context.

This was corrected as follows:

"However, these reconstructions neither provide unequivocal information about air temperature changes nor allow to distinguish between the relative contribution of natural drivers and human impact to these changes."

Line 66: Once you explained the abbreviation MIS in line 24, later on in the text you can use it without referring to the full name "Marine Isotope Stage". Just write MIS.

This was corrected accordingly.

Line 84: The Word "recreate" does not fit the context.

This sentence was completely rephrased.

Line 88: Maybe 'analysed' would be better than "conducted" in this case.

This sentence was completely rephrased.

Maybe I am wrong but I think there is no sense of connection between sentences in line 94 and lines 95-96.

We restructured the entire paragraph on other MIS 11 sites in Europe.

Line 100: Maybe 'brings the' would be better than "gives" in this case?

This sentence was deleted.

Line 104: Should Quaternary start here from a capital letter?

This was corrected accordingly.

Line 105: Please change "ecological requirements" to 'species ecological preferences'.

The sentence was reformulated as follows:

"In addition, we discuss the potential of chironomid analysis for palaeoecological studies of Quaternary sediments as well as the challenges for chironomid analysis arising from both the evolution and interchanging adaptations to species ecological preferences and the preservation of fossil remains."

Line 107: Please delete the phrase "on the map".

We completely removed the entire section 2.1 as the presentation of the other Polish sites will not be a part of the revised manuscript anymore.

Line 114: Please change "they are focused" to 'they aggregate'.

We completely removed the entire section 2.1 as the presentation of the other Polish sites is not a part of the revised manuscript anymore.

Line 199: What method was used for pollen zonation?

The method used was CONISS cluster analysis function. This paragraph after reformulating reads as follows:

"Local Pollen Assemblage Zones (LPAZ) were established using the CONISS cluster analysis function within riojaPlot and were visually adjusted if necessary."

**Section 2.6: Please give the total number of Chironomidae head capsules.**

The total number of Chironomidae HC in the investigated core was 716.

**Line 211: Please change "Pillot" to 'Moller Pillot' as it is a double surname.**

This was corrected accordingly.

Table 1: Please, move Table 1 to the supplements. Please change in the table and everywhere in the text (i.e. line 479) "dominance" to 'domination', also in scientific papers word "significance" is restricted to statistical significance, please change in Table 1 and elsewhere to 'clear', or 'distinct' or 'substantial', etc. At line 410 maybe to 'lower impact' or 'smaller impact'.

The information contained so far in Table 1 was integrated in the main text as a new section 3.2 "Vegetation changes during the Holsteinian Interglacial and the Early Saalian Glacial at Krępa site"

**Tab. 1 KR-11b: Please give space between "percentages" and "and".**

The information contained so far in Table 1 was integrated in the main text, the suggested changes were made accordingly.

**Tab. 1 KR-12a section Chironomidae - please replace "amounts" with 'number'.**

The information contained so far in Table 1 was integrated in the main text, the suggested changes were made accordingly.

**Tab. 1 KR-12b section Chironomidae: please change "content" to 'share', also next sentence about G. pallens-type and G. severini-type is not complete - What do you mean?**

The information contained so far in Table 1 was integrated in the main text, the suggested changes were made accordingly.

**Tab. 1 KR-13b section Chironomidae - please change the font in "occur" to regular.**

The information contained so far in Table 1 was integrated with the main text, the suggested changes were made accordingly.

Lines 234-235: Delete "remains of" from "remains of head capsules", also change "," to '.' in "1222,5 cm". It should be '1222.5 cm'.

This was corrected accordingly.

Line 237: Please change "amounts" to 'number' 'populations' or 'abundance'.

According to the suggestion, "amounts" was changed to "abundance".

Lines 239-240: G. pallens-type and G. severini-type are warm stenotherms. Please rephrase the sentence.

We corrected this paragraph as follows:

"LPAZ KR-12b (1072.5-1122.5 cm) contains mainly cold-adapted species like Corynocera ambigua and freeze-resistant species like Glyptotendipes pallens-type and Glyptotendipes severini-type, which are often associated with algae and diatoms or mine leaves, (Tarkowska-Kukuryk, 2014). LPAZ KR-12c is characterised by species highly resistant to difficult environmental conditions, such as Chironomus anthracinus-type, which is typical for nutrient-rich conditions with wide environmental tolerances (Seather 1979, Self et al. 2011), Corynocera ambigua, which has a broad thermal tolerance (Brodersen & Lindegaard 1999 and Glyptotendipes pallens-type, which can better tolerate harsh winter conditions and lives in different types of substrates (Moller Pilot 2013, Cerba et al. 2022).

Line 246: Please add space between "(Corynocera ambigua-type)" and "(Brooks et al., 2007)".

This was corrected accordingly.

Line 247: Please change "amount" to 'abundance' or 'number'.

According to the suggestion, "amount" was changed to "number".

Line 247: Please add 'stenotherm' between "warm" and "species".

This was corrected accordingly.

Lines 255-256: Please change "has a growth period" to 'larvae develop'.

This sentence was deleted.

Lines 259, 282, 326, 327, 353, 358, 385, 389, 414: Please change here and elsewhere in the text "reservoir" to 'lake'/'lakes/water body/water bodies' (at line 389 to 'bottom').

This was corrected accordingly.

Line 260: Please change 'number' to 'abundance'. Also, it should be 'has been shown'.

This sentence was deleted.

I think that sentence in lines 273-274 is unnecessary. I suggest to delete it.

As suggested, the sentence was deleted.

Please move the paragraph from lines 290-297 to the section 2.7.

As suggested, the paragraph was moved to the new section "2.6. Chironomid-based mean July air temperature reconstruction."

Fig. 4: Please indicate in the figure caption whether Chironomidae are presented in the percent shares or counted numbers of specimens. Also, if you want to be super-correct give Tanytarsini indet. in the regular font. The figure presents not only the mean July air temperature reconstruction but also a stratigraphic diagram of the Chironomidae assemblages.

The figure caption was corrected as suggested. As we combined Fig.1 and 2 of our initial submission into one figure, the figure numbering will shift and Fig. 4 will be Fig. 3 in the revised manuscript.

Line 320: Please change " and an important element of" to 'conducted in'.

This was changed accordingly.

Line 321: Please change "order" to 'suborder' (!).

This was changed accordingly.

Line 325: Please delete: "the diversity of", and "centuries".

This was corrected accordingly.

Line 340: Please change "amount" to 'number'.

This was changed accordingly.

Lines 346, 348, and elsewhere: Please keep American or British English throughout the

whole text.

In the revised manuscript we used British English.

Sentence at lines 358-360: Meaning unclear, please rephrase. Maybe "attract" is used

inadequately and should be replaced by another word, but then still, the sentence needs to

be rephrased.

This was corrected as follows:

"Large lakes like the one that most probably existed at Krepa (1) have a greater variety of habitats,

thus being characterised by a larger biodiversity of Chironomidae (Allen et al., 1999; Heino, 2000;

Tarr et al., 2005). and (2) are more resilient to extreme droughts and other extreme events. In

contrast, small lakes with less diverse and isolated habitats reveal a reduced species diversity and

dispersal (Roberts, 2003)."

Line 361: What do you mean by the "remote habitats"?

This sentence was corrected. Please see our response to the comment above.

Line 374: Chaoboridae and Ceratopogonidae are the Family names and should be given in

regular font, not italics.

This sentence was deleted as we completely rewritten section 4.1.1.

Line 379: Please delete "the amount of".

This was corrected accordingly.

Line 385: "morphological" - do you mean bathymetry?

This sentence was deleted as we completely rewrote section 4.1.1.

Sentence at lines 392-393: Do you mean living larvae of Chironomidae or rather head capsules (subfossils)? If you mean the subfossils then you can't refer to "behaviour", rather you mean redeposition processes.

The complete sentence was rewritten as follows:

"Another factor limiting the preservation of chironomid head capsules in sediments are mechanical factors that cause damage to the head capsules."

Lines 401-402: Please add 'subfossils of' before "multivoltine" and change "being" to 'can be'.

This was corrected accordingly.

Line 403: Maybe 'parameters' would be more suitable than "properties".

This sentence was deleted as we completely rewrote section 4.1.1.

Lines 405-408: Please cut the sentence to a few shorter ones, also "extinction" seems to be a bit too big word in this context.

This sentence was deleted as we completely rewrote section 4.1.1.

Line 413: Please delete "and the bottom of the reservoir".

This was corrected accordingly.

Line 418: Please add 'waters at' before "cold", also for whom is favourable? And from what is more favourable? The anaerobic environments, peat bogs, and aquatic habitats in deserts and cold regions are usually less favourable for chironomids than i.e. lakes with good oxygen conditions, and neutral pH that is localised in temperate regions.

This sentence was deleted as we completely rewrote section 4.1.1.

Sentence at lines 418-419: Meaning unclear.

This sentence was deleted as we completely rewrote section 4.1.1.

Line 419: I am not sure if the word "properties" is properly used in this sentence.

This sentence was deleted as we completely rewrote section 4.1.1.

Line 426: "sites" - do you mean 'samples'?

This paragraph was moved to section "2.5 Chironomidae analysis" and changed as follows:

"Preliminary tests of sample preparation avoided the use of chemicals and included soaking the samples in water for a long time instead to reduce mechanical stress exerted to the head capsules during sample sieving as much as possible. Nevertheless, intact head capsules could not be extracted from some sediment samples even when using this gentle way of sample preparation, likely because of the already highly compacted sediment."

Paragraphs at the lines 425-429 and 438-444 should be moved to the section 2.6.

The first paragraph was moved to section "2.6. Chironomidae analysis" according to the suggestion. The second paragraph was deleted as we completely rewrote section 4.1.1.

Line 439: "functional" - do you mean 'functioning'?

This sentence was deleted as we completely rewrote section 4.1.1.

Line 441: Please change "macroremians" to 'subfossils'.

This sentence was deleted as we completely rewrote section 4.1.1.

The section 4.1.2 is dedicated mostly to the comparison of Chironomidae-inferred summer temperatures with the interpretation of pollen data. Please change the title of the section focusing more on pollen-based reconstructions.

This was corrected accordingly. We suggest the following:

"Chironomid-inferred reconstructions from the Krępa site in relation to pollen-based reconstructions."

Line 454: Please add space between "...C" and "(Kotrys...".

This was corrected accordingly.

Lines 486-487: Something is missing in this sentence, please rephrase.

We deleted the complete sentence.

Lines 507-508: I recommend comparing the trend in the temperature with trends in Chironomidae assemblages reflected by some ordination analysis - i.e. plotting temperatures against DCA/PCA Ax 1 and Ax 2 values. PCA can be very easily calculated in C2.

PCA values were calculated and added to Table 1 (now in section 3.4).

Line 508: Please delete the sentence "They are an environmental indicator." It is redundant.

This sentence was deleted as suggested.

Line 518: Please change "abundantly" to 'with higher concentration'.

This sentence was deleted.

Line 746: Please change "Pillot" to "Moller Pillot" as it is a double surname. Also, he is cited usually as H. K. M.

This was corrected accordingly.

---

## Editor Decision (ED1)

**1 Chironomid- and pollen-based quantitative climate reconstructions**

**for the post-Holsteinian (MIS 11b) in Central Europe**

- 3 Tomasz Polkowski1, Agnieszka Gruszczyńska1,7, Bartosz Kotrys2, Artur Górecki3, Anna Hrynowiecka4,
- 4 Marcin Żarski5, Mirosław Błaszkiewicz1, Jerzy Nitychoruk6, Monika Czajkowska1, Stefan Lauterbach1,8,
- 5 Michał Słowiński¹
- 1Institute of Geography and Spatial Organization Polish Academy of Sciences, Warsaw, 00-818, Poland
- 2Polish Geological Institute National Research Institute, Szczecin, 71-130, Poland
- 3Institute of Botany, Jagiellonian University, Cracow, 30-387, Poland
- 9 Polish Geological Institute National Research Institute, Gdańsk, 80-328, Poland
- 5Polish Geological Institute National Research Institute, Warsaw, 00-975, Poland
- 6Pope John Paul II State School of Higher Education, Biała Podlaska, 21-500, Poland
- 7Faculty of Physics and Earth System Sciences, Leipzig University, Linnéstraße 5, 04103 Leipzig,
- 13 Germany
- 8GFZ Helmholtz Centre for Geosciences, Section 4.6 Geomorphology, Working Group Terrestrial
- 15 Climate Archives, 14473 Potsdam, Germany

16

31

- 17 Correspondence to: Tomasz Polkowski (tomasz.polkowski@twarda.pan.pl)
- 18 Abstract. Investigating climatic and environmental changes during past interglacials is crucial to improve the understanding 19 of the mechanisms that govern the changes related to current global warming. Among the numerous proxies that can be used 20 to reconstruct past environmental and climatic conditions, pollen allow quantitative reconstructions of annual, warmest month 21 and coldest month air temperatures as well as precipitation sums, while the head capsules of Chironomidae larvae are widely 22 used to infer past summer air temperature as well as in the trophic state or pH of water bodies. Nevertheless, the latter have so 23 far mostly been used for reconstructing Holocene and Late Weichselian summer temperatures while there are to date only four 24 sites in Europe with chironomid-based summer air temperature reconstructions for the Late Pleistocene and no such records 25 for any Middle Pleistocene warm period. In this study we present the first quantitative palaeoclimate reconstruction for the 26 post-Holsteinian (Marine Isotope Stage - (MIS) 11b) in Central Europe that is based on both pollen and fossil chironomid 27 remains preserved in palaeolake sediments recovered at Krępa, southeastern Poland. Besides being used for the palaeoclimatic 28 reconstruction, pollen analyses provide the biostratigraphic framework and a broader perspective of climate development at 29 the end of Holsteinian Interglacial. Fossil Chironomidae assemblages at Krepa consist mainly of oligotrophic and mesotrophic 30 taxa (e.g. Corynocera ambigua, Chironomus anthracinus-type) while eutrophic taxa (e.g. Chironomus plumosus-type) are less

abundant. The chironomid-based summer temperature reconstruction yields July air temperature between 15.3 and 20.1°C

during the early post-Holsteinian. Similar summer air temperature changes during the first stadial after the Holstein Interglacial are also reflected by the pollen data, which, however, show a certain delay compared to the chironomid-based temperature reconstruction. In any case, results from Krępa prove that conducting Chironomidae analysis is feasible for periods as early as the Middle Pleistocene, improving our understanding of the mechanisms that control present-day climatic and environmental changes.

**1 Introduction**

32

33

34

35

36

37

38

39

40

41

42

43

44

45

46

47

48

49

50

51

52

53

54

55

56

5758

59

60

61

62

63

Earth's history is characterised by repeated climate fluctuations, which had until the present interglacial, the Holocene (marine isotope stage (MIS) 1), only natural causes and were not influenced by humans. 
[revised manuscript text omitted]
 first Picea-Alnus and then 127 Carpinus and Abies, as well as by a significant proportion of Taxus, and a frequent occurrence of thermophilic taxa such as 128 Pterocarya, Celtis, Juglans, Ilex, Carya, Parrotia, Buxus, Vitis, Brasenia, Trapa, and Azolla (Janczyk-Kopikowa, 1991). 129 Temperature reconstructions based on the indicator species method suggest for the warmest period, the Carpinus-Abies phase, 130 temperatures of 0-3 °C in January and 21-26 °C in July, which along with high precipitation amounts created a suitable 131 environment for the spread of rare warmth-adapted taxa (Krupiński, 1995; Hrynowiecka and Winter, 2016), Palaeotemperature reconstructions from Dethlingen (Koutsodendris et al., 2012) suggest, however, slightly lower temperatures in Western Europe 132 133 for both January (-2.2  $\pm$  3.1 °C) and July (17.8  $\pm$  2.1 °C).
- The warm character of the Holsteinian Interglacial was also confirmed by oxygen isotope analyses on endogenic lake carbonates (Nitychoruk et al., 2005) and snail shells (Szymanek, 2018). These showed significant changes in climatic conditions throughout the Holsteinian Interglacial, during which, continental and maritime influences intertwined in Central
- occurred under maritime influence, i.e. the vegetation period was significantly longer, temperatures were milder and

Europe. Continental influences resulted in a shortened vegetation period with long winters, while the opposite situation

[revised manuscript text omitted]

**3. Results and interpretation**

208

209

210

211

212

213

214

215

216

217

218

219

220

221

222

223

224

225

226

227

228

229

230

231

232

233

234

235

236

**3.1 Lithological description of the Krepa sediment succession and palaeoenvironmental interpretation**

The basal part of the 23.8-m-long sediment core that was recovered from the Krepa sediment succession in 2015 (Fig. 1) consists of a 2-m-thick layer of massive, light grevish brown sandy clays with a large number of rock fragments (unit 1), which is interpreted as till. As indicated by its stratigraphic position and its petrographic characteristics (Drozd and Trzepla, 2007), this till was accumulated during the Elsterian glaciation (Sanian 2 glaciation in Poland), which is considered to correspond to MIS 12. Directly above the till, a 0.6-m-thick layer of laminated sandy silts and sandy-clayey silts is found (unit 2). These sediments are interpreted as the result of glaciolimnic sedimentation in a relatively shallow water body between blocks of dead ice during the recession of the Elsterian ice-sheet. The glaciolimnic sediments of unit 2 gradually turn into a carbonate gyttja with small interlayers of carbonatic-minerogenic gyttja (unit 3), which was most likely deposited in the profundal of an already relatively deep lake. Between 1187 and 760 cm core depth, non-carbonatic organic-minerogenic gyttias with a generally increasing mineral content towards the top are found (unit 4). The limnic sediments of unit 4 are interpreted to reflect the gradual shallowing of the lake due to continuing sediment infill. At the same time, the systematic increase in mineral components in the sediments most probably reflects increased denudation and erosion in the catchment, likely favoured by reduced vegetation cover in response to a change towards colder climate conditions. The gyttja sequence of unit 4 is overlain by a 1.9-m-thick layer of massive clavs (unit 5), which probably represent accumulation in a periglacial lake. The following 1.1-m-thick layer of fine- to medium-grained sands (unit 6) as well as the overlying 3.1-m-thick layer of rhythmically laminated sandy silts (unit 7) are interpreted as proglacial sediments (units 6 and 7) of the transgressing Early Saalian (MIS 6) ice sheet. Above this succession, the profile is capped by a 1.5-m-thick layer of sandy morainic till with rock fragments (unit 8) related to the Saalian glaciation. The origin of the sedimentary basin at Krepa is difficult to interpret. Most sites with deposits from the Holsteinian Interglacial in this region of Poland are associated with tunnel valleys that formed during the Elsterian glaciation (Zarski et al., 2005; Nitychoruk et al., 2006). However, these sites are usually located beyond the maximum extent of the Older Saalian glaciation (Drenthe Stage in Germany; Odra glaciation in Poland; MIS 6) and thus subtly visible in the present surface morphology. In the case of Krepa, the covering of these deposits by the Older Saalian glacial advance has resulted in the complete obliteration of the post-Elsterian landscape. Based on the geological cross section presented in the DGMP sheet 676 - Kock (Drozd and Trzepla, 2007) and the distribution of interglacial deposits in the study area (Jesionkiewicz, 1982), it can only be inferred that the depression hosting the Krepa palaeolake was a relatively extensive kettle hole, formed during the recession of the Elsterian ice sheet.

- 3.2 Vegetation changes during the Holsteinian Interglacial and the Early Saalian Glacial at Krępa site
- 238 LPAZ KR-1 (2120.0-2180.0 cm) NAP values peak at >40 % (mostly *Poaceae*, but also *Artemisia* and *Betula nana*). Open
- 239 communities are dominant. Tree pollen primarily comprises *Pinus* and *Betula* with both taxa potentially existing locally as
- small trees. Pollen of temperate species is sourced from redeposition. No Chironomidae.
- 241 LPAZ KR-2 (2027.5-2110.0 cm) Initially, a conspicuous dominance of pollen originating from pioneering arboreal species,
- notably *Pinus* (up to 61 %) and *Betula* (up to 38%), coupled with a negligible representation of herbaceous plant pollen,
- signifies the prevalence of dense birch and pine forests. Subsequently, *Picea* (up to 24%) becomes established and *Alnus* (up
- 244 to 35 %) colonises areas with higher soil moisture, probably adjacent to the lake. Rising pollen values of riparian species, e.g.
- 245 Fraxinus (3.,5 %), Ulmus (2.,5 %), and Quercus (4 %), suggest local presence. No Chironomidae.
- 246 LPAZ KR-3 (1957.5-2027.5 cm) At the beginning, the percentage of *Taxus* increases sharply (<40 %), suggesting a key role
- in the formation of forest communities. Continued presence of riparian forests. Corylus, Viburnum, Sambucus nigra and
- 248 thermophilic species such as Pterocarya fraxinifolia, Vitis, Hedera helix, Ligustrum and Buxus sempervirens appeared in the
- forest understorey. Despite favourable climatic conditions, high *Pinus* percentages (>40 %) persisted, suggesting that this
- 250 taxon was still important in the formation of forest communities. No Chironomidae.
- 251 LPAZ KR-4 (1892.5-2027.5 cm) Rapid decline of Taxus forests (

Figure 2: Percentage diagram of selected pollen, spore, and algal taxa form the Krępa 2015 sediment core on depth scale (cm) with zonation of the diagram.

**3.4 Summer air temperature reconstruction based on Chironomidae assemblages from the Krepa site**

 Due to the low number of chironomid head capsules preserved in the Krępa sediments, a chironomid-based summer temperature reconstruction was only possible for the uppermost part of the sediment core, encompassing the post-Holsteinian stadial that is most likely equivalent to MIS 11b (Table 1). In LPAZ KR-12a, which marks the onset of MIS 11b that directly follows the Holsteinian Interglacial, average summer temperatures still ranged between 17 and 19 °C before shortly dropping to about 16 °C and increasing again to 18-20 °C in LPAZ KR-12b. Summer temperatures remained at this level throughout LPAZ KR-12c before significantly dropping to 15-17 °C in the middle of LPAZ KR-13a. Only at the end of LPAZ KR-13a, which is equivalent to the transition to the following interstadial that most likely corresponds to MIS 11a, summer temperatures markedly increased again to about 20 °C.

Table 1: Air temperature reconstruction from Chironomidae preserved in the Krępa sediments with reconstructed mean July air temperature  $(T_{jul-Ch})$ , error of the estimated  $T_{jul-Ch}$ , minimum dissimilarity between the chironomid assemblage in the Krępa sediments training set samples (MinDC), principal component analysis values (PCA) and corresponding LPAZ

| Core depth (cm) | Tjul-Ch | error of est. (T jul-Ch ) | MinDC   | PCA        | Number of
Chironomidae
head capsules | LPAZ   |
|-----------------|---------|--------------------------------------|---------|------------|--------------------------------------------|--------|
| 969             | 20.10   | 1.60                                 | 9.82830 | -1.8144135 | 51                                         | KR-13a |
| 975             | 15.26   | 1.64                                 | 6.08105 | -1.2383560 | 48                                         |        |
| 980             | 16.82   | 1.57                                 | 7.89471 | -1.7518844 | 67                                         | KR-12c |
| 985             | 17.23   | 1.59                                 | 8.37351 | -1.4636110 | 78                                         |        |
| 990             | 15.93   | 1.70                                 | 7.35685 | -1.9244674 | 52                                         | KR-12b |
| 995             | 15.84   | 1.72                                 | 6.77137 | -0.6709448 | 53                                         |        |
| 1000            | 18.77   | 1.52                                 | 8.27430 | 6.5934818  | 42                                         | KR-12a |
| 1011            | 18.09   | 1.63                                 | 7.90763 | 0.4039345  | 51                                         |        |
| 1022            | 19.25   | 1.50                                 | 7.06444 | 0.4114688  | 53                                         |        |
| 1080            | 20.20   | 1.53                                 | 8.02666 | 0.5281182  | 52                                         |        |
| 1102            | 18.55   | 1.52                                 | 8.95789 | 1.3132870  | 48                                         |        |
| 1125            | 17.69   | 1.52                                 | 8.63666 | -0.2629876 | 64                                         |        |
| 1148            | 18.97   | 1.56                                 | 6.86405 | -0.1236256 | 57                                         |        |

According to the SNP training set-based reconstruction, 10 samples with good modern analogues remain below the 5 % percentile threshold (minDC), while 3 samples with average modern analogues have values above the 5 % percentile threshold (6.08105 < minDC > 9.82830). PCA values range between  $\sim -1.92$  and 6.59 (Tab. 1).

Figure 3: Chironomid-inferred mean July temperature reconstruction from Krępa with stratigraphic diagram of the Chironomidae assemblages. Caption: Chironomidae species are presented as counted numbers of specimens. Grey bars in the  $_{Tjul-Ch}$  curve indicate error range.

**3.5 Pollen-based climate reconstructions from the Krepa site**

The pollen-based climate reconstructions from the Krępa sediment core reveal a distinct climate variability throughout MIS 11b, in general following the vegetation-indicated stadial-interstadial transitions (Fig. 4). Conducted cross-validation indicated that MAT reconstructions achieved the highest predictive skill, particularly for the reconstructed temperatures (Table 2). WAPLS reconstructions were somewhat less robust, especially for precipitation, while the Tann and  $T_{jan}$  estimates still showed moderate predictive ability (Tab. 2).

Table. 2 Cross-validation results for pollen-based MAT and WA-PLS reconstructions, showing optimal model parameters, R2, and RMSE for each climate variable.

| Method | Climate Variable | Optimal component/k | R 2 | RMSE    |
|--------|------------------|---------------------|----------------|---------|
| WA-PLS | P ann | 1                   | 0.21           | 328 mm  |
|        | T ann | 2                   | 0.59           | 3.53 °C |
|        | Tjul             | 1                   | 0.46           | 3.42 °C |
|        | $T_{ m jan}$     | 2                   | 0.59           | 4.7 °C  |
| MAT    | Pann             | 2                   | 0.56           | 252 mm  |
|        | Tann             | 2                   | 0.77           | 2.71 °C |
|        | Tjul             | 2                   | 0.69           | 2.63 °C |
|        | Tjan             | 2                   | 0.81           | 3.23 °C |

Reconstructed  $T_{jul}$  from both pollen-based methods generally ranged between approximately 14 °C and 19 °C. In the LPAZ where chironomid data are available, the pollen-based  $T_{jul}$  values show a consistent trend and are within the respective uncertainty ranges broadly in line with the chironomid-inferred temperatures. During LPAZ KR-12a, MAT- and WA-PLS-derived  $T_{jul}$  averaged at ~17.3 and ~15.9 °C, respectively, aligning with the chironomid-inferred mean  $T_{jul-Ch}$  of 18.3 °C. In LPAZ KR-12b, both pollen methods indicate further warming (~18.8 °C MAT, ~17.1 °C WA-PLS), which is in good agreement with the chironomid-based estimate of 19.4 °C, reflecting peak interstadial conditions. In LPAZ KR-12c,  $T_{jul}$  values dropped to ~14.7 °C (MAT) and ~15.0 °C (WA-PLS), indicating cooling during this interval. A moderate rebound is evident

in LPAZ KR-13a, which is reconstructed similarly across pollen- and chironomid-based models, with  $Tj_{ul}$  increasing again to ~17.5 °C (MAT) and ~17.3 °C (WA-PLS), while the mean  $T_{iul-Ch}$  is 17.5 °C.

 $T_{ann}$  values generally follow the summer temperature trends, beginning with relatively warm conditions in LPAZ KR-12a, where  $T_{ann}$  was 1.5°C. A slight increase is observed in LPAZ KR-12b, reaching peak interstadial warmth at approximately 2.0°C. This is followed by a marked cooling in LPAZ KR-12c, where  $T_{ann}$  drops to about -2.0°C. In LPAZ KR-13a, a modest recovery occurs with  $T_{ann}$  values rising to around -0.9°C.

Tjan generally exhibits a greater variability. In LPAZ KR-12a and LPAZ KR-12b, winters were comparably cold, with  $T_{jan}$  values around -11.1 and -10.5 °C, respectively. LPAZ KR-12c shows a slight increase in winter severity with  $T_{jan}$  values of approximately -11.3 °C. A more pronounced decline follows in LPAZ KR-13a, where  $T_{jan}$  reaches around -12.7 °C.

 $P_{ann}$  reconstructions show some uncertainty but annual precipitation sums generally range between 500 and 900 mm. LPAZ KR-12a is characterized by relatively high precipitation ( $\sim$ 690–770 mm), followed by still moderately high values in LPAZ KR-12b ( $\sim$ 670–740 mm). A notable decrease occurs in LPAZ KR-12c, with Pann values around 640 mm, indicating drier conditions. In LPAZ KR-13a,  $P_{ann}$  remains lower, typically between 615 and 655 mm, suggesting a continued reduction in annual precipitation.

Figure 4: Pollen-based reconstructions of mean July air temperature  $(T_{jul})$ , mean annual temperature  $(T_{ann})$ , mean January air temperature  $(T_{jan})$ , and annual precipitation sum  $(P_{ann})$  for the Krępa site using MAT and WA-PLS. Error bars indicate the standard error of prediction (SEP). The chironomid-based  $T_{jul-Ch}$  reconstruction is given for comparison.

**4. Discussion**

**4.1 Chironomidae analysis as a method of palaeoclimate reconstruction**

The analysis of subfossil Chironomidae is part of the palaeoecological analysis conducted in geological, geomorphological, and archaeological research. Chironomidae, which are insects belonging to the suborder of Nematocera, are common and inhabit various types of aquatic environments, from moist soil to lakes. Their development cycle can last from 20 days to several years as they can extend the duration of the larval stage depending on environmental conditions (Butler, 1982). Because of the excellent preservation of their larvae's head capsules in lake and peat bog sediments for several hundreds of thousands of years, the analysis of their subfossil remains offers the possibility to reconstruct environmental and climatic changes in the past, quantitative reconstructions of the average July air temperature and the trophic state of the inhabited water body as well as the type and dynamics of the lake, the water pH, and microhabitats. Furthermore, training sets are also available to reconstruct the water level, salinity or oxygen content (Lotter et al., 1997).

**4.1.1 Possible difficulties in climate reconstruction based on Chironomidae analysis during past interglacials**

The basic principle of palaeoecological reconstructions is geological actuality implying that processes taking place on Earth in the past were the same as today (Krzeminski and Jarzembowski, 1999). This, for example, allows to reconstruct temperature based on fossil Chironomidae *assemblages by assuming* that a given species still has the same habitat requirements as thousands or hundreds of thousands years ago. The oldest recorded chironomid remains date back to the Late Triassic, i.e. ~200 1 Ma BP (Krzeminski and Jarzembowski, 1999). Data from the MIS 11 Krępa sediments indicate a large difference in the number and state of preservation of the chironomid remains compared to Holocene sites. In general, at least 50 individuals per sample are required for robust reconstructions of the average July temperature. As smaller numbers of identified head capsules considerably increase the error range of the air temperature reconstruction, it is commonly recommended to combine adjacent samples in case of low head capsule amounts (Heiri and Lotter, 2001). To enable selection of sites that could potentially yield chironomid-based palaeoenvironmental reconstructions, it is therefore critical to analyse the factors that could limit the degree of preservation or cause the disappearance or decrease in the number of individuals.

[revised manuscript text omitted]

Chironomid species found in the Krępa sediments have a wide range of environmental conditions in which they occur. In particular, we observe dominance of species resilient to harsh conditions such as the oxygen-deficiency-resistant *Chironomus anthracinus*-type and the eutrophic *Chironomus plumosus*-type (18.7 and 22.2 % of the total number of head capsules, respectively), the cold-adapted *Corynocera ambigua* (25.7%) and the freeze-resistant *Propsilocerus lacustris*-type (7.5 %). However, both eutrophic and oligotrophic species as well as warm- and cold-adapted species occur in the Krępa sediments. As there are obviously only very few habitats where no invertebrates occur, the absence of chironomid remains during most of the Holsteinian Interglacial might be best explained by sediment-related disintegration and/or anoxic conditions at the bottom of a fairly deep lake. The lack of Chironomidae remains in the Krępa sediments could also be explained by mineralisation of the chitin. This would be in agreement with the parallel observed lack of cellulose remains from plants as well as with the very low number of Tanypodinae head capsules, which are particularly prone to disintegration. However, satisfyingly explaining the lack of chironomid remains in most of the interglacial lake deposits requires further research as well as a comparison of our results with other lake sediments that lack chitinous remains.

**4.1.2 Chironomid-inferred reconstructions from the Krepa site in relation to pollen-based reconstructions**

A chironomid-based summer temperature reconstruction was only possible for the part of the Krępa sediment core that corresponds to LPAZ KR-12 and early LPAZ KR-13, which most probably correspond to MIS 11b. Chironomid-based summer temperatures during the early part of this interval (LPAZ KR-12a and LPAZ KR-12b), i.e. directly after the Holsteinian Interglacial, were most probably still relatively high and stable, ranging from 19 to 21 °C, but dropping rapidly in LPAZ KR-12c and LPAZ KR-13a to 15-17 °C. The following re-increase to about 20 °C at the top of LPAZ KR-13a possibly reflects the transition into the post-Holsteinian interstadial that corresponds to MIS 11a. These data indicate that the summer temperature maximum during the post-Holsteinian is consistent with the temperature range of the SNP training set (3.5-20.0 °C) (Kotrys et al., 2020). On the other hand, there were periods with colder summers than today (15 °C). Comparing MIS 11 to the Holocene, it is crucial to mention that insolation patterns for both periods differ - MIS 11 was characterised by two insolation maxima, while there was only one (though more distinct) during the Holocene (Rohling et al., 2010). In fact, summer temperature increase during MIS 11b might be explained by increasing insolation at that time.

In general, Chironomidae remains in the Krepa sediments occur mostly during cool periods while they are absent during warm periods. For example, in the interglacial part of the sediment record isolated remains were only found in LPAZ KR-4, which precedes the OHO, and in LPAZ KR-7, which corresponds to the YHO. In contrast, Chironomidae were most abundant in LPAZ KR-12, which is thought to roughly correspond to MIS 11b, the first cold phase after the Holsteinian Interglacial (Imbrie

[revised manuscript text omitted]

during most of MIS 11b, i.e. only about 1 °C below the MIS 11c level (Oliveira et al., 2016). A similar pattern with still

relatively high air temperature during early MIS 11b and a temperature drop only during late MIS 11b is also seen in

palynological data from Lake Ohrid in SE Europe (Kousis et al., 2018). In line with our chironomid-based summer temperature reconstruction from Krepa, these results show that the temperature decline at the demise of the Holsteinian Interglacial was not abrupt and that at least summer temperatures most probably remained at a relatively high level for several thousand years. The general summer temperature variability that is seen in the Krepa record throughout the post-Holsteinian, i.e. the moderate drop during early MIS 11b, the following increase and the more pronounced drop during late MIS 11b as well as the marked re-increase at the transition into MIS 11a, closely resembles vegetation and sea surface temperature variability at the Iberian margin and might indicate a substantial impact of insolation variability (Oliveira et al., 2016).

**Conclusion**

617

618

619

620

621

622

623

624

625

626

627

628

629

630

631

632

633

641

642

643

644

645

646

647

- This study presents the first combined chironomid- and pollen-based palaeoclimatic reconstruction for the post-Holsteinian i.e. MIS 11b. offering a new perspective on climate variability in Eastern Europe during this time interval. The results highlight the complementarity and reliability of both proxy types, as pollen-based MAT and WA-PLS reconstructions show strong internal consistency and correspond well with chironomid-inferred summer temperatures where data are available. The summer temperatures range from 14 to 19 °C and between 15 and 20 °C for the pollen- and chironomid-based reconstruction respectively. This indicates colder summers compared to present times for most of the post-Holsteinian period. Among the models, the pollen-based MAT reconstructions exhibit particularly high predictive skill, especially for temperature variables. The analysed part of Krepa sediment record reveals a progressive shift towards more continental climate conditions throughout MIS 11b. This is reflected by gradually cooling summers, increasingly severe winters, and a decline in annual precipitation. These climatic trends coincide with marked vegetation changes, including forest retreat and a rise in herbaceous taxa during
- 634 635

colder phases.

636 To date, the vast majority of studies addressing palaeoclimate variability in the terrestrial realm during the Middle Pleistocene 637 relies on pollen analysis. Nevertheless, this does not prejudge the lack or low abundance of Chironomid-inferred reconstruction 638 in sites other than Holocene. More than that, they might prove to be a priceless source of knowledge about temperature, 639 considering potential differences between pollen and Chironomid-inferred records. By comparing the results from different

640 sites, it will be possible to find the factor that influenced the preservation of subfossil remains of Chironomidae.

[revised manuscript text omitted]

- 711 Brooks, S. J.: Fossil midges (Diptera: Chironomidae) as palaeoclimatic indicators for the Eurasian region, Quat. Sci. Rev., 25,
- 712 1894–1910, https://doi.org/10.1016/j.guascirev.2005.03.021, 2006.
- 713 Brooks, S. J., Langdon, P. G., and Heiri, O.: The identification and use of Palaearctic Chironomidae larvae in palaeoecology,
- 714 Quat. Res. Assoc. Tech. Guide, i-vi,1, 2007.
- 715 Brundin, L.: Chironomiden und andere bodentiere der südschwedischen urgebirgsseen: Ein beitrag zur kenntnis der
- 5716 bodenfaunistischen charakterzüge schwedischer oligotropher seen, Fiskeristyrelsen, 1949.
- 717 Butler, M. G.: A 7-year life cycle for two Chironomus species in arctic Alaskan tundra ponds (Diptera: Chironomidae), Can.
- 718 J. Zool., 60, 58–70, https://doi.org/10.1139/z82-008, 1982.
- 719 Camarero, J. J., Sánchez-Salguero, R., Sangüesa-Barreda, G., Lechuga, V., Viñegla, B., Seco, J. I., Taïqui, L., Carreira, J. A.,
- and Linares, J. C.: Reply to the letter to editor regarding Camarero et al. (2021): Overgrazing and pollarding threaten Atlas
- 721 cedar conservation under forecasted aridification regardless stakeholders' nature, For. Ecol. Manag., 503, 119779,
- 722 https://doi.org/10.1016/j.foreco.2021.119779, 2022.
- 723 Candy, I., Schreve, D. C., Sherriff, J., and Tye, G. J.: Marine Isotope Stage 11: Palaeoclimates, palaeoenvironments and its
- 724 role as an analogue for the current interglacial, Earth-Sci. Rev., 128, 18–51, https://doi.org/10.1016/j.earscirev.2013.09.006,
- 725 2014.
- 726 Caudullo, G., Tinner, W., and De Rigo, D.: Picea abies in Europe: distribution, habitat, usage and threats, 2016.
- 727 Čerba, D., Koh, M., Vlaičević, B., Turković Čakalić, I., Milošević, D., and Stojković Piperac, M.: Diversity of Periphytic
- 728 Chironomidae on Different Substrate Types in a Floodplain Aquatic Ecosystem, Diversity, 14, 264,
- 729 https://doi.org/10.3390/d14040264, 2022.
- 730 Charlton, M. N.: Hypolimnion Oxygen Consumption in Lakes: Discussion of Productivity and Morphometry Effects, Can. J.
- 731 Fish. Aquat. Sci., 37, 1531–1539, https://doi.org/10.1139/f80-198, 1980.
- 732 Chevalier, M., Davis, B. A. S., Heiri, O., Seppä, H., Chase, B. M., Gajewski, K., Lacourse, T., Telford, R. J., Finsinger, W.,
- Guiot, J., Kühl, N., Maezumi, S. Y., Tipton, J. R., Carter, V. A., Brussel, T., Phelps, L. N., Dawson, A., Zanon, M., Vallé, F.,
- Nolan, C., Mauri, A., de Vernal, A., Izumi, K., Holmström, L., Marsicek, J., Goring, S., Sommer, P. S., Chaput, M., and
- 735 Kupriyanov, D.: Pollen-based climate reconstruction techniques for late Quaternary studies, Earth-Sci. Rev., 210, 103384,
- 736 https://doi.org/10.1016/j.earscirev.2020.103384, 2020.
- 737 Cornette, R., Gusev, O., Nakahara, Y., Shimura, S., Kikawada, T., and Okuda, T.: Chironomid Midges (Diptera,
- 738 Chironomidae) Show Extremely Small Genome Sizes, Zoolog. Sci., 32, 248–254, https://doi.org/10.2108/zs140166, 2015.
- 739 Danks, H. V.: OVERWINTERING OF SOME NORTH TEMPERATE AND ARCTIC CHIRONOMIDAE: II.
- 740 CHIRONOMID BIOLOGY, Can. Entomol., 103, 1875–1910, https://doi.org/10.4039/Ent1031875-12, 1971.
- Davis, S., Golladay, S. W., Vellidis, G., and Pringle, C. M.: Macroinvertebrate Biomonitoring in Intermittent Coastal Plain
- 742 Streams Impacted by Animal Agriculture, J. Environ, Oual., 32, 1036–1043, https://doi.org/10.2134/jeq2003.1036, 2003.
- Del Wayne, R. N., Herrmann, S. J., Sublette, J. E., Melnykov, I. V., Helland, L. K., Romine, J. A., Carsella, J. S., Herrmann-
- Hoesing, L. M., Turner, J. A., and Heuvel, B. D. V.: Occurrence of Chironomid Species (Diptera: Chironomidae) in the High

[revised manuscript text omitted]

---

## Author Response (AR2)

Hereby we thank the Editor for valuable and constructive comments. We provide the response to main comments as well as point-by-point response to detailed comments from the manuscript file.

**Main comments:**

1. The main issue with the manuscript at this stage is the structure, which remains unclear. Some of the newly added sections do not belong in the Results part, as they mix results and interpretation. A restructuring of these sections is necessary, as outlined in my comments.

Suggested changes were applied to improve structure of the manuscript:

- Former section 2.2 concerning Holsteinian Interglacial was integrated with Introduction part and information about post-Holsteinian (MIS 11b) climatic conditions were added;
- Former section 3.1 was splitted into two parts and integrated with 2.1 Study area section (lithological description part) and with 4.1.2 (interpretative part);
- Order of pollen- and chironomid part was reversed, both in section 2 (Materials and methods) and 3 (Results), placing chironomid part first (as 2.2, 2.3, 3.1, 3.2) and the pollen part the second (as 2.4, 2.5, 3.3, 3.4).
- 2. I would also like to see, in the next version, a synthesis figure comparing your record with other MIS 11b sites this is currently missing and would greatly enhance the paper for C. Past.

Respective figure (Fig. 5) was added to the revised manuscript (section 4.1.2) as suggested. It includes comparison of the climatic reconstructions from the following MIS 11b records:

- the Marine Isotope Stage (MIS) 11b pollen- and chironomid-based summer temperature reconstructions from Krępa
- a summer temperature reconstruction based on branched glycerol dialkyl glycerol tetraethers (brGDGTs) from Tenaghi Philippon, Greece (Ardenghi et al., 2019);
- a pollen-based summer temperature reconstruction from Lake Ohrid, Balkan Peninsula (Kousis et al., 2018; Kountsodendris et al., 2020)
- a pollen-based summer temperature reconstruction from ODP Site 976, Alboran Sea (Sassoon et al., 2025)

• a biomarker-based (Uk'37) sea surface temperature (SST) reconstruction from marine core MD03-2699, Iberian margin (Rodrigues et al., 2011)

The results were also discussed in the text.

3. The age model (or the absence of one) also needs to be discussed.

Due to the inability to apply radiocarbon dating (e.g. 14C) and the challenges in developing an age—depth model, the direct dating of Holstein interglacial sediments is highly limited. In this context, palynology plays a key role, as pollen analysis enables biostratigraphic comparison between sites. Vegetation changes that occurred during the Holstein interglacial show a relatively uniform pattern across Central Europe from boreal phases to the development of thermophilus deciduous forests. Thanks to the repeatability of this vegetational succession, it is possible to correlate sediment profiles from different locations and assign them to a common stratigraphic framework. Thus, palynology becomes the primary tool for reconstructing and comparing environmental records from this period, despite the lack of precise absolute dating.

4. Additionally, the number of components and analogues selected in the WAPLS and MAT reconstructions should be checked: using only 1 or 2 components is very low for WAPLS (3 is more typical), and 2 analogues is not acceptable for MAT.

The pollen-based temperature reconstruction was reconducted, with increased number of components (4) for WA-PLS method and analogues (7) for MAT.

5. Finally, I recommend that the manuscript be proofread and corrected by a native English speaker. Although the language has improved significantly, some unclear phrasing still remains and should be polished before publication.

According to the suggestion, the manuscript was proofread and corrected by a native English speaker (British).

**Point-by-point response to detailed comments:**

Line 22: "as well as in the trophic state or pH of water bodies" – you can remove this part.

This fragment was deleted.

Line 33: "...which, however, show a certain delay compared to the chironomid-based temperature 33 reconstruction" - I don't agree with that: not seen in the figures and not discussed in your discussion part.

This fragment was deleted.

Line 49: foraminiferas without S.

This was corrected accordingly.

Line 55: "...human impact on the 55 environment during the last 300 years." - human impact is recorded from the Iron age to now.

This fragment was corrected as follows:

"In general, palaeoecological and palaeoclimatological reconstructions record human impact on the environment from the Iron Age (Dumayne-Peaty, 1998; Szal et al., 2014)."

Line 60: Why the MIS 11 instead of, for example, the Eemian? or MIS 19? Please justify.

This was corrected as follows:

"In this regard, a particularly suitable targets are interglacial periods, e.g. Holsteinian Interglacial (or Mazovian Interglacial in Poland), which is commonly estimated to have lasted from 423 to 395 ka BP, thus corresponding to MIS 11c (Lauer and Weiss, 2018; Lauer et al., 2020; Fernández Arias et al., 2023). Holsteinian Interglacial is considered the analogue of the Holocene in terms of astronomical parameters (eccentricity, precession, insolation), climatic conditions and greenhouse gases levels (Koutsodendris et al., 2010; Yin and Berger, 2012; Kleinen et al., 2016)."

Line 64: Please also add Lapellegerie et al., 2024.

Respective reference was added.

Line 67: And the marine core in Alboran sea (Sassoon et al., 2023, 2025), and Lake fucino in Italy (Vera-Polo et al., 2024).

Respective reference was added.

**Line 75: Check the English**

Manuscript was proofread by a native English speaker (see our response to main comment #5 above). The sentence was corrected as follows:

"The contemporary state of knowledge on MIS 11 has been reviewed by Candy et al. (2014)."

**Line 83: Verify with the Dael's paper.**

This issue was verified and the sentence was changed as follows:

""Although the OHO has been described at multiple sites across northern Europe (Koutsodendris et al., 2012), it has so far been identified in few southern European sites (Kousis et al., 2018; Sassoon et al., 2023, 2025)

Line 90: Here, you have to be more precise as you mainly focus your introduction on the MIS11c. But you don't give the state of the art for the MIS 11b. Why is this time period important? What are your objectives? Your hypothesis?

This part was extended and the necessary justifications were added. The fragment after changes reads as follows:

"Aiming at improving the knowledge about climate variability at the demise of the Holsteinian Interglacial, we present the first quantitative climate reconstructions for the post-Holsteinian in Central Europe, based on chironomid and pollen analyses. The aim of analysing this post-interglacial period is to investigate temperature and vegetation changes and to determine if climate at the time was considerably cooler than today. This choice was also dictated by Chironomidae head capsules' presence in post-Holsteinian section of the core (unlike the Holsteinian part). In addition, we discuss the potential of chironomid analysis for palaeoecological study of Quaternary sediments as well as the challenges for chironomid analysis arising from both the evolution and interchanging adaptations to species ecological preferences and the preservation of fossil remains."

Line 106: Details on the core, the stratigraphy and the age model are lacking. You can here move your part in the results on the core description.

The requested details was moved to section 2.1 as requested. As far as the age model is concerned – please see our response to the main comment #3 where we provide justification of depth-age model lacking.

**Line 111: foto → picture**

This was corrected accordingly.

Line 114: I don't understand this part here: better to move to the introduction part, not here in the mat/meth part.

Please see our response to main comments.

**Line 120: Italic.**

This and other similar mistakes were corrected throughout the manuscript.

**Lines 123-125: What about the 11b which is the core of your paper?**

Requested information was added:

"Holsteinian Interglacial was followed by gradual cooling period (MIS 11b) which resulted in annual temperature decline and forest contractions (Tzedakis et al., 2006; Kousis et al., 2018; Hrynowiecka et al., 2019; Sassoon et al., 2025)."

[...]

"MIS 11b brought the AP percentages decrease in Central Europe (Hrynowiecka et al., 2019). Lake Ohrid pollen record reveals the domination of *Pinus* and plant open communities at the time, with Poaceae and *Artemisia* species included (Kousis et al., 2018). ODP Site 976 pollen-based climate reconstructions shows annual temperature drop to around 10 °C and summer temperature to 20 °C (Sassoon et al., 2025)."

Lines 131-132: You can also compare with the temperature reconstructed at Ohrid (Kousis et al., 2018).

This was corrected accordingly.

**Line 156: More details are needed**

Section 2.3 (now 2.4) was rewritten as pollen-based reconstruction parameters changed and details requested further were added – please see our response to the main comment #4 and comments below.

Lines 157-158: Mean annual precipitation; temperature of the warmest month, because with the pollen we don't necessary reconstruct the temperature of July, it can be those of August; MTWA; mean temperature of the coldest month; MTCO.

These were corrected accordingly.

Line 159: Replace by: Guiot, J.: Methodology of the last climatic reconstruction in France from pollen data, Palaeogeogr. Palaeoecol., 80, 49–69, 1990

This was corrected accordingly.

Line 165: Standard errors of prediction (SEP) – how do you calculate them?

Following information was added to the text:

"In the WAPLS approach, sample-specific SEP were obtained via a bootstrapping implemented in the rioja package (Juggins, 2022). For the MAT model we used the cross-validated RMSE as a uniform error estimate for the fossil MAT reconstructions."

Line 167: How many modern pollen records? How many taxa? How the climate parameters are calculated?

The following fragment was added to the manuscript (section 2.4. Pollen analysis) to address the questions above:

"This geographic filtering yielded a regional calibration set of 4955 modern pollen samples, out of the original global dataset. From the fossil pollen dataset, only taxa present in at least 50% of the samples and reaching at least 1% pollen value at least once were included. Additionally we ensured taxonomic consistency between the modern and fossil pollen data by harmonizing taxa names and then removing taxa with zero abundance in the filtered modern set. After this filtering, 10 pollen taxa remained in common between the modern calibration set and the fossil record (primarily major arboreal and herb taxa such as Larix, Betula, Pinus, Salix, Picea, Juniperus, Artemisia, Asteraceae, Poaceae, and Amaranthaceae). Using only these common taxa helps avoid noise from spurious taxa and improves model robustness. All data processing and modeling were carried out in R (RStudio), making use of the analogue and rioja packages for calibration and reconstruction (Simpson, 2007; Juggins, 2022; Simpson and Oksanen, 2025)."

**Line 170: How many samples?**

The pollen-based reconstructions were restricted to the interval of the succession where chironomid remains were also present and were performed on 44 samples.

**Line 175: Please remove this part.**

This part was removed as requested.

**Lines 198-203: Move this part at the end of 2.5.**

This was corrected accordingly (now section 2.2 Chironomidae analysis).

**Line 205: 14 in the figure.**

This was corrected accordingly.

Line 209: I'm having a structural issue with this new section, which I don't think should be included in the Results. I suggest splitting it into two parts: the entire lithological description should go in the Materials and Methods section, where such information is currently lacking; and the interpretative part (lines 228–236) should be moved to the Discussion.

As suggested, lithological description was moved to section 2.1 Study area, and interpretative part to section 4.1.2.

**Line 237: Move this part after the Chironomid part.**

This was corrected accordingly – please see also our response to the main comment #1.

Line 238: Lack a few words to introduce this part. How many PAZ? obtained with CONISS? As the goal of your paper is on MIS 11b, please remove the parts on MIS 11c (in your text and in your pollen diagram) as you don't discuss it after.

Initially, 14 Local Pollen Assemblages Zones (LPAZ) covering the end of MIS 12 and MIS 11 period were extracted (using CONISS and were adjusted visually). Post-holsteinian (MIS 11b) covers LPAZ from 12a to 13a.

LPAZs covering MIS 11c part were removed from the text as well as from the pollen diagram.

**Line 240: "Pollen of temperate species is sourced from redeposition" – why?**

This sentence was deleted in the revised version of the manuscript.

**Comments in lines 256-293**

The whole section was removed from the manuscript as it concerned MIS 11c period.

**Line 312: Italic**

This was corrected accordingly.

**Line 334: Move before the pollen part**

This was corrected accordingly.

Lines 336-364: Move this part to the discussion part: its very interesting but its not your results, it's a discussion!

As suggested, this part was integrated with section 4.1.2.

Lines 365-367: I disagree with this argument: these are two independent proxies, and PAZs cannot be used to describe chironomid variations. The description should be based on depth if defining zones is not feasible. The following paragraph needs to be fully revised and synthesized. No interpretation should be included at this stage in the Results section.

In the revised version of the manuscript, chironomid assemblages variations were described using depths instead of LPAZs. This fragment itself was deleted. The following paragraph was completely rewritten to restrain from interpretation in the results part.

Lines 391-392: Simplified pollen diagram: please changes the colors as its done usually in pollen data as follows: Orange is for steppic and/ or NAP and green colors are for arboreal pollen. How do you calculate your pollen sum? With or without *Pinus*? Most important: remove the part MIS 11c: not discussed in the texte, not important for this paper!

Colors in the figure were adjusted accordingly and MIS 11c section was removed – please see modified Figure 3.

Pollen sum was calculated including Pinus.

**Line 393: July in the text – homogenise.**

This was corrected accordingly – as far as chironomid-based temperature reconstruction is concerned, "July" is used throughout the text and as section title.

Line 396: I agree, you have to remove the pollen characterising the MIS 11c.

This was corrected accordingly.

Line 405: Remove "Air".

This was corrected accordingly.

Line 414: The figure legend needs to be written in a larger font — it is currently too small. I suggest splitting this into two separate figures: one figure to show the variations in chironomid assemblages only. The chironomid-based summer temperatures can be added to the existing figure showing pollen-based temperature reconstructions, as is already partially the case.

Figure 3 was edited taking into account suggestions above – the legend font was enlarged and July temperature reconstruction was removed and is present now in the Figure 4 (and additionally in the new Figure 5 for comparison).

Line 420: "The pollen-based climate reconstructions from the Krępa sediment core reveal a distinct climate variability throughout MIS 419 11b, in general following the vegetation-indicated stadial-interstadial transitions" Please avoid: for me its circular reasoning as you did the transfer function on the same pollen record.

This sentence was deleted in the revised version of the manuscript.

Line 424: Move this Table in the supplémentary. Could you check carefully your methods and results? Usually, we kept at least 3 components for WAPLS and we never kept 2 analogs (k?) for the MAT (it's too low). Please check!

The table was moved to the Supplement as Supplement Table 1. For changes in pollen-based reconstruction methods – see our response to the main comment #4.

Line 427: Are the two pollen-based methods in agreement? This should be clarified, as it is important — especially since WAPLS is used for both chironomid and pollen reconstructions.

This issue was clarified in the revised manuscript:

"Pollen-based climate reconstructions from the Krępa sediment core reveal distinct climate variability throughout MIS 11b, reflecting stadial—interstadial transitions (Fig. 4). Conducted cross-validation indicated that MAT reconstructions achieved the highest predictive skill, particularly for the reconstructed temperatures (Table 2). The two pollen-based methods show broadly similar trends across all zones, with MAT generally producing warmer summer values than WAPLS except in KR-12c. Where chironomid data are available, pollen-based MTWA reconstructions reproduce similar patterns, with differences falling within their respective uncertainty ranges. Among the two pollen-based models, MAT generally corresponds better to the chironomid WAPLS reconstructions, showing overall closer alignment in reconstructed summer temperatures.

WA-PLS reconstructions were somewhat less robust, especially for precipitation, while the TANN and MTWA estimates still showed moderate predictive ability (Tab. 2). Reconstructed MTWA from both pollen-based methods generally ranged between approximately 15°C and 19°C. The two pollen-based methods show similar trends across all zones, with MAT often producing slightly warmer summer values than WAPLS."

Line 450: Remove the pollen percentages and keep only the climate data in the figure. Clearly indicate "pollen-inferred climate reconstruction" on the figure. Do not show precipitation as histograms — use line curves instead. Include the modern (present-day) value for each climate parameter. Also, there are 14 samples for the chironomid data, not 13 as stated in the text.

Figure 4 was edited according to the suggestions.

**Line 461: The sentence may be too long.**

This sentence was divided in two and corrected as follows:

"Because of the excellent preservation of their larvae's head capsules in lake and peat bog sediments, the analysis of their subfossil remains offers the possibility to reconstruct environmental and climatic changes in the past. This includes quantitative reconstructions of the average July air temperature and the trophic state of the inhabited water body as well as the type and dynamics of the lake, the water pH, and microhabitats."

Line 464: "climate" → "temperature".

This was corrected accordingly.

Line 465: "The basic principle of palaeoecological reconstructions is geological actuality implying that processes taking place on Earth 465 in the past were the same as today (Krzeminski and Jarzembowski, 1999)." – not clear.

"Geological actuality" was replaced with "uniformitarianism".

Line 535: "Chironomid-inferred temperature reconstruction [...] pollen-based climate reconstruction".

This was corrected accordingly.

Line 537: Age? not clear! Not possible to build an age model?

Please see our response to the main comment #3.

Lines 548-549: How do you know that these PAZ correspond to these colds events of MIS 11c?

This sentence was removed from the revised version of the manuscript.

Line 555: You may add Lapellegerie et al., 2024

Respective reference was added to the text.

Line 558: Would be interesting to compare your results (MIS 11b) with cold part just after the Eemian.

Information concerning Chironomidae assemblages from Siberia was added to the text:

"A similar phenomenon has so far only been observed in the Laptev Sea region (Arctic Siberia), where Chironomidae also appear only in the cold period after the Eemian Interglacial, when the site was surrounded by wet grass-sedge shrub tundra period (Andreev et al., 2004). Assemblages from this site consist mostly of unidentified Tanytarsini individuals, eutrophic *Chironomus plumosus* and semi-aquatic taxa such as *Limnophyes/Paralimnophyes*, *Smittia* and *Paraphaenocladius*. The three species from the latter group were not identified at Krępa as opposed to *Chironomus plumosus* and Tanytarsini."

**Line 563: What about the rapid events depicted in your reconstruction?**

There are no rapid changes in MTWA reconstructed by both MAT and WA-PLS during LPAZ 12a.

Line 581: "Our MAT and WA-PLS reconstructions support this shift" – Avoid this circular reasoning: climate reconstruction are based on the same pollen record!

This sentence was deleted.

Line 588: "The gradual cooling indicated by our chironomid- and pollen-based reconstructions during subsequent LPAZ KR-13a is 588 consistent with the presence of sparse Betula forests at the onset of this zone." – circular reasoning.

This part was reformulated as follows:

"The gradual cooling indicated by our chironomid-based reconstruction during LPAZ KR-13a is consistent with the presence of sparse *Betula* forests at the onset of this zone. Pollen-based reconstructions suggest that MTWA remained relatively mild (~17.3 °C MAT, ~15.3 °C WA-PLS), closely aligning with the chironomid-inferred mean Tiul-Ch value of ~17.5 °C for this interval."

Line 596-598 and 609-612: I would like to see a climate synthesis figure with the values of your study and results of these studies.

Suggested figure was added to the revised version of the manuscript as Figure 5 – please see also our response to the main comment #2.

Line 599: "These Mediterranean records indicate generally warm conditions during MIS 11b, punctuated by recurrent 598 cooling and drying events that led to repeated forest contractions." – avoid, circular reasoning.

This was corrected accordingly.

Lines 695-697: Replace with: ter Braak, C. J. F. and Juggins, S.: Weighted averaging partial least squares regression (WA-PLS): an improved method for reconstructing environmental variables from species assemblages, Hydrobiologia, 269–270, 485–502, https://doi.org/10.1007/BF00028046, 1993.

This was corrected accordingly.

---

## Editor Decision (ED2)

**Dear Authors.**

I would like to thank Dr. Polkowski and co-authors for submitting this new version of their manuscript "Chironomid- and pollen-based quantitative climate reconstructions I for the post-Holsteinian (MIS 11b) in Central Europe". All the comments have been addressed and this version is much better.

I have checked carefully this version, and I will accept it with minor changes (see below).

- **Abstract:** Your abstract should be more informative; a comparison between the chironomids -based and the pollen-based climate reconstruction is lacking; I would like you add more details on the key results obtained.

**-Introduction**

-line 63: "Hence, knowledge about climatic conditions at this time is mainly derived from pollen data", could you be more precise: "is mainly derived from Southern Europe pollen data"

-line 66: "and marine core from ODP site 976 in the Alboran Sea (Sassoon et al., 2023, 2025)". Could you replace by: "and marine cores from ODP site 976 in the Alboran Sea (Sassoon et al., 2023, 2025) or from the North Atlantic off the Iberian coast (Oliveira et al., 2016)".

-line 100: For clarity, could you move the sentence "Holsteinian Interglacial was followed by gradual cooling period (MIS 11b) which resulted in annual temperature decline and forest 100 contractions (Tzedakis et al., 2006; Kousis et al., 2018; Hrynowiecka et al., 2019; Sassoon et al., 2025)." to the line 111, just before "MIS 11b brought the AP percentages"?

-line 115: Could you move the sentences: "The warm phase of the Holsteinian Interglacial was also confirmed by oxygen isotope analyses on endogenic lake carbonates 115 (Nitychoruk et al., 2005) and snail shells (Szymanek, 2018). These showed significant changes in climatic conditions 116 throughout the Holsteinian Interglacial, during which, continental and maritime influences intertwined in Central Europe. 117 Continental influences resulted in a shortened vegetation period with long winters, whilst the opposite occurred under maritime 118 influence, i.e. the vegetation period was significantly longer, temperatures were milder and precipitation rates were higher, 119 also reflected by the appearance of stenothermal plant species (Nitychoruk et al., 2005)" at the end of the line 110.

-line 122: Replace "based on chironomid and pollen analyses." by ""based on chironomid and pollen analyses of the Krepa core (southeastern Poland)"

-line 124: Replace "section of the core" by "section of the Krępa core"

**-Study site and methods**

-line 129: Please replace "2.1 Study area " by "2.1 Study area, coring and lithology"; please add few sentences on the dating (lack of age model).

-line 194: Replace" transfer function" by "regression"; RMSEP: give the name not just the acronym

-line 219: "All reconstructed climatic factors were based on modern data sourced from the Northern Hemisphere database compiled by Herzschuh et al. (2023a, b)". You can delete this sentence as the same sentence is written line 232 (better to keep the second one): "Modern pollen data used in the reconstructions were sourced from the Northern Hemisphere database 232 compiled by Herzschuh et al. (2023a, b)".

-Results and interpretation: just "Results", keep the interpretation for the discussion

- lines 247-275: "Corynocera ambigua is a species often described as cold-adapted oligotrophic (Fjellberg, 1972; Pinder and Reiss, 1983; Walker and Mathewes, 1988; Brooks et al., 2007; Luoto et al., 2008; van Asch et al., 2012), inhabiting shallow lakes in arctic and subarctic regions, ... Both Chironomus anthracinus-type and Corynocera ambigua are found in stratified lakes (e.g., Saether, 1979; Heiri, 2004). As we observe in our record, both species are relatively resistant to unfavourable environmental conditions, so possess a wide range of conditions in which they can occur. "

The same sentences (line 247-275) appear also in the discussion part: better to remove it to the results part and keep it in the discussion.

-line 294: "Figure 2: Stratigraphic diagram of the Chironomidae assemblages. Caption: Chironomidae species are presented as counted numbers of specimens." Some details are lacking, please add in your caption: Chironomidae counts before merging; simplified pollen diagram at the right and for the pollen diagram caption, see figure 3; LPAZ=...

-line 300: "In this period, average July temperatures still ranged between 17 and 19 °C before rapidly dropping to about 16 300 °C and increasing again to 18-20 °C in LPAZ KR-12b." just refers to the figure 4

-Table 1: the LPAZ in function of the different depths does not correspond to the LPAZ of the figure 2; please chek and correct it in the table

-line 324: Please replace "3.3 Vegetation changes during the Early Saalian Glaciation at Krępa site" by "3.3 Vegetation changes during the Early Saalian Glaciation at Krępa site and comparison with Chironomids assemblages changes"

-line 331: "Corynocera ambigua » in italic

-line 357 : "The two pollen-based... WA-PLS" please delete, the same sentence is written line 350, 351

**-Discussion**

-lines 533-534: Replace by "Contrary to the chironomids-based temperature reconstruction, the pollen-based reconstructions using MAT and WAPLS methods provide continuous temperature and precipitation records."

-line 537: "This is in agreement with the observed dominance of *Pinus* forests with some admixed *Picea* during this phase, reflecting more humid but not necessarily warmer conditions (Caudullo et al., 2016)." If your sentence refers to the Krespa pollen sequence, delete this sentence as its a circular reasoning

-line 562: replace "Tann" by "TANN", check also in other parts of the discussion (Pann also)

-line 584: ODP 976 not 986

**-conclusion**

-line 618: "These climatic trends coincide with marked vegetation changes, including forest retreat and a rise in herbaceous taxa during colder phases" please avoid: circular reasoning as the climate reconstruction is based on the same pollen sequence

Best regards,

Odile Peyron, Editor for Climate of the Past

---

## Author Response (AR3)

We greatly appreciate the Editor's thorough review and constructive suggestions. Here we provide the point-by-point response to all the comments:

Abstract: Your abstract should be more informative; a comparison between the chironomids -based and the pollen-based climate reconstruction is lacking; I would like you add more details on the key results obtained.

The comparison between chironomid-based and pollen-based climate reconstructions was added including MTWA values. Moreover, other elements from the latter reconstruction such as winter temperatures (MTCO) and annual precipitation sum (PANN) are now briefly presented in the revised abstract.

**Introduction**

Line 63: "Hence, knowledge about climatic conditions at this time is mainly derived from pollen data", could you be more precise: "is mainly derived from Southern Europe pollen data".

This was corrected accordingly.

Line 66: "and marine core from ODP site 976 in the Alboran Sea (Sassoon et al., 2023, 2025)". Could you replace by: "and marine cores from ODP site 976 in the Alboran Sea (Sassoon et al., 2023, 2025) or from the North Atlantic off the Iberian coast (Oliveira et al., 2016)".

This was corrected accordingly.

Line 100: For clarity, could you move the sentence "Holsteinian Interglacial was followed by gradual cooling period (MIS 11b) which resulted in annual temperature decline and forest 100 contractions (Tzedakis et al., 2006; Kousis et al., 2018; Hrynowiecka et al., 2019; Sassoon et al., 2025)." to the line 111, just before "MIS 11b brought the AP percentages"?

This was corrected accordingly.

Line 115: Could you move the sentences: "The warm phase of the Holsteinian Interglacial was also confirmed by oxygen isotope analyses on endogenic lake carbonates 115 (Nitychoruk et al., 2005) and snail shells (Szymanek, 2018). These showed significant changes in climatic conditions 116 throughout the Holsteinian Interglacial, during which, continental and maritime influences intertwined in Central Europe. 117 Continental influences resulted in a shortened vegetation period with long winters, whilst the opposite occurred under maritime 118 influence, i.e. the vegetation period was significantly longer, temperatures were milder and precipitation rates were higher, 119 also reflected by the appearance of stenothermal plant species (Nitychoruk et al., 2005)" at the end of the line 110.

This was corrected accordingly.

Line 122: Replace "based on chironomid and pollen analyses." by ""based on chironom tym razem id and pollen analyses of the Krepa core (southeastern Poland)".

This was corrected accordingly.

Line 124: Replace "section of the core" by "section of the Krepa core".

This was corrected accordingly.

**Study site and methods**

Line 129: Please replace "2.1 Study area " by "2.1 Study area, coring and lithology"; please add few sentences on the dating (lack of age model).

Respective changes were made in the section title. The paragraph concerning lack of depth-age model was added and reads as follows:

"Due to the inability to apply radiocarbon dating (e.g. 14C) and the challenges in developing an age—depth model, the direct dating of Holstein interglacial sediments is highly limited. In this context, palynology plays a key role, as pollen analysis enables biostratigraphic comparison between sites. Vegetation changes that occurred during the Holstein interglacial show a relatively uniform pattern across Central Europe from boreal phases to the development of thermophilus deciduous forests (Nitychoruk et al., 2005; Koutsodendris et al., 2010; Hrynowiecka and Winter, 2016) and cooling period (MIS 11b) thereafter (Hrynowiecka et al., 2019). Thanks to the

repeatability of this vegetational succession, it is possible to correlate sediment profiles from different locations and assign them to a common stratigraphic framework. Thus, palynology becomes the primary tool for reconstructing and comparing environmental records from this period, despite the lack of precise absolute dating."

Line 194: Replace" transfer function" by "regression"; RMSEP: give the name not just the acronym.

This was corrected accordingly.

Line 219: "All reconstructed climatic factors were based on modern data sourced from the Northern Hemisphere database compiled by Herzschuh et al. (2023a, b)". You can delete this sentence as the same sentence is written line 232 (better to keep the second one): "Modern pollen data used in the reconstructions were sourced from the Northern Hemisphere database 232 compiled by Herzschuh et al. (2023a, b)".

The first sentence was deleted as suggested.

Results and interpretation: just "Results", keep the interpretation for the discussion.

This was corrected accordingly.

Lines 247-275: "Corynocera ambigua is a species often described as cold-adapted oligotrophic (Fjellberg, 1972; Pinder and Reiss, 1983; Walker and Mathewes, 1988; Brooks et al., 2007; Luoto et al., 2008; van Asch et al., 2012), inhabiting shallow lakes in arctic and subarctic regions, ... Both Chironomus anthracinus-type and Corynocera ambigua are found in stratified lakes (e.g., Saether, 1979; Heiri, 2004). As we observe in our record, both species are relatively resistant to unfavourable environmental conditions, so possess a wide range of conditions in which they can occur." The same sentences (line 247-275) appear also in the discussion part: better to remove it to the results part and keep it in the discussion.

According to the suggestion, these sentences was removed from Results section.

Line 294: "Figure 2: Stratigraphic diagram of the Chironomidae assemblages. Caption: Chironomidae species are presented as counted numbers of specimens." Some details are lacking, please add in your caption: Chironomidae counts before merging; simplified pollen diagram at the right and for the pollen diagram caption, see figure 3; LPAZ=...

This fragment was edited as follows:

"Figure 2: Stratigraphic diagram of the Chironomidae assemblages with simplified pollen diagram (left) and Local Pollen Assemblages Zones (LPAZ). Caption: Chironomidae species are presented as counted numbers of specimens before merging."

Line 300: "In this period, average July temperatures still ranged between 17 and 19  $^{\circ}$ C before rapidly dropping to about 16 300  $^{\circ}$ C and increasing again to 18-20  $^{\circ}$ C in LPAZ KR-12b." just refers to the figure 4

Respective reference was added.

Table 1: the LPAZ in function of the different depths does not correspond to the LPAZ of the figure 2; please check and correct it in the table.

Table 1 was corrected accordingly.

Line 324: Please replace "3.3 Vegetation changes during the Early Saalian Glaciation at Krępa site" by "3.3 Vegetation changes during the Early Saalian Glaciation at Krępa site and comparison with Chironomids assemblages changes".

This was corrected accordingly.

Line 331: "Corynocera ambigua » in italic.

This was corrected accordingly.

Line 357 : "The two pollen-based... WA-PLS" please delete, the same sentence is written line 350, 351.

This sentence was deleted in the revised version of the manuscript.

**Discussion**

Lines 533-534: Replace by "Contrary to the chironomids-based temperature reconstruction,

the pollen-based reconstructions using MAT and WAPLS methods provide continuous

temperature and precipitation records."

This was corrected accordingly.

Line 537: "This is in agreement with the observed dominance of Pinus forests with some

admixed Picea during this phase, reflecting more humid but not necessarily warmer

conditions (Caudullo et al., 2016)." If your sentence refers to the Krespa pollen sequence,

delete this sentence as it's a circular reasoning.

This sentence was deleted according to the suggestion.

Line 562: replace "Tann" by "TANN", check also in other parts of the discussion (Pann also).

The manuscript was inspected again and mistakes of this sort were corrected.

Line 584: ODP 976 not 986

This was corrected accordingly.

Conclusion

Line 618: "These climatic trends coincide with marked vegetation changes, including forest

retreat and a rise in herbaceous taxa during colder phases" please avoid: circular reasoning

as the climate reconstruction is based on the same pollen sequence.

This sentence was removed from revised version of the manuscript.